# THE INFLUENCE OF LEARNING RULE ON REPRESENTATION DYNAMICS IN WIDE NEURAL NETWORKS

**Blake Bordelon** & **Cengiz Pehlevan**
School of Engineering and Applied Science
Harvard University
Cambridge, MA 02138, USA
`{blake_bordelon,cpehlevan}@g.harvard.edu`

## ABSTRACT

It is unclear how changing the learning rule of a deep neural network alters its learning dynamics and representations. To gain insight into the relationship between learned features, function approximation, and the learning rule, we analyze infinite-width deep networks trained with gradient descent (GD) and biologically-plausible alternatives including feedback alignment (FA), direct feedback alignment (DFA), and error modulated Hebbian learning (Hebb), as well as gated linear networks (GLN). We show that, for each of these learning rules, the evolution of the output function at infinite width is governed by a time varying effective neural tangent kernel (eNTK). In the lazy training limit, this eNTK is static and does not evolve, while in the rich mean-field regime this kernel's evolution can be determined self-consistently with dynamical mean field theory (DMFT). This DMFT enables comparisons of the feature and prediction dynamics induced by each of these learning rules. In the lazy limit, we find that DFA and Hebb can only learn using the last layer features, while full FA can utilize earlier layers with a scale determined by the initial correlation between feedforward and feedback weight matrices. In the rich regime, DFA and FA utilize a temporally evolving and depth-dependent NTK. Counterintuitively, we find that FA networks trained in the rich regime exhibit more feature learning if initialized with smaller correlation between the forward and backward pass weights. GLNs admit a very simple formula for their lazy limit kernel and preserve conditional Gaussianity of their preactivations under gating functions. Error modulated Hebb rules show very small task-relevant alignment of their kernels and perform most task relevant learning in the last layer.

## 1 INTRODUCTION

Deep neural networks have now attained state of the art performance across a variety of domains including computer vision and natural language processing (Goodfellow et al., 2016; LeCun et al., 2015). Central to the power and transferability of neural networks is their ability to flexibly adapt their layer-wise internal representations to the structure of the data distribution during learning.

In this paper, we explore how the learning rule that is used to train a deep network affects its learning dynamics and representations. Our primary motivation for studying different rules is that exact gradient descent (GD) training with the back-propagation algorithm is thought to be biologically implausible (Crick, 1989). While many alternatives to standard GD training were proposed (Whittington & Bogacz, 2019), it is unclear how modifying the learning rule changes the functional inductive bias and the learned representations of the network. Further, understanding the learned representations could potentially offer more insight into which learning rules account for representational changes observed in the brain (Poort et al., 2015; Kriegeskorte & Wei, 2021; Schumacher et al., 2022). Our current study is a step towards these directions.

The alternative learning rules we study are error modulated Hebbian learning (Hebb), Feedback alignment (FA) (Lillicrap et al., 2016) and direct feedback alignment (DFA) (Nøkland, 2016). These rules circumvent one of the biologically implausible features of GD: the weights used in the backward pass computation of error signals must be dynamically identical to the weights used on the

forward pass, known as the weight transport problem. Instead, FA and DFA algorithms compute an approximate backward pass with independent weights that are frozen through training. Hebb rule only uses a global error signal. While these learning rules do not perform exact GD, they are still able to evolve their internal representations and eventually fit the training data. Further, experiments have shown that FA and DFA can scale to certain problems such as view-synthesis, recommendation systems, and small scale image problems (Launay et al., 2020), but they do not perform as well in convolutional architectures with more complex image datasets (Bartunov et al., 2018). However, significant improvements to FA can be achieved if the feedback-weights have partial correlation with the feedforward weights (Xiao et al., 2018; Moskovitz et al., 2018; Boopathy & Fiete, 2022).

We also study gated linear networks (GLNs), which use frozen gating functions for nonlinearity (Fiat et al., 2019). Variants of these networks have bio-plausible interpretations in terms of dendritic gates (Sezener et al., 2021). Fixed gating can mitigate catastrophic forgetting (Veness et al., 2021; Budden et al., 2020) and enable efficient transfer and multi-task learning Saxe et al. (2022).

Here, we explore how the choice of learning rule modifies the representations, functional biases and dynamics of deep networks at the infinite width limit, which allows a precise analytical description of the network dynamics in terms of a collection of evolving kernels. At infinite width, the network can operate in the *lazy* regime, where the feature embeddings at each layer are constant through time, or the *rich/feature-learning* regime (Chizat et al., 2019; Yang & Hu, 2021; Bordelon & Pehlevan, 2022). The richness is controlled by a scalar parameter related to the initial scale of the output function.

In summary, our novel contributions are the following:

1. We identify a class of learning rules for which function evolution is described by a dynamical effective Neural Tangent Kernel (eNTK). We provide a dynamical mean field theory (DMFT) for these learning rules which can be used to compute this eNTK. We show both theoretically and empirically that convergence to this DMFT occurs at large width $N$ with error $O(N^{-1/2})$.

2. We characterize precisely the inductive biases of infinite width networks in the lazy limit by computing their eNTKs at initialization. We generalize FA to allow partial correlation between the feedback weights and initial feedforward weights and show how this alters the eNTK.

3. We then study the *rich* regime so that the features are allowed to adapt during training. In this regime, the eNTK is dynamical and we give a DMFT to compute it. For deep linear networks, the DMFT equations close algebraically, while for nonlinear networks we provide a numerical procedure to solve them.

4. We compare the learned features and dynamics among these rules, analyzing the effect of richness, initial feedback correlation, and depth. We find that rich training enhances gradient-pseudogradient alignment for both FA and DFA. Counterintuitively, smaller initial feedback correlation generates more dramatic feature evolution for FA. The GLN networks have dynamics comparable to GD, while Hebb networks, as expected, do not exhibit task relevant adaptation of feature kernels, but rather evolve according to the input statistics.

## 1.1 RELATED WORKS

GLNs were introduced by Fiat et al. (2019) as a simplified model of ReLU networks, allowing the analysis of convergence and generalization in the lazy kernel limit. Veness et al. (2021) provided a simplified and biologically-plausible learning rule for deep GLNs which was extended by Budden et al. (2020) and provided an interpretation in terms of dendritic gating Sezener et al. (2021). These works demonstrated benefits to continual learning due to the fixed gating. Saxe et al. (2022) derived exact dynamical equations for a GLN with gates operating at each node and each edge of the network graph. Krishnamurthy et al. (2022) provided a theory of gating in recurrent networks.

Lillicrap et al. (2016) showed that, in a two layer linear network the forward weights will evolve to align to the frozen feedback weights under the FA dynamics, allowing convergence of the network to a loss minimizer. This result was extended to deep networks by Frenkel et al. (2019), who also introduced a variant of FA where only the direction of the target is used. Refinetti et al. (2021) studied DFA in a two-layer student-teacher online learning setup, showing that the network first undergoes an alignment phase before converging to one the degenerate global minima of the loss. They argued that FA's worse performance in CNNs is due to the inability of the forward pass gradients to align under the block-Toeplitz connectivity strucuture that arises from enforced weight sharing (d'Ascoli et al., 2019). Garg & Vempala (2022) analyzed matrix factorization with FA,

proving that, when overparameterized, it converges to a minimizer under standard conditions, albeit more slowly than GD. Cao et al. (2020) analyzed the kernel and loss dynamics of linear networks trained with learning rules from a space that includes GD, contrastive Hebbian, and predictive coding rules, showing strong dependence of hierarchical representations on learning rule.

Recent works have utilized DMFT techniques to analyze typical performance of algorithms trained on high-dimensional random data (Agoritsas et al., 2018; Mignacco et al., 2020; Celentano et al., 2021; Gerbelot et al., 2022). In the present work, we do not average over random datasets, but rather over initial random weights and treat data as an input to the theory. Wide NNs have been analyzed at infinite width in both lazy regimes with the NTK (Jacot et al., 2018; Lee et al., 2019) and rich feature learning regimes (Mei et al., 2018). In the feature learning limit, the evolution of kernel order parameters have been obtained with both Tensor Programs framework (Yang & Hu, 2021) and with DMFT (Bordelon & Pehlevan, 2022). Song et al. (2021) recently analyzed the lazy infinite width limit of two layer networks trained with FA and weight decay, finding that only one layer effectively contributes to the two-layer NTK. Boopathy & Fiete (2022) proposed alignment based learning rules for networks at large width in the lazy regime, which performs comparably to GD and outperform standard FA. Their *Align-Ada* rule corresponds to our $\rho$-FA with $\rho = 1$ in *lazy* large width networks.

## 2 EFFECTIVE NEURAL TANGENT KERNEL FOR A LEARNING RULE

We denote the output of a neural network for input $\boldsymbol{x}_\mu \in \mathbb{R}^D$ as $f_\mu$. For concreteness, in the main text we will focus on scalar targets $f_\mu \in \mathbb{R}$ and MLP architectures. Other architectures such as multi-class outputs and CNN architectures with infinite channel count can also be analyzed as we show in the Appendix C. For the moment, we let the function be computed recursively from a collection of weight matrices $\boldsymbol{\theta} = \text{Vec}\{\boldsymbol{W}^0, \boldsymbol{W}^1, ..., \boldsymbol{w}^L\}$ in terms of preactivation vectors $\boldsymbol{h}_\mu^\ell \in \mathbb{R}^N$ where,

$$f_\mu = \frac{1}{\gamma_0 N} \boldsymbol{w}^L \cdot \phi(\boldsymbol{h}_\mu^L) \ , \ \boldsymbol{h}_\mu^{\ell+1} = \frac{1}{\sqrt{N}} \boldsymbol{W}^\ell \phi(\boldsymbol{h}_\mu^\ell) \ , \ \boldsymbol{h}_\mu^1 = \frac{1}{\sqrt{D}} \boldsymbol{W}^0 \boldsymbol{x}_\mu \tag{1}$$

where nonlinearity $\phi$ is applied element-wise. The scalar parameter $\gamma_0$ controls how rich the network training is: small $\gamma_0$ corresponds to lazy learning while large $\gamma_0$ generates large changes to the features (Chizat et al., 2019). For gated linear networks, we follow Fiat et al. (2019) and modify the forward pass equations by replacing $\phi(\boldsymbol{h}_\mu^\ell)$ with a multiplicative gating function $\dot{\phi}(\boldsymbol{m}_\mu^\ell)\boldsymbol{h}_\mu^\ell$ where gating variables $\boldsymbol{m}_\mu^\ell = \frac{1}{\sqrt{D}} \boldsymbol{M}^\ell \boldsymbol{x}_\mu$ are fixed through training with $M_{ij} \sim \mathcal{N}(0, 1)$. To minimize loss $\mathcal{L} = \sum_\mu \ell(f_\mu, y_\mu)$, we consider learning rules to the parameters $\boldsymbol{\theta}$ of the form

$$\frac{d}{dt} \boldsymbol{w}^L = \gamma_0 \sum_\mu \phi(\boldsymbol{h}_\mu^L(t)) \Delta_\mu \ , \ \frac{d}{dt} \boldsymbol{W}^\ell = \frac{\gamma_0}{\sqrt{N}} \sum_\mu \Delta_\mu \, \tilde{\boldsymbol{g}}_\mu^{\ell+1} \, \phi(\boldsymbol{h}_\mu^\ell)^\top \ , \ \frac{d}{dt} \boldsymbol{W}^0 = \frac{\gamma_0}{\sqrt{D}} \sum_\mu \Delta_\mu \tilde{\boldsymbol{g}}_\mu^1 \boldsymbol{x}_\mu^\top \tag{2}$$

where the error signal is $\Delta_\mu(t) = -\frac{\partial \mathcal{L}}{\partial f_\mu}|_{f_\mu(t)}$. The last layer weights $\boldsymbol{w}^L$ are always updated with their true gradient. This corresponds to the biologically-plausible and local delta-rule, which merely correlates the error signals $\Delta_\mu$ and the last layer features $\phi(\boldsymbol{h}_\mu^L)$ (Widrow & Hoff, 1960). In intermediate layers, the *pseudo-gradient* vectors $\tilde{\boldsymbol{g}}_\mu^\ell$ are determined by the choice of the learning rule. For concreteness, we provide below the recursive definitions of $\tilde{\boldsymbol{g}}^\ell$ for our five learning rules of interest.

$$\tilde{\boldsymbol{g}}_\mu^\ell = \begin{cases} \dot{\phi}(\boldsymbol{h}_\mu^\ell) \odot \left[ \frac{1}{\sqrt{N}} \boldsymbol{W}^\ell(t)^\top \tilde{\boldsymbol{g}}_\mu^{\ell+1} \right] \ , \ \tilde{\boldsymbol{g}}_\mu^L = \dot{\phi}(\boldsymbol{h}_\mu^L) \odot \boldsymbol{w}^L & \text{GD} \\ \dot{\phi}(\boldsymbol{h}_\mu^\ell) \odot \left[ \frac{1}{\sqrt{N}} \left( \rho \boldsymbol{W}^\ell(0) + \sqrt{1-\rho^2} \tilde{\boldsymbol{W}}^\ell \right)^\top \tilde{\boldsymbol{g}}^{\ell+1} \right] \ , \ \tilde{W}_{ij}^\ell \sim \mathcal{N}(0,1) & \rho\text{-FA} \\ \dot{\phi}(\boldsymbol{h}_\mu^\ell) \odot \tilde{\boldsymbol{z}}^\ell \ , \ \tilde{z}_i^\ell \sim \mathcal{N}(0,1) & \text{DFA} \\ \dot{\phi}(\boldsymbol{m}_\mu^\ell) \odot \left[ \frac{1}{\sqrt{N}} \boldsymbol{W}^\ell(t)^\top \tilde{\boldsymbol{g}}_\mu^{\ell+1} \right] \ , \ \tilde{\boldsymbol{g}}^L = \dot{\phi}(\boldsymbol{m}_\mu^\ell) \odot \boldsymbol{w}^L(t) & \text{GLN} \\ \Delta_\mu(t) \phi(\boldsymbol{h}_\mu^\ell(t)) & \text{Hebb} \end{cases} \tag{3}$$

While GD uses the instantaneous feedforward weights on the backward pass, $\rho$-FA uses the weight matrices which do not evolve throughout training. These weights have correlation $\rho$ with the initial forward pass weights $\boldsymbol{W}^\ell(0)$. This choice is motivated by the observation that partial correlation between forward and backward pass weights at initialization can improve training (Liao et al., 2016;

Xiao et al., 2018; Moskovitz et al., 2018), though the cost is partial weight transport at initialization. However, we consider partial correlation at initialization more biologically plausible than the demanding weight transport at each step of training, like in GD. For DFA, the weight vectors $\tilde{z}^\ell$ are sampled randomly at initialization and do not evolve in time. For GLN, the gating variables $m_\mu^\ell$ are frozen through time but the exact feedforward weights are used in the backward pass. Lastly, we modify the classic Hebb rule (Hebb, 1949) to get $\Delta W^\ell \propto \sum_\mu \Delta_\mu(t)^2 \phi(h_\mu^{\ell+1}) \phi(h_\mu^\ell)^\top$, which weighs each example by its current error. Unlike standard Hebbian updates, this learning rule gives stable dynamics without regularization (App. G). For all rules, the evolution of the function is determined by a time-dependent eNTK $K_{\mu\nu}$ which is defined as

$$\frac{\partial f_\mu}{\partial t} = \frac{\partial f_\mu}{\partial \boldsymbol{\theta}} \cdot \frac{d\boldsymbol{\theta}}{dt} = \sum_\nu \Delta_\nu K_{\mu\nu}(t, t), \quad K_{\mu\nu}(t, s) = \sum_{\ell=0}^L \tilde{G}_{\mu\nu}^{\ell+1}(t, s) \Phi_{\mu\nu}^\ell(t, s)$$

$$\tilde{G}_{\mu\nu}^\ell(t, s) = \frac{1}{N} \boldsymbol{g}_\mu^\ell(t) \cdot \tilde{\boldsymbol{g}}_\nu^\ell(s), \quad \Phi_{\mu\nu}^\ell(t, s) = \frac{1}{N} \phi(\boldsymbol{h}_\mu^\ell(t)) \cdot \phi(\boldsymbol{h}_\nu^\ell(s)), \tag{4}$$

where the base cases $\tilde{G}_{\mu\nu}^{L+1}(t, s) = 1$ and $\Phi_{\mu\nu}^0(t, s) = \frac{1}{D} \boldsymbol{x}_\mu \cdot \boldsymbol{x}_\nu$ are time-invariant. The kernel $\tilde{G}^\ell$ computes an inner product between the true gradient signals $\boldsymbol{g}_\mu^\ell = \gamma_0 N \frac{\partial f_\mu}{\partial \boldsymbol{h}_\mu^\ell}$ and the pseudo-gradient $\tilde{\boldsymbol{g}}_\nu^\ell$ which is set by the chosen learning rule. We see that because $\tilde{G}^\ell$ is not necessarily symmetric, $K$ is also not necessarily symmetric. The matrix $\tilde{G}^\ell$ quantifies pseudo-gradient / gradient alignment.

## 3 DYNAMICAL MEAN FIELD THEORY FOR VARIOUS LEARNING RULES

For each of these learning rules considered, the infinite width $N \to \infty$ limit of network learning can be described by a dynamical mean field theory (DMFT) (Bordelon & Pehlevan, 2022). At infinite width, the dynamics of the kernels $\Phi^\ell$ and $\tilde{G}^\ell$ become deterministic over random Gaussian initialization of parameters $\boldsymbol{\theta}$. The activity of neurons in each layer become i.i.d. random variables drawn from a distribution defined by these kernels, which themselves are averages over these single-site distributions. Below, we provide DMFT formulas which are valid for all of our learning rules

$$h_\mu^\ell(t) = u_\mu^\ell(t) + \gamma_0 \int_0^t ds \sum_{\nu=1}^P \left[ A_{\mu\nu}^{\ell-1}(t, s) g_\nu^\ell(s) + C_{\mu\nu}^{\ell-1}(t, s) \tilde{g}_\nu^\ell(s) + \Phi_{\mu\nu}^{\ell-1}(t, s) \Delta_\nu(s) \tilde{g}_\nu^\ell(s) \right]$$

$$z_\mu^\ell(t) = r_\mu^\ell(t) + \gamma_0 \int_0^t ds \sum_{\nu=1}^P \left[ B_{\mu\nu}^\ell(t, s) + \tilde{G}_{\mu\nu}^{\ell+1}(t, s) \Delta_\nu(s) \right] \phi(h_\nu^\ell(s)), \quad g_\mu^\ell(t) = \dot{\phi}(h_\mu^\ell(t)) z_\mu^\ell(t)$$

$$\{u_\mu^\ell(t)\} \sim \mathcal{GP}(0, \Phi^{\ell-1}), \quad \Phi_{\mu\nu}^\ell(t, s) = \langle \phi(h_\mu^\ell(t)) \phi(h_\nu^\ell(s)) \rangle, \quad A_{\mu\nu}^\ell(t, s) = \gamma_0^{-1} \left\langle \frac{\delta}{\delta r_\nu^\ell(s)} \phi(h_\mu^\ell(t)) \right\rangle$$

$$\{r_\mu^\ell(t)\} \sim \mathcal{GP}(0, \boldsymbol{G}^{\ell+1}), \quad \tilde{G}_{\mu\nu}^\ell(t, s) = \langle g_\mu^\ell(t) \tilde{g}_\nu^\ell(s) \rangle, \quad B_{\mu\nu}^\ell(t, s) = \gamma_0^{-1} \left\langle \frac{\delta}{\delta u_\nu^{\ell+1}(s)} g_\mu^{\ell+1}(t) \right\rangle$$

$$\tag{5}$$

The definitions of $\tilde{g}_\mu^\ell(t)$ depend on the learning rule and are described in Table 1. The $z_\mu^\ell(t)$ is the *pre-gradient field* defined so that $g_\mu^\ell(t) = \dot{\phi}(h_\mu^\ell(t)) z_\mu^\ell(t)$. The dependence of these DMFT equations on data comes from the base case $\Phi_{\mu\nu}^0(t, s) = \frac{1}{D} \boldsymbol{x}_\mu \cdot \boldsymbol{x}_\nu$ and error signal $\Delta_\mu = -\frac{\partial \mathcal{L}}{\partial f_\mu}$.

| Rule | GD | $\rho$-FA | DFA | GLN | Hebb |
|------|-----|-----------|-----|-----|------|
| $\tilde{g}_\mu^\ell(t)$ | $\dot{\phi}(h_\mu^\ell(t)) z_\mu^\ell(t)$ | $\dot{\phi}(h_\mu^\ell(t)) \tilde{z}_\mu^\ell(t)$ | $\dot{\phi}(h_\mu^\ell(t)) \tilde{z}^\ell$ | $\dot{\phi}(m_\mu^\ell) z_\mu^\ell(t)$ | $\Delta_\mu(t) \phi(h_\mu^\ell(t))$ |

Table 1: The field definitions for each learning rule. For $\rho$-FA, the field has definition $\tilde{z}_\mu^\ell(t) = \rho v_\mu^\ell(t) + \sqrt{1 - \rho^2} \tilde{\zeta}_\mu^\ell(t) + \gamma_0 \int_0^t ds \sum_\nu D_{\mu\nu}^\ell(t, s) \phi(h_\nu^\ell(s))$ where $\{v_\mu^\ell(t), \tilde{\zeta}_\mu^\ell(t)\}$ are Gaussian with $\langle r_\mu^\ell(t) v_\nu^\ell(s) \rangle = \tilde{G}_{\mu\nu}^{\ell+1}(t, s)$. The $\tilde{\zeta}^\ell$ field is an independent Gaussian with correlation $\left\langle \tilde{\zeta}_\mu^\ell(t) \tilde{\zeta}_\nu^\ell(s) \right\rangle = \langle \tilde{g}_\mu^{\ell+1}(t) \tilde{g}_\nu^{\ell+1}(s) \rangle = \tilde{\tilde{G}}_{\mu\nu}^{\ell+1}(t, s)$. For DFA, the $\tilde{z}^\ell$ field is static $\tilde{z}^\ell \sim \mathcal{N}(0, 1)$. For GLN, we use $\{m_\mu^\ell\} \sim \mathcal{N}(0, \boldsymbol{K}^x)$ as a gating variable. $C^\ell = 0$ except for $\rho$-FA with $\rho > 0$.

We see that, for {GD, $\rho$-FA, DFA, Hebb} the distribution of $h_\mu^\ell(t), z_\mu^\ell(t)$ are Gaussian throughout training only in the lazy $\gamma_0 \to 0$ limit for general nonlinear activation functions $\phi(h)$. However, conditional on $\{m_\mu^\ell\}$, the $\{h^\ell, z^\ell\}$ fields are all Gaussian for GLNs. For all algorithms except $\rho$-FA, $C^\ell = 0$. For $\rho$-FA we have $C_{\mu\alpha}^\ell(t,s) = \gamma_0^{-1} \left\langle \frac{\delta}{\delta v_\nu^\ell(s)} \phi(h_\mu^\ell(t)) \right\rangle$.

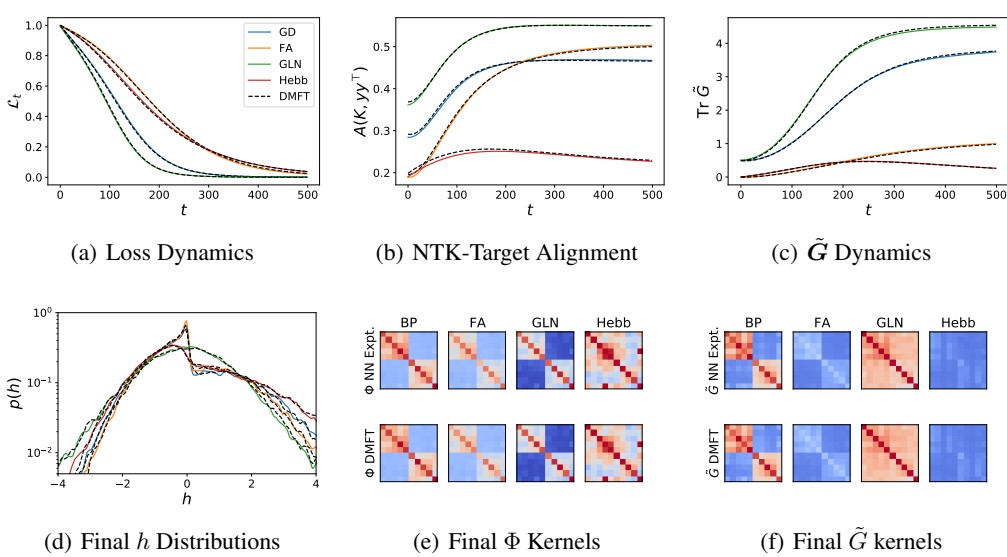

(a) Loss Dynamics      (b) NTK-Target Alignment      (c) $\tilde{G}$ Dynamics

(d) Final $h$ Distributions      (e) Final $\Phi$ Kernels      (f) Final $\tilde{G}$ kernels

Figure 1: The DMFT predicts feature dynamics of wide networks trained with gradient descent (GD), feedback alignment (FA) with $\rho = 0$, gated linear network (GLN), and a error-modulated $\beta = 1$ Hebb rule (Hebb) in the feature learning regime. (a) The loss dynamics in a two layer ($L = 1, N = 2000$) network trained with these learning rules at richness $\gamma_0 = 2$. The network is trained on a collection of $P = 10$ random vectors in $D = 50$ dimensions. (b) The cosine similarity of the eNTK with the targets $A(\boldsymbol{K}, \boldsymbol{y}\boldsymbol{y}^\top) = \frac{\boldsymbol{y}^\top \boldsymbol{K} \boldsymbol{y}}{|\boldsymbol{K}|_F |\boldsymbol{y}|^2}$ reveals increasing alignment for all algorithms. Though FA starts with the lowest alignment, its final NTK task alignment exceeds that of GD. (c) The dynamics of the gradient-pseudogradient kernel $\tilde{G}$ also reveals increasing correlation of $g$ with $\tilde{g}$. FA starts with $\tilde{G} = 0$ but $\tilde{G}$ increases to non-zero value. (d) The distribution of hidden layer preactivations after training reveals non-Gaussian statistics for both GD and FA, but approximately Gaussian statistics for GLN. (e)-(f) The final $\Phi$ and $\tilde{G}$ kernels from theory and experiment.

As described in prior results on the GD case (Bordelon & Pehlevan, 2022), the above equations can be solved self-consistently in polynomial (in train-set size $P$ and training steps $T$) time. With an estimate of the dynamical kernels $\{\Phi_{\mu\nu}^\ell(t,s), \tilde{G}_{\mu\nu}^\ell(t,s), G_{\mu\nu}^\ell(t,s)\}$, one computes the eNTK $K_{\mu\nu}(t)$ and error dynamics $\Delta_\mu(t)$. From these objects, we can sample the stochastic processes $\{h^\ell, z^\ell, \tilde{z}^\ell\}$ which can then be used to derive new refined estimates of the kernels. This procedure is repeated until convergence. This algorithm can be found in App. A. An example of such a solution is provided in Figure 1 for two layer ReLU networks trained with GD, FA, GLN, and Hebb. We show that our self-consistent DMFT accurately predicts training and kernel dynamics, as well as the density of preactivations $\{h_\mu(t)\}$ and final kernels $\{\Phi_{\mu\nu}, \tilde{G}_{\mu\nu}\}$ for each learning rule. We observe substantial differences in the learned representations (Figure 1e), all predicted by our DMFT.

### 3.1 LAZY OR EARLY TIME STATIC-KERNEL LIMITS

When $\gamma_0 \to 0$, we see that the fields $h_\mu^\ell(t)$ and $z_\mu^\ell(t)$ are equal to the Gaussian variables $u_\mu^\ell(0)$ and $r_\mu^\ell(0)$. In this limit, the eNTK $K_{\mu\nu}$ remains static and has the form summarized in Table 2 in terms of the initial feature kernels $\Phi^\ell$ and gradient kernels $G^\ell$. We derive these kernels in Appendix D.

The feature $P \times P$ matrices $\boldsymbol{\Phi}^\ell, \boldsymbol{G}^\ell$ in Table 2 are computed recursively as

$$\boldsymbol{\Phi}^\ell = \left\langle \phi(\boldsymbol{u})\phi(\boldsymbol{u})^\top \right\rangle_{\boldsymbol{u}\sim\mathcal{N}(0,\boldsymbol{\Phi}^{\ell-1})}, \quad \boldsymbol{G}^\ell = \boldsymbol{G}^{\ell+1} \odot \left\langle \dot{\phi}(\boldsymbol{u})\dot{\phi}(\boldsymbol{u})^\top \right\rangle_{\boldsymbol{u}\sim\mathcal{N}(0,\boldsymbol{\Phi}^{\ell-1})} \tag{6}$$

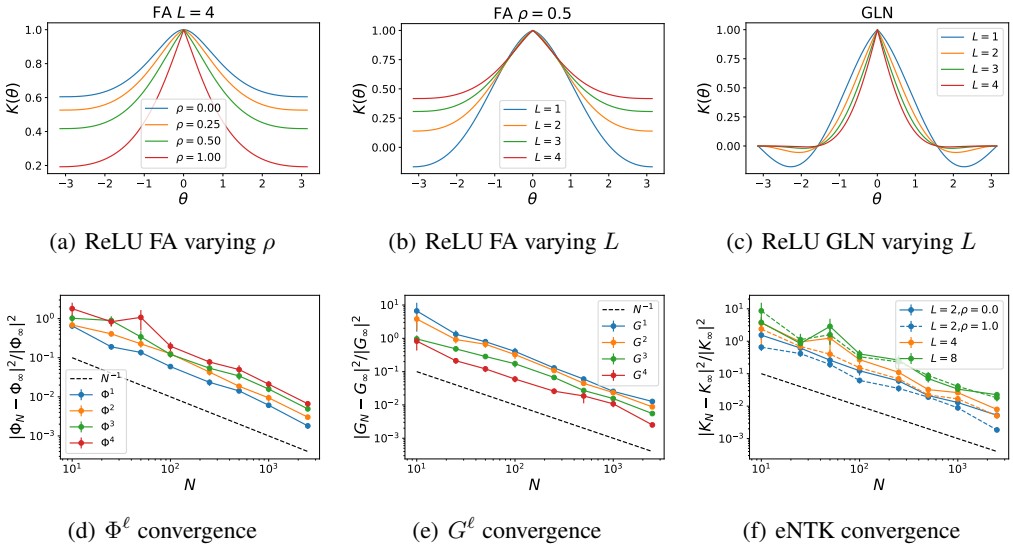

Figure 2: The lazy infinite width limits of the various learning rules can be fully summarized with their initial eNTK. (a) The kernels of $\rho$-aligned ReLU FA and ReLU GLN for inputs separated by angle $\theta$. (a) The kernels for varying $\rho$ in $\rho$-aligned FA. Larger $\rho$ has a sharper peak in the kernel around $\theta = 0$. The $\rho \to 0$ limit recovers the NNGP kernel $\Phi^L$ while the $\rho \to 1$ limit gives the back-prop NTK. (b) Deeper networks with partial alignment $\rho = 0.5$. (c) ReLU-GLN kernel sharpens with depth. (d)-(e) The relative error of the infinite width $\Phi^\ell, G^\ell$ kernels in a width $N$ ReLU neural network. The late layer $\Phi^\ell$ and early layer $G^\ell$ kernels have highest errors since finite size effects accumulate on forward and backward passes respectively. (f) Finite width corrections to eNTK are larger for small $\rho$ and large depth $L$. All square errors go as $|K_N - K_\infty|^2 \sim O_N(1/N)$.

| Rule | GD | $\rho$-FA | DFA | GLN | Hebb |
|------|-----|-----------|-----|-----|------|
| $K_{\mu\nu}$ | $\sum_{\ell=0}^{L} G_{\mu\nu}^{\ell+1}\Phi_{\mu\nu}^{\ell}$ | $\sum_{\ell=0}^{L} \rho^{L-\ell} G_{\mu\nu}^{\ell+1}\Phi_{\mu\nu}^{\ell}$ | $\Phi_{\mu\nu}^{L}$ | $\left[\left\langle \dot{\phi}(m_\mu)\dot{\phi}(m_\nu)\right\rangle\right]^L K_{\mu\nu}^x$ | $\Phi_{\mu\nu}^{L}$ |

Table 2: The initial eNTK $K_{\mu\nu}$ for each learning rule. The GD kernel is the usual initial NTK of Jacot et al. (2018). For $\rho$-aligned FA, each layer $\ell$'s contribution to the eNTK is suppressed by a factor $\rho^{L-\ell}$. For DFA and Hebb, only the last layer feature kernel $\Phi^L$ contributes to the NTK. For GLN, each layer has an identical contribution.

with base cases $\boldsymbol{\Phi}^0 = \boldsymbol{K}^x$ and $\boldsymbol{G}^{L+1} = \boldsymbol{1}\boldsymbol{1}^\top$. We provide interpretations of this result below.

- Backpropagation (GD) and $\rho = 1$ FA recover the usual depth $L$ NTK, with contributions from every layer $K_{\mu\nu} = \sum_\ell G_{\mu\nu}^{\ell+1}\Phi_{\mu\nu}^\ell$ at initialization. This kernel governs both training dynamics and test predictions in the lazy limit $\gamma_0 \to 0$ (Jacot et al., 2018; Lou et al., 2022).

- $\rho = 0$ FA, DFA and Hebb are equivalent to using the NNGP kernel $K_{\mu\nu} \sim \Phi_{\mu\nu}^L$, giving the Bayes posterior mean (Matthews et al., 2018; Lee et al., 2018; Hron et al., 2020). In the $\gamma_0, \rho \to 0$ limit, only the dynamics of the readout weights $\boldsymbol{w}^L$ contribute to the evolution of $f_\mu$ since error signals cannot successfully propagate backward and gradients cannot align with pseudo-gradients (App D). The standard $\rho = 0$ FA will be indistinguishable from merely training $\boldsymbol{w}^L$ with the delta-rule unless the network is trained in the rich feature learning regime $\gamma_0 > 0$, where $\tilde{G}^\ell$ can evolve. This effect was also noted in two layer networks by Song et al. (2021).

- $\rho$-FA weighs each layer $\ell$ with scale $\rho^{L-\ell}$, since each layer's pseudo-gradient is only partially correlated with the true gradient, giving recursion $\tilde{G}^\ell = \rho\tilde{G}^{\ell+1}$ with base case $\tilde{G}^{L+1} = G^{L+1}$.

- GLN's kernel in lazy limit is determined by the Gaussian gating variables $\{m_\mu^\ell\} \sim \mathcal{N}(0, \boldsymbol{K}^x)$.

We visualize these kernels for deep ReLU networks and ReLU GLNs for normalized inputs $|\boldsymbol{x}|^2 = |\boldsymbol{x}'|^2 = D$, by plotting the kernel as a function of the angle $\theta$ separating two inputs $\cos(\theta) =$

$\frac{1}{D} \boldsymbol{x}^\top \boldsymbol{x}'$. We find that the kernels develop a sharp discontinuity at the origin $\theta = 0$, which becomes more exaggerated as $\rho$ and $L$ increase. We further show that the square difference of width $N$ kenels and infinite width kernels go as $O(N^{-1})$. We derive this scaling with a perturbative argument in App. H, which enables analytical prediction of leading order finite size effects (Figure 7). In the lazy $\gamma_0 \to 0$ limit, these kernels define the eNTK and the network prediction dynamics.

### 3.2 FEATURE LEARNING ENABLES GRADIENT/PSEUDO-GRADIENT ALIGNMENT AND KERNEL/TASK ALIGNMENT

In the last section, we saw that, in the $\gamma_0 \to 0$ limit, all algorithms have frozen preactivations and pregradient features $\{h_\mu^\ell(t), z_\mu^\ell(t)\}$. A consequence of this fact is that FA and DFA cannot increase their gradient-pseudogradient alignment throughout training in the lazy limit $\gamma_0 = 0$. However, if we increase $\gamma_0$, then the gradient features $g_\mu^\ell(t)$ and pseudo-gradients $\tilde{g}_\mu^\ell(t)$ evolve in time and can increase their alignment. In Figure 3, we show the effect of increasing $\gamma_0$ on alignment dynamics in a depth 4 tanh network trained with DFA. In (b), we see that larger $\gamma_0$ is associated with high task-alignment of the last layer feature kernel $\Phi^L$, which becomes essentially rank one and aligned to $\boldsymbol{y}\boldsymbol{y}^\top$. The asymptotic cosine similarity between gradients and pseudogradients also increase with $\gamma_0$. The eNTK also becomes aligned with the task relevant directions (shown in Figure 3 c), like has been observed in GD training (Baratin et al., 2021; Shan & Bordelon, 2021; Geiger et al., 2021; Atanasov et al., 2022). We see that width $N$ networks have a dynamical eNTK $\boldsymbol{K}_N(t)$ which deviates from the DMFT eNTK $\boldsymbol{K}_\infty(t)$ by $O(1/N)$ in square loss. DMFT is more predictive for larger $\gamma_0$ networks, suggesting a reduction in finite size variability due to task-relevant feature evolution.

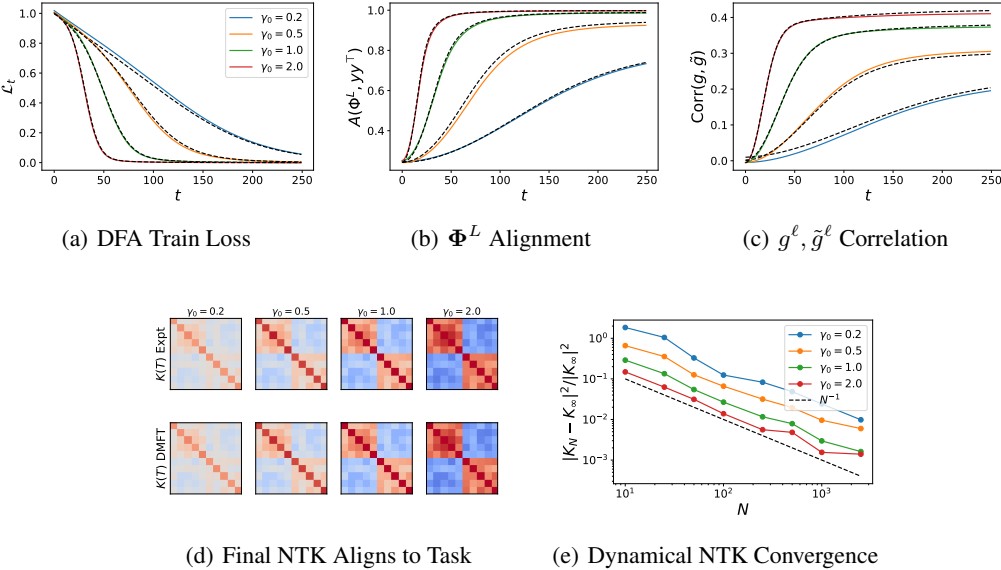

(a) DFA Train Loss  (b) $\boldsymbol{\Phi}^L$ Alignment  (c) $g^\ell, \tilde{g}^\ell$ Correlation

(d) Final NTK Aligns to Task  (e) Dynamical NTK Convergence

Figure 3: Feature Learning enables alignment for a depth 4 ($L = 3$ hidden layers) tanh network trained with direct feedback alignment (DFA) with varying $\gamma_0$. (a) Training loss for DFA networks with width $N = 4000$ with varying richness $\gamma_0$ shows that feature learning accelerates training, as predicted by DMFT (black). (b) The alignment (cosine similarity) of the last layer kernel $\boldsymbol{\Phi}^L$ with the target function reveals successful task depedent feature learning at large $\gamma_0$. (c) The dynamics of pseudo-grad./grad. correlation $\text{corr}(\boldsymbol{g}, \tilde{\boldsymbol{g}}) = \frac{1}{LP} \sum_{\ell,\mu} \frac{\boldsymbol{g}_\mu^\ell(t) \cdot \tilde{\boldsymbol{g}}_\mu^\ell(t)}{|\boldsymbol{g}_\mu^\ell(t)||\tilde{\boldsymbol{g}}_\mu^\ell(t)|}$ averaged over layers $\ell$ and datapoints $\mu$. Larger $\gamma_0$ generates more significant alignment between pseudogradients and gradients. (d) The final NTKs as a function of $\gamma_0$ reveals increasing clustering of the data points by class. (e) The error of the DMFT approximation for $K$'s dynamics as a function of $N$: $\frac{\langle |\boldsymbol{K}_N(t) - \boldsymbol{K}_\infty(t)|^2 \rangle_t}{\langle |\boldsymbol{K}_\infty(t)|^2 \rangle_t} \sim O(N^{-1})$, where the averages are computed over the time interval of training. This error is smaller for larger feature learning strength $\gamma_0$.

### 3.3 DEEP LINEAR NETWORK KERNEL DYNAMICS

When $\gamma_0 > 0$ the kernels and features in the network evolve according to the DMFT equations. For deep linear networks we can analyze the equations for the kernels in closed form without sampling since the correlation functions close algebraically (App. E). In Figure 4, we utilize our algebraic DMFT equations to explore $\rho$-FA dynamics in a depth 4 linear network. Networks with larger $\rho$ train faster, which can be intuited by noting that the initial function time derivative $\frac{df}{dt}\big|_{t=0} \sim \sum_{\ell=0}^{L} \rho^{L-\ell} \sim \frac{1-\rho^{L+1}}{1-\rho}$ is an increasing function of $\rho$. We observe higher final gradient pseudogradient alignment in each layer with larger $\rho$, which is also intuitive from the initial condition $\tilde{G}^\ell(0) = \rho^{L-\ell}$. However, surprisingly, for large initial correlation $\rho$, the NTK achieves lower task alignment, despite having larger $\tilde{G}^\ell(t)$. We show that this is caused by smaller overlap of each layer's feature kernel $\boldsymbol{H}^\ell(t)$ with $\boldsymbol{yy}^\top$. Though this phenomenon is counterintuitive, we gain more insight in the next section by studying an even simpler two layer model.

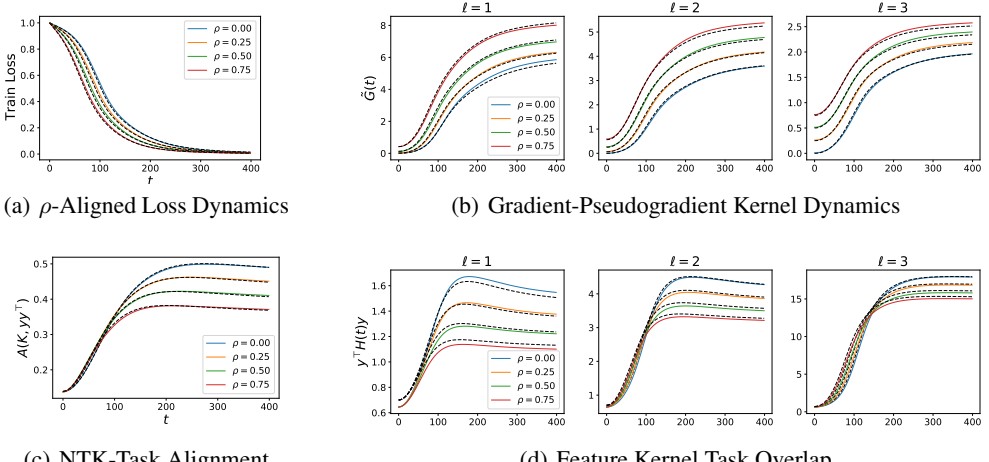

(a) $\rho$-Aligned Loss Dynamics  (b) Gradient-Pseudogradient Kernel Dynamics

(c) NTK-Task Alignment  (d) Feature Kernel Task Overlap

Figure 4: The initial feedback correlation $\rho$ alters alignment dynamics in on the FA dynamics in a depth 4 ($L = 3$ hidden layer) linear network. (a) Larger $\rho$ leads to faster initial training since the scale of the eNTK is larger. (b) Further, larger $\rho$ leads to larger scales of $\tilde{G}(t) = \frac{1}{N}\boldsymbol{g}^\ell(t) \cdot \tilde{\boldsymbol{g}}^\ell(t)$. (c) However, smaller $\rho$ leads to more alignment of the NTK $\boldsymbol{K}(t)$ with the task-relevant subspace, measured with cosine similarity $A(\boldsymbol{K}, \boldsymbol{yy}^\top)$. (d) The feature kernel $\boldsymbol{H}(t)$ overlaps with $\boldsymbol{y}$ reveal that $\boldsymbol{H}^\ell(t)$ aligns more significantly in the small $\rho$ networks.

### 3.3.1 EXACTLY SOLVEABLE DYNAMICS IN TWO LAYER LINEAR NETWORK

We can provide exact solutions to the infinite width GD and $\rho$-FA dynamics in the setting of Saxe et al. (2013), specifically a two layer linear network trained with whitened data $K^x_{\mu\nu} = \delta_{\mu\nu}$. Unlike Saxe et al. (2013)'s result, however, we do not demand small initialization scale (or equivalently large $\gamma_0$), but rather provide the exact solution for all positive $\gamma_0$. We will establish that large initial correlation $\rho$ results in higher gradient/pseudogradient alignment but lower alignment of the hidden feature kernel $\boldsymbol{H}(t)$ with the task relevant subspace $\boldsymbol{yy}^\top$.

We first note that when $\boldsymbol{K}^x = \boldsymbol{I}$, the GD or FA hidden feature kernel $\boldsymbol{H}(t)$ only evolves in the rank-one $\boldsymbol{yy}^\top$ subspace. It thus suffices to track the projection of $\boldsymbol{H}(t)$ on this rank one subspace, which we call $H_y(t)$. In the App. F we derive dynamics for $H_y$ for GD and $\rho$-FA

$$H_y(t) = \begin{cases} \tilde{G}(t) = \sqrt{1 + \gamma_0^2(y - \Delta(t))^2} \,, \ \frac{d\Delta}{dt} = -\sqrt{1 + \gamma_0^2(y - \Delta(t))^2}\Delta(t) & \text{GD} \\ 2\tilde{G}(t) + 1 - 2\rho = 1 + a^2 \,, \ \frac{da}{dt} = \gamma_0 y - \frac{1}{2}a^3 - (1 + \rho)a & \rho\text{-FA} \end{cases} \tag{7}$$

We illustrate these dynamics in Figure 5. The fixed points are $H_y = \sqrt{1 + \gamma_0^2 y^2}$ for GD and for $\rho$-FA, $H_y = 1 + a^2$ where $a$ is the smallest positive root of $\frac{1}{2}a^3 + (1 + \rho)a = \gamma_0 y$. For both GD and FA, we see that increasing $\gamma_0$ results in larger asymptotic values for $H_y$ and $\tilde{G}$. For $\rho$-FA the fixed point of $a$'s dynamics is a strictly decreasing function of $\rho$ since $\frac{da}{d\rho} < 0$, showing that the final

value of $H_y$ is smaller for larger $\rho$. On the contrary, we have that the final $\tilde{G} = \rho + \frac{1}{2}a^2$ is a strictly increasing function of $\rho$ since $\frac{d}{d\rho}\tilde{G} = 1 - \frac{a^2}{\frac{3}{2}a^2+(1+\rho)} > \frac{1}{3} > 0$. Thus, this simple model replicates the phenomenon of increasing $\tilde{G}$ and decreasing $H_y$ as $\rho$ increases. For the Hebb rule with $\boldsymbol{K}^x = \boldsymbol{I}$, the story is different. Instead of aligning $\boldsymbol{H}$ along the rank-one task relevant subspace, the dynamics instead decouple over samples, giving the following $P$ separate equations

$$\frac{d}{dt}\Delta_\mu = -[H_{\mu\mu}(t) + \gamma_0 \Delta_\mu(y_\mu - \Delta_\mu)]\Delta_\mu(t) \ , \ \frac{d}{dt}H_{\mu\mu} = 2\gamma_0\Delta_\mu(t)^2 H_{\mu\mu}. \tag{8}$$

From this perspective, we see that the hidden feature kernel does not align to the task, but rather increases its entries in overall scale as is illustrated in Figure 5 (b).

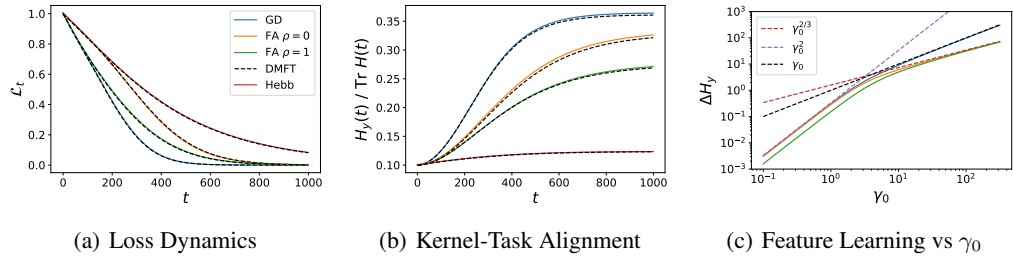

|  (a) Loss Dynamics | (b) Kernel-Task Alignment | (c) Feature Learning vs $\gamma_0$ |

Figure 5: The feature kernel dynamics and scaling with $\gamma_0^2$ for GD, $\rho$-FA, and Hebbian rules in an exactly solveable two layer linear network. (a) The loss dynamics for all algorithms reveals that $\rho = 0$ FA and Hebb rules have same early time dynamics and that $\rho = 1$ FA and GD have same early-time dynamics. However all loss curves become distinct at late times due to different eNTK dynamics. (b) The alignment of the kernel to the target function $H_y(t) = \frac{1}{|\boldsymbol{y}|^2}\boldsymbol{y}^\top \boldsymbol{H}\boldsymbol{y}/\text{Tr}\boldsymbol{H}(t)$ increases significantly for GD, and FA, but not for Hebb, reflecting the task-independence of the learned representation. (c) The movement of the feature kernel $\Delta H_y = \lim_{t\to\infty} H_y(t) - H_y(0)$ as a function of $\gamma_0$ for GD, and $\rho = 0, 1$ FA. At small feature learning strength, all algorithms have updates on the order of $\Delta H_y \sim \gamma_0^2$. At large $\gamma_0$, GD has $\Delta H_y \sim \gamma_0$ while FA has $\Delta H_y \sim \gamma_0^{2/3}$. The $\rho = 1$ FA (green) has lower $\Delta H_y$ than the $\rho = 0$ FA across all $\gamma_0$.

## 4 DISCUSSION

We provided an analysis of the training dynamics of a wide range of learning rules at infinite width. This set of rules includes (but is not limited to) GD, $\rho$-FA, DFA, GLN and Hebb as well as many others. We showed that each of these learning rules has an dynamical effective NTK which concentrates over initializations at infinite width. In the lazy $\gamma_0 \to 0$ regime, it suffices to compute the the initial NTK, while in the rich regime, we provide a dynamical mean field theory to compute the NTK's dynamics. We showed that, in the rich regime, FA learning rules do indeed align the network's gradient vectors to their pseudo-gradients and that this alignment improves with $\gamma_0$. We show that initial correlation $\rho$ between forward and backward pass weights alters the inductive bias of FA in both lazy and rich regimes. In the rich regime, larger $\rho$ networks have smaller eNTK evolution. Overall, our study is a step towards understanding learned representations in neural networks, and the quest to reverse-engineer learning rules from observations of evolving neural representations during learning in the brain.

Many open problems remain unresolved with the present work. We currently have only implemented our theory in MLPs. An implementation in CNNs could explain some of the observed advantages of partial initial alignment in $\rho$-FA (Xiao et al., 2018; Moskovitz et al., 2018; Bartunov et al., 2018; Refinetti et al., 2021). In addition, our framework is sufficiently flexible to propose and test new learning rules by providing new $\tilde{g}_\mu^\ell(t)$ formulas. Our DMFT gives a recipe to compute their initial kernels, function dynamics and analyze their learned representations. The generalization performance of these learning rules at varying $\gamma_0$ is yet to be explored. Lastly, our DMFT is numerically expensive for large datasets and training intervals, making it difficult to scale up to realistic datsets. Future work could provide theoretical convergence guarantees for our DMFT solver.

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

## APPENDIX

## A    ALGORITHM TO SOLVE NONLINEAR DMFT EQUATIONS

---

**Algorithm 1:** Alternating Monte-Carlo Solution to Saddle Point Equations

---

**Data:** $\boldsymbol{K}^x, \boldsymbol{y}$, Initial Guesses $\{\boldsymbol{\Phi}^\ell, \boldsymbol{G}^\ell, \tilde{\boldsymbol{G}}^\ell, \tilde{\tilde{\boldsymbol{G}}}^\ell\}_{\ell=1}^L$, $\{\boldsymbol{A}^\ell, \boldsymbol{B}^\ell, \boldsymbol{C}^\ell, \boldsymbol{D}^\ell\}_{\ell=1}^{L-1}$, Sample count $\mathcal{S}$, Update Speed $\beta$

**Result:** Network predictions through training $f_\mu(t)$, correlation functions
$\{\boldsymbol{\Phi}^\ell, \boldsymbol{G}^\ell, \tilde{\boldsymbol{G}}^\ell, \tilde{\tilde{\boldsymbol{G}}}^\ell\}_{\ell=1}^L$, response functions $\{\boldsymbol{A}^\ell, \boldsymbol{B}^\ell, \boldsymbol{C}^\ell, \boldsymbol{D}^\ell\}_{\ell=1}^{L-1}$,

1   $\boldsymbol{\Phi}^0 = \boldsymbol{K}^x \otimes \mathbf{1}\mathbf{1}^\top$, $\boldsymbol{G}^{L+1} = \mathbf{1}\mathbf{1}^\top$ ;

2   **while** *Kernels Not Converged* **do**

3     From $\{\boldsymbol{\Phi}^\ell, \boldsymbol{G}^\ell\}$ compute $\boldsymbol{K}^{NTK}(t,t)$ and solve $\frac{d}{dt} f_\mu(t) = \sum_\alpha \Delta_\alpha(t) K^{NTK}_{\mu\alpha}(t,t)$;

4     $\ell = 1$;

5     **while** $\ell < L+1$ **do**

6       Draw $\mathcal{S}$ samples $\{u^\ell_{\mu,n}(t)\}_{n=1}^{\mathcal{S}} \sim \mathcal{GP}(0, \boldsymbol{\Phi}^{\ell-1})$,
$\{r^\ell_{\mu,n}(t), v^\ell_{\mu,n}(t)\}_{n=1}^{\mathcal{S}} \sim \mathcal{GP}\left(0, \begin{bmatrix} \boldsymbol{G}^{\ell+1} & \tilde{\boldsymbol{G}}^{\ell+1} \\ \tilde{\boldsymbol{G}}^{\ell+1\top} & \tilde{\tilde{\boldsymbol{G}}}^{\ell+1} \end{bmatrix}\right)$;

7       Solve equation 5 for each sample to get $\{h^\ell_{\mu,n}(t), z^\ell_{\mu,n}(t), \tilde{g}^\ell_{\mu,n}(t)\}_{n=1}^{\mathcal{S}}$;

8       Use learning rule (Table 1) to compute $\{\tilde{g}^\ell_{\mu,n}(t)\}_{n=1}^{\mathcal{S}}$;

9       Compute new correlation function $\{\boldsymbol{\Phi}^\ell, \boldsymbol{G}^\ell, \tilde{\boldsymbol{G}}^\ell, \tilde{\tilde{\boldsymbol{G}}}^\ell\}$ estimates:

10       $\Phi^{\ell,new}_{\mu\nu}(t,s) = \frac{1}{\mathcal{S}} \sum_{n\in[\mathcal{S}]} \phi(h^\ell_{\mu,n}(t)) \phi(h^\ell_{\nu,n}(s))$,

11       $G^{\ell,new}_{\mu\nu}(t,s) = \frac{1}{\mathcal{S}} \sum_{n\in[\mathcal{S}]} g^\ell_{\mu,n}(t) g^\ell_{\nu,n}(s)$,

12       $\tilde{G}^{\ell,new}_{\mu\nu}(t,s) = \frac{1}{\mathcal{S}} \sum_{n\in[\mathcal{S}]} g^\ell_{\mu,n}(t) \tilde{g}^\ell_{\nu,n}(s)$,

13       $\tilde{\tilde{G}}^{\ell,new}_{\mu\nu}(t,s) = \frac{1}{\mathcal{S}} \sum_{n\in[\mathcal{S}]} \tilde{g}^\ell_{\mu,n}(t) \tilde{g}^\ell_{\nu,n}(s)$ ;

14       Solve for Jacobians on each sample $\frac{\partial \phi(\boldsymbol{h}^\ell_n)}{\partial \boldsymbol{r}^{\ell\top}_n}, \frac{\partial \phi(\boldsymbol{h}^\ell_n)}{\partial \boldsymbol{v}^{\ell\top}_n}, \frac{\partial \boldsymbol{g}^\ell_n}{\partial \boldsymbol{u}^{\ell\top}_n}, \frac{\partial \tilde{\boldsymbol{g}}^\ell_n}{\partial \boldsymbol{u}^{\ell\top}_n}$ ;

15       Compute new response functions $\{\boldsymbol{A}^\ell, \boldsymbol{B}^{\ell-1}, \boldsymbol{C}^\ell, \boldsymbol{D}^{\ell-1}\}$ estimates:

16       $\boldsymbol{A}^{\ell,new} = \frac{1}{\mathcal{S}} \sum_{n\in[\mathcal{S}]} \frac{\partial \phi(\boldsymbol{h}^\ell_n)}{\partial \boldsymbol{r}^{\ell\top}_n}$ , $\boldsymbol{B}^{\ell-1,new} = \frac{1}{\mathcal{S}} \sum_{n\in[\mathcal{S}]} \frac{\partial \boldsymbol{g}^\ell_n}{\partial \boldsymbol{u}^{\ell\top}_n}$ ;

17       $\boldsymbol{C}^{\ell,new} = \frac{1}{\mathcal{S}} \sum_{n\in[\mathcal{S}]} \frac{\partial \phi(\boldsymbol{h}^\ell_n)}{\partial \boldsymbol{v}^\ell_n}$ , $\boldsymbol{D}^{\ell-1,new} = \frac{1}{\mathcal{S}} \sum_{n\in[\mathcal{S}]} \frac{\partial \tilde{\boldsymbol{g}}^\ell_n}{\partial \boldsymbol{u}^{\ell\top}_n}$ ;

18       $\ell \leftarrow \ell + 1$;

19     **end**

20     $\ell = 1$;

21     **while** $\ell < L+1$ **do**

22       Update correlation functions

23       $\boldsymbol{\Phi}^\ell \leftarrow (1-\beta)\boldsymbol{\Phi}^\ell + \beta\boldsymbol{\Phi}^{\ell,new}$, $\boldsymbol{G}^\ell \leftarrow (1-\beta)\boldsymbol{G}^\ell + \beta\boldsymbol{G}^{\ell,new}$ ;

24       $\tilde{\boldsymbol{G}}^\ell \leftarrow (1-\beta)\tilde{\boldsymbol{G}}^\ell + \beta\tilde{\boldsymbol{G}}^{\ell,new}$, $\tilde{\tilde{\boldsymbol{G}}}^\ell \leftarrow (1-\beta)\tilde{\tilde{\boldsymbol{G}}}^\ell + \beta\tilde{\tilde{\boldsymbol{G}}}^{\ell,new}$ ;

25       **if** $\ell < L$ **then**

26         Update response functions

27         $\boldsymbol{A}^\ell \leftarrow (1-\beta)\boldsymbol{A}^\ell + \beta\boldsymbol{A}^{\ell,new}$, $\boldsymbol{B}^\ell \leftarrow (1-\beta)\boldsymbol{B}^\ell + \beta\boldsymbol{B}^{\ell,new}$

28         $\boldsymbol{C}^\ell \leftarrow (1-\beta)\boldsymbol{C}^\ell + \beta\boldsymbol{C}^{\ell,new}$, $\boldsymbol{D}^\ell \leftarrow (1-\beta)\boldsymbol{D}^\ell + \beta\boldsymbol{D}^{\ell,new}$

29       **end**

30       $\ell \leftarrow \ell + 1$

31     **end**

32 **end**

33 **return** $\{f_\mu(t)\}_{\mu=1}^P, \{\boldsymbol{\Phi}^\ell, \boldsymbol{G}^\ell, \tilde{\boldsymbol{G}}^\ell, \tilde{\tilde{\boldsymbol{G}}}^\ell\}_{\ell=1}^L, \{\boldsymbol{A}^\ell, \boldsymbol{B}^\ell, \boldsymbol{C}^\ell, \boldsymbol{D}^\ell\}_{\ell=1}^{L-1}$

---

The sample-and-solve procedure we developed and describe below for nonlinear networks is based on numerical recipes used in the dynamical mean field simulations in computational physics Manacorda et al. (2020) and is similar to recent work in the GD case Bordelon & Pehlevan (2022). The basic principle is to leverage the fact that, conditional on order parameters, we can easily draw

samples $\{u_\mu^\ell(t), r_\mu^\ell(t), \zeta_\mu^\ell(t), \tilde{\zeta}_\mu^\ell(t)\}$ from their appropriate GPs. From these sampled fields, we can identify the kernel order parameters by simple estimation of the appropriate moments. The algorithm is provided in Algorithm 1.

The parameter $\beta$ controls recency weighting of the samples obtained at each iteration. If $\beta = 1$, then the rank of the kernel estimates is limited to the number of samples $\mathcal{S}$ used in a single iteration, but with $\beta < 1$ smaller sample sizes $\mathcal{S}$ can be used to still obtain accurate results. We used $\beta = 0.6$ in our deep network experiments.

## B  DERIVATION OF DMFT EQUATIONS

In this section, we derive the DMFT description of infinite network dynamics. The path integral theory we develop is based on the Martin-Siggia-Rose-De Dominicis-Janssen (MSRDJ) framework Martin et al. (1973). A useful review of this technique applied to random recurrent networks can be found here Crisanti & Sompolinsky (2018). This framework was recently extended for deep learning with GD in (Bordelon & Pehlevan, 2022).

### B.1  WRITING EVOLUTION EQUATIONS IN FEATURE SPACE

First, we will express all of the learning dynamics in terms of preactivation features $\boldsymbol{h}_\mu^\ell(t) = \frac{1}{\sqrt{N}} \boldsymbol{W}^\ell(t)\phi(\boldsymbol{h}_\mu^\ell(t))$, pre-gradient features $\boldsymbol{z}_\mu^\ell(t) = \frac{1}{\sqrt{N}} \boldsymbol{W}^\ell(t)^\top \boldsymbol{g}^{\ell+1}$ and pseudogradient features $\tilde{\boldsymbol{g}}_\mu^\ell(t)$. Since we would like to understand typical behavior over random initializations of weights $\boldsymbol{\theta}(0) = \{\boldsymbol{W}^0(0), \boldsymbol{W}^1(0), ..., \boldsymbol{w}^L(0)\}$, we want to isolate the dependence of our evolution equations by $\boldsymbol{W}^\ell(0)$. We achieve this separation by using our learning dynamics for $\boldsymbol{W}^\ell(t)$

$$\boldsymbol{W}^\ell(t) = \boldsymbol{W}^\ell(0) + \frac{\gamma_0}{\sqrt{N}} \int_0^t ds \sum_{\mu=1}^P \Delta_\mu(s)\tilde{\boldsymbol{g}}_\mu^{\ell+1}(s)\phi(\boldsymbol{h}_\mu^\ell(s))^\top. \tag{9}$$

The inclusion of the prefactor $\frac{\gamma_0}{\sqrt{N}}$ in the weight dynamics ensures that $\frac{d}{dt}f = O_{\gamma_0, N}(1)$ and $\frac{d}{dt}h^\ell = O_{\gamma_0, N}(\gamma_0)$ at initialization (Chizat et al., 2019; Bordelon & Pehlevan, 2022). Using the forward and backward pass equations, we find the following evolution equations for our feature vectors

$$\boldsymbol{h}_\mu^\ell(t) = \boldsymbol{\chi}_\mu^\ell(t) + \gamma_0 \int_0^t ds \sum_{\nu=1}^P \Delta_\nu(s)\tilde{\boldsymbol{g}}_\mu^{\ell+1}(s)\Phi_{\mu\nu}^{\ell-1}(t,s) \, , \; \boldsymbol{\chi}_\mu^\ell(t) = \frac{1}{\sqrt{N}} \boldsymbol{W}^\ell(0)\phi(\boldsymbol{h}_\mu^\ell(t))$$

$$\boldsymbol{z}_\mu^\ell(t) = \boldsymbol{\xi}_\mu^\ell(t) + \gamma_0 \int_0^t ds \sum_{\nu=1}^P \Delta_\nu(s)\phi(\boldsymbol{h}_\mu^\ell(s))\tilde{G}_{\mu\nu}^{\ell+1}(t,s) \, , \; \boldsymbol{\xi}_\mu^\ell(t) = \frac{1}{\sqrt{N}} \boldsymbol{W}^\ell(0)^\top \boldsymbol{g}_\mu^{\ell+1}(t) \, , \tag{10}$$

where we introduced the following feature and gradient/pseudo-gradient kernels

$$\Phi_{\mu\nu}^\ell(t,s) = \frac{1}{N}\phi(\boldsymbol{h}_\mu^\ell(t)) \cdot \phi(\boldsymbol{h}_\nu^\ell(s)) \, , \; \tilde{G}_{\mu\nu}^\ell(t,s) = \frac{1}{N}\boldsymbol{g}_\mu^\ell(t) \cdot \tilde{\boldsymbol{g}}_\nu^\ell(s). \tag{11}$$

The particular learning rule defines the definition of the pseudo-gradient $\tilde{\boldsymbol{g}}_\mu^\ell(t)$. We note that, for all learning rules considered, the pseudogradient $\tilde{g}_{i,\mu}^\ell(t)$ is a function of the fields $\{h_{i,\mu}^\ell(t), z_{i\mu}^\ell(t), m_{i\mu}^\ell(t), \zeta_{i,\mu}^\ell(t), \tilde{\zeta}_{i,\mu}^\ell(t)\}_{\mu \in [P], t \in \mathbb{R}_+}$, conditional on the value of the kernels $\{\Phi^\ell, \tilde{G}^\ell\}$. The additional fields have definitions

$$\boldsymbol{\zeta}_\mu^\ell(t) = \frac{1}{\sqrt{N}} \boldsymbol{W}^\ell(0)^\top \tilde{\boldsymbol{g}}_\mu^{\ell+1}(t) \, , \; \tilde{\boldsymbol{\zeta}}_\mu^{\ell+1}(t) = \frac{1}{\sqrt{N}} \tilde{\boldsymbol{W}}^{\ell\top} \tilde{\boldsymbol{g}}_\mu^{\ell+1}(t) \tag{12}$$

and are specifically required for $\rho$-FA with $\rho > 0$ since $\tilde{\boldsymbol{g}}_\mu^\ell(t) = \rho\dot{\phi}(\boldsymbol{h}_\mu^\ell(t)) \odot \boldsymbol{\zeta}_\mu^\ell(t) + \sqrt{1-\rho^2}\dot{\phi}(\boldsymbol{h}_\mu^\ell(t)) \odot \tilde{\boldsymbol{\zeta}}_\mu^\ell(t)$. The fields $\boldsymbol{m}_\mu^\ell = \frac{1}{\sqrt{D}} \boldsymbol{M}^\ell \boldsymbol{x}_\mu$ are required for GLNs.

All of the necessary fields $\{\boldsymbol{h}_\mu^\ell(t), \boldsymbol{z}_\mu^\ell(t), \tilde{\boldsymbol{g}}_\mu^\ell(t)\}$ are thus causal functions of the stochastic fields $\{\boldsymbol{\chi}_\mu^\ell(t), \boldsymbol{\xi}_\mu^\ell(t), \boldsymbol{m}_\mu^\ell, \boldsymbol{\zeta}_\mu^\ell(t), \tilde{\boldsymbol{\zeta}}_\mu^\ell(t)\}$ and the kernels $\{\Phi^\ell, \tilde{G}^\ell\}$. It thus suffices to characterize the distribution of these latter objects over random initialization of $\boldsymbol{\theta}(0)$ in the $N \to \infty$ limit, which we study in the next section.

## B.2 MOMENT GENERATING FUNCTIONAL

We will now attempt to characterize the probability density of the random fields

$$\boldsymbol{\chi}_\mu^{\ell+1}(t) = \frac{1}{\sqrt{N}} \boldsymbol{W}^\ell(0) \phi(\boldsymbol{h}_\mu^\ell(t)) \,, \; \boldsymbol{\xi}_\mu^\ell(t) = \frac{1}{\sqrt{N}} \boldsymbol{W}^\ell(0)^\top \boldsymbol{g}_\mu^{\ell+1}(t) \,, \; \boldsymbol{m}_\mu^\ell = \frac{1}{\sqrt{D}} \boldsymbol{M}^\ell \boldsymbol{x}_\mu$$

$$\boldsymbol{\zeta}_\mu^\ell(t) = \frac{1}{\sqrt{N}} \boldsymbol{W}^\ell(0)^\top \tilde{\boldsymbol{g}}_\mu^{\ell+1}(t) \,, \; \tilde{\boldsymbol{\zeta}}_\mu^\ell(t) = \frac{1}{\sqrt{N}} \tilde{\boldsymbol{W}}^{\ell\top} \tilde{\boldsymbol{g}}_\mu^{\ell+1}(t). \tag{13}$$

It is readily apparent that the fields $\boldsymbol{m}_\mu^\ell$ are independent of the others and have a Gaussian distribution over random Gaussian $\boldsymbol{M}^\ell$. These fields, therefore do not can be handled independently from the others, which are statistically coupled through the initial conditions. We will thus characterize the moment generating functional of the remaining fields $\{\boldsymbol{\chi}_\mu^\ell(t), \boldsymbol{\xi}_\mu^\ell(t), \boldsymbol{\zeta}_\mu^\ell(t), \tilde{\boldsymbol{\zeta}}_\mu^\ell(t)\}$ over random initial condition and random backward pass weights

$$Z[\{\boldsymbol{j}_\mu^\ell(t), \boldsymbol{k}_\mu^\ell(t), \boldsymbol{n}_\mu^\ell(t), \boldsymbol{p}_\mu^\ell(t)\}]$$

$$= \mathbb{E}_{\boldsymbol{\theta}(0), \{\tilde{\boldsymbol{W}}^\ell\}} \exp \left( \sum_{\mu=1}^P \int_0^\infty dt \left[ \boldsymbol{j}_\mu^\ell(t) \cdot \boldsymbol{\chi}_\mu^\ell(t) + \boldsymbol{k}_\mu^\ell(t) \cdot \boldsymbol{\xi}_\mu^\ell(t) + \boldsymbol{n}_\mu^\ell(t) \cdot \boldsymbol{\zeta}_\mu^\ell(t) + \boldsymbol{p}_\mu^\ell(t) \cdot \tilde{\boldsymbol{\zeta}}_\mu^\ell(t) \right] \right)$$

$$\tag{14}$$

where $\boldsymbol{\chi}^\ell, \boldsymbol{\xi}, \boldsymbol{\zeta}, \tilde{\boldsymbol{\zeta}}$ are regarded as functions of $\boldsymbol{\theta}(0), \{\tilde{\boldsymbol{W}}^\ell\}$. Arbitrary moments of these random variables can be computed by differentiation of $Z$ near zero source. For example, a two-point correlation function can be obtained as

$$\left\langle \chi_{i,\mu}^\ell(t) \zeta_{i',\nu}^{\ell'}(s) \right\rangle = \lim_{\boldsymbol{j}, \boldsymbol{k}, \boldsymbol{n}, \boldsymbol{p} \to 0} \frac{\delta}{\delta j_{i,\mu}^\ell(t)} \frac{\delta}{\delta n_{i'\nu}^{\ell'}(s)} Z[\{\boldsymbol{j}_\mu^\ell(t), \boldsymbol{k}_\mu^\ell(t), \boldsymbol{n}_\mu^\ell(t), \boldsymbol{p}_\mu^\ell(t)\}]. \tag{15}$$

More generally, we let $\boldsymbol{\mu} = (i, \mu, t)$ be a tuple containing the neuron, time, and sample index for an entry of one of these fields so that $\chi_{\boldsymbol{\mu}}^\ell = \chi_{i,\mu}^\ell(t)$. Further, we let $\mathcal{N}_{\chi^\ell}, \mathcal{N}_{\xi^\ell}, \mathcal{N}_{\zeta^\ell}, \mathcal{N}_{\tilde{\zeta}^\ell}$ be index sets which contain sample and time indices as well as neuron indices $\mathcal{N}_\chi = \{\boldsymbol{\mu}_1^\chi, ..., \boldsymbol{\mu}_{|\mathcal{N}_\chi|}^\chi\}$ for all of the indices we wish to compute an average over. Then arbitrary moments can be computed with the formula

$$\left\langle \prod_\ell \left[ \prod_{\boldsymbol{\mu} \in \mathcal{N}_{\chi^\ell}} \chi_{\boldsymbol{\mu}}^\ell \prod_{\boldsymbol{\nu} \in \mathcal{N}_{\xi^\ell}} \xi_{\boldsymbol{\nu}}^\ell \prod_{\boldsymbol{\alpha} \in \mathcal{N}_{\zeta^\ell}} \zeta_{\boldsymbol{\alpha}}^\ell \prod_{\boldsymbol{\beta} \in \mathcal{N}_{\tilde{\xi}^\ell}} \tilde{\zeta}_{\boldsymbol{\beta}}^\ell \right] \right\rangle$$

$$= \lim_{\boldsymbol{j}, \boldsymbol{k}, \boldsymbol{n}, \boldsymbol{p} \to 0} \prod_\ell \left[ \prod_{\boldsymbol{\mu} \in \mathcal{N}_{\chi^\ell}} \frac{\delta}{\delta j_{\boldsymbol{\mu}}^\ell} \prod_{\boldsymbol{\nu} \in \mathcal{N}_{\xi^\ell}} \frac{\delta}{\delta k_{\boldsymbol{\nu}}^\ell} \prod_{\boldsymbol{\alpha} \in \mathcal{N}_{\zeta^\ell}} \frac{\delta}{\delta n_{\boldsymbol{\alpha}}^\ell} \prod_{\boldsymbol{\beta} \in \mathcal{N}_{\tilde{\xi}^\ell}} \frac{\delta}{\delta p_{\boldsymbol{\mu}}^\ell} \right] Z[\{\boldsymbol{j}_\mu^\ell(t), \boldsymbol{k}_\mu^\ell(t), \boldsymbol{n}_\mu^\ell(t), \boldsymbol{p}_\mu^\ell(t)\}]. \tag{16}$$

We now to study this moment generating functional $Z$ in the large width $N \to \infty$ limit.

## B.3 PATH INTEGRAL FORMULATION AND INTEGRATION OVER WEIGHTS

To enable the average over the weights, we multiply $Z$ by an integral representation of unity that enforces the relationship between $\boldsymbol{\chi}_\mu^{\ell+1}(t), \boldsymbol{W}^\ell(0), \phi(\boldsymbol{h}_\mu^\ell(t))$

$$1 = \int_{\mathbb{R}^N} d\boldsymbol{\chi}_\mu^{\ell+1}(t) \, \delta \left( \boldsymbol{\chi}_\mu^{\ell+1}(t) - \frac{1}{\sqrt{N}} \boldsymbol{W}^\ell(0) \phi(\boldsymbol{h}_\mu^\ell(t)) \right)$$

$$= \int_{\mathbb{R}^N} \int_{\mathbb{R}^N} \frac{d\boldsymbol{\chi}_\mu^{\ell+1}(t) d\hat{\boldsymbol{\chi}}_\mu^{\ell+1}(t)}{(2\pi)^N} \exp \left( i \hat{\boldsymbol{\chi}}_\mu^{\ell+1}(t) \cdot \left[ \boldsymbol{\chi}_\mu^{\ell+1}(t) - \frac{1}{\sqrt{N}} \boldsymbol{W}^\ell(0) \phi(\boldsymbol{h}_\mu^\ell(t)) \right] \right) \,. \tag{17}$$

In the second line, we used the Fourier representation of the Dirac-Delta function for each of the $N$ neuron indices $\delta(r) = \int_{-\infty}^\infty \frac{d\hat{r}}{2\pi} \exp(i\hat{r}r)$. We repeat this procedure for the other fields $\boldsymbol{\xi}_\mu^\ell(t), \boldsymbol{\zeta}_\mu^\ell(t), \tilde{\boldsymbol{\zeta}}_\mu^\ell(t)$ at each time $t$ and each sample $\mu$. After inserting these delta functions, we find

the following form of the moment generating functional

$$
Z = \int \prod_{\ell\mu t} \frac{d\boldsymbol{\chi}_\mu^\ell(t)d\hat{\boldsymbol{\chi}}_\mu^\ell(t)}{(2\pi)^N} \frac{d\boldsymbol{\xi}_\mu^\ell(t)d\hat{\boldsymbol{\xi}}_\mu^\ell(t)}{(2\pi)^N} \frac{d\boldsymbol{\zeta}_\mu^\ell(t)d\hat{\boldsymbol{\zeta}}_\mu^\ell(t)}{(2\pi)^N} \frac{d\tilde{\boldsymbol{\zeta}}_\mu^\ell(t)d\hat{\tilde{\boldsymbol{\zeta}}}_\mu^\ell(t)}{(2\pi)^N}
$$

$$
\times \exp\left( \int_0^\infty dt \sum_{\ell,\mu} \left[ \boldsymbol{\chi}_\mu^\ell(t)\cdot(\boldsymbol{j}_\mu^\ell(t)+i\hat{\boldsymbol{\chi}}_\mu^\ell(t)) + \boldsymbol{\xi}_\mu^\ell(t)\cdot(\boldsymbol{k}_\mu^\ell(t)+i\hat{\boldsymbol{\xi}}_\mu^\ell(t)) \right] \right)
$$

$$
\times \exp\left( \int_0^\infty dt \sum_{\ell,\mu} \left[ \boldsymbol{\zeta}_\mu^\ell(t)\cdot(\boldsymbol{n}_\mu^\ell(t)+i\hat{\boldsymbol{\zeta}}_\mu^\ell(t)) + \tilde{\boldsymbol{\zeta}}_\mu^\ell(t)\cdot(\boldsymbol{p}_\mu^\ell(t)+i\hat{\tilde{\boldsymbol{\zeta}}}_\mu^\ell(t)) \right] \right)
$$

$$
\times \prod_\ell \mathbb{E}_{\boldsymbol{W}^\ell(0)} \exp\left( -\frac{i}{\sqrt{N}} \mathrm{Tr}\, \boldsymbol{W}^\ell(0)^\top \left[ \int dt \sum_\mu \hat{\boldsymbol{\chi}}_\mu^{\ell+1}(t)\phi(\boldsymbol{h}_\mu^\ell(t))^\top + \boldsymbol{g}_\mu^{\ell+1}(t)\hat{\boldsymbol{\xi}}_\mu^\ell(t)^\top \right] \right)
$$

$$
\times \exp\left( -\frac{i}{\sqrt{N}} \boldsymbol{W}^\ell(0)^\top \left[ \int dt \sum_\mu \tilde{\boldsymbol{g}}_\mu^{\ell+1}(t)\boldsymbol{\zeta}_\mu^\ell(t)^\top \right] \right)
$$

$$
\times \prod_\ell \mathbb{E}_{\tilde{\boldsymbol{W}}^\ell} \exp\left( -\frac{i}{\sqrt{N}} \mathrm{Tr}\tilde{\boldsymbol{W}}^{\ell\top} \left[ \int dt \sum_\mu \tilde{\boldsymbol{g}}_\mu^{\ell+1}(t)\hat{\tilde{\boldsymbol{\zeta}}}_\mu^\ell(t)^\top \right] \right) . \tag{18}
$$

We see that we often have simultaneous integrals over time $t$ and sums over samples $\mu$ so we will again adopt a shorthand notation for indices $\boldsymbol{\mu} = (\mu, t)$ and define a summmation convention $\sum_{\boldsymbol{\mu}} a_{\boldsymbol{\mu}} b_{\boldsymbol{\mu}} = \int_0^\infty dt \sum_{\mu=1}^P a_\mu(t)b_\mu(t)$. To perform the averages over weights, we note that for a standard normal variable $W_{ij}$, that $\mathbb{E}_{W_{ij}} \exp\left(iW_{ij}a_ib_j\right) = \exp\left(-\frac{1}{2}a_i^2 b_i^2\right)$. Using this fact for each of the weight matrix averages, we have

$$
\mathbb{E}_{\boldsymbol{W}^\ell(0)} \exp\left( -\frac{i}{\sqrt{N}} \mathrm{Tr}\, \boldsymbol{W}^\ell(0)^\top \left[ \sum_{\boldsymbol{\mu}} \hat{\boldsymbol{\chi}}_{\boldsymbol{\mu}}^{\ell+1}\phi(\boldsymbol{h}_{\boldsymbol{\mu}}^\ell)^\top + \boldsymbol{g}_{\boldsymbol{\mu}}^{\ell+1}\hat{\boldsymbol{\xi}}_{\boldsymbol{\mu}}^{\ell\top} + \tilde{\boldsymbol{g}}_{\boldsymbol{\mu}}^{\ell+1}\hat{\boldsymbol{\zeta}}_{\boldsymbol{\mu}}^{\ell\top} \right] \right)
$$

$$
= \exp\left( -\frac{1}{2N} \left| \sum_{\boldsymbol{\mu}} \hat{\boldsymbol{\chi}}_{\boldsymbol{\mu}}^{\ell+1}\phi(\boldsymbol{h}_{\boldsymbol{\mu}}^\ell)^\top + \boldsymbol{g}_{\boldsymbol{\mu}}^{\ell+1}\hat{\boldsymbol{\xi}}_{\boldsymbol{\mu}}^{\ell\top} + \tilde{\boldsymbol{g}}_{\boldsymbol{\mu}}^{\ell+1}\hat{\boldsymbol{\zeta}}_{\boldsymbol{\mu}}^{\ell\top} \right|_F^2 \right)
$$

$$
= \exp\left( -\frac{1}{2} \sum_{\boldsymbol{\mu},\boldsymbol{\nu}} \left[ \hat{\boldsymbol{\chi}}_{\boldsymbol{\mu}}^{\ell+1}\cdot\hat{\boldsymbol{\chi}}_{\boldsymbol{\nu}}^{\ell+1}\Phi_{\boldsymbol{\mu},\boldsymbol{\nu}}^\ell + \hat{\boldsymbol{\xi}}_{\boldsymbol{\mu}}^\ell\cdot\hat{\boldsymbol{\xi}}_{\boldsymbol{\nu}}^\ell G_{\boldsymbol{\mu}\boldsymbol{\nu}}^{\ell+1} + \hat{\boldsymbol{\zeta}}_{\boldsymbol{\mu}}^\ell\cdot\hat{\boldsymbol{\zeta}}_{\boldsymbol{\nu}}^\ell\tilde{\tilde{G}}_{\boldsymbol{\mu},\boldsymbol{\nu}}^{\ell+1} + \hat{\boldsymbol{\xi}}_{\boldsymbol{\mu}}^\ell\cdot\hat{\boldsymbol{\zeta}}_{\boldsymbol{\nu}}^\ell\tilde{G}_{\boldsymbol{\mu},\boldsymbol{\nu}}^{\ell+1} \right] \right)
$$

$$
\times \exp\left( -i\sum_{\boldsymbol{\mu}\boldsymbol{\nu}} \left[ \hat{\boldsymbol{\chi}}_{\boldsymbol{\mu}}^{\ell+1}\cdot\boldsymbol{g}_{\boldsymbol{\nu}}^{\ell+1}A_{\boldsymbol{\mu}\boldsymbol{\nu}}^\ell + \hat{\boldsymbol{\chi}}_{\boldsymbol{\mu}}^{\ell+1}\cdot\tilde{\boldsymbol{g}}_{\boldsymbol{\nu}}^{\ell+1}C_{\boldsymbol{\mu}\boldsymbol{\nu}}^\ell \right] \right) . \tag{19}
$$

In the above, we introduced a collection of order parameters $\{\Phi, G, \tilde{G}, \tilde{\tilde{G}}, A, C\}$, which will correspond to correlation and response functions of our DMFT. These are defined as

$$
\Phi_{\boldsymbol{\mu},\boldsymbol{\nu}}^\ell = \frac{1}{N}\phi(\boldsymbol{h}_{\boldsymbol{\mu}}^\ell)\cdot\phi(\boldsymbol{h}_{\boldsymbol{\nu}}^\ell) , \; G_{\boldsymbol{\mu},\boldsymbol{\nu}}^\ell = \frac{1}{N}\boldsymbol{g}_{\boldsymbol{\mu}}^\ell\cdot\boldsymbol{g}_{\boldsymbol{\nu}}^\ell , \; \tilde{G}_{\boldsymbol{\mu}\boldsymbol{\nu}}^\ell = \frac{1}{N}\boldsymbol{g}_{\boldsymbol{\mu}}^\ell\cdot\tilde{\boldsymbol{g}}_{\boldsymbol{\nu}}^\ell
$$

$$
\tilde{\tilde{G}}_{\boldsymbol{\mu},\boldsymbol{\nu}}^{\ell+1} = \frac{1}{N}\tilde{\boldsymbol{g}}_{\boldsymbol{\mu}}^\ell\cdot\tilde{\boldsymbol{g}}_{\boldsymbol{\nu}}^\ell , \; iA_{\boldsymbol{\mu}\boldsymbol{\nu}}^\ell = \frac{1}{N}\phi(\boldsymbol{h}_{\boldsymbol{\mu}}^\ell)\cdot\hat{\boldsymbol{\xi}}_{\boldsymbol{\nu}}^\ell , \; iC_{\boldsymbol{\mu}\boldsymbol{\nu}}^\ell = \frac{1}{N}\phi(\boldsymbol{h}_{\boldsymbol{\mu}}^\ell)\cdot\hat{\boldsymbol{\zeta}}_{\boldsymbol{\nu}}^\ell . \tag{20}
$$

We perform a similar average over $\tilde{W}^\ell$ can be obtained directly

$$
\mathbb{E}_{\tilde{\boldsymbol{W}}^\ell} \exp\left( -\frac{i}{\sqrt{N}} \mathrm{Tr}\tilde{\boldsymbol{W}}^{\ell\top} \left[ \sum_{\boldsymbol{\mu}} \tilde{\boldsymbol{g}}_{\boldsymbol{\mu}}^{\ell+1}\hat{\tilde{\boldsymbol{\zeta}}}_{\boldsymbol{\mu}}^{\ell\top} \right] \right) = \exp\left( -\frac{1}{2} \sum_{\boldsymbol{\mu}\boldsymbol{\nu}} \hat{\tilde{\boldsymbol{\zeta}}}_{\boldsymbol{\mu}}^\ell\cdot\hat{\tilde{\boldsymbol{\zeta}}}_{\boldsymbol{\nu}}^\ell\tilde{G}_{\boldsymbol{\mu}\boldsymbol{\nu}}^{\ell+1} \right) . \tag{21}
$$

Now that we have defined our collection of order parameters, we enforce their definitions with Dirac-Delta functions by multiplying by one. For example,

$$
1 = N\int d\Phi_{\boldsymbol{\mu}\boldsymbol{\nu}}^\ell \delta\left( N\Phi_{\boldsymbol{\mu}\boldsymbol{\nu}}^\ell - \phi(\boldsymbol{h}_{\boldsymbol{\mu}}^\ell)\cdot\phi(\boldsymbol{h}_{\boldsymbol{\nu}}^\ell) \right)
$$

$$
= \int \frac{d\Phi_{\boldsymbol{\mu}\boldsymbol{\nu}}^\ell d\hat{\Phi}_{\boldsymbol{\mu}\boldsymbol{\nu}}^\ell}{2\pi N^{-1}} \exp\left( N\hat{\Phi}_{\boldsymbol{\mu}\boldsymbol{\nu}}^\ell\Phi_{\boldsymbol{\mu}\boldsymbol{\nu}}^\ell - \hat{\Phi}_{\boldsymbol{\mu}\boldsymbol{\nu}}^\ell\phi(\boldsymbol{h}_{\boldsymbol{\mu}}^\ell)\cdot\phi(\boldsymbol{h}_{\boldsymbol{\nu}}^\ell) \right) . \tag{22}
$$

We enforce these definitions for all order parameters $\{\Phi^\ell_{\mu\nu}, G^\ell_{\mu\nu}, \tilde{G}^\ell_{\mu\nu}, \tilde{\tilde{G}}^\ell_{\mu\nu}, A^\ell_{\mu,\nu}, C^\ell_{\mu,\nu}\}$. We let the corresponding Fourier duals for each of these order parameters be $\{\hat{\Phi}^\ell_{\mu\nu}, \hat{G}^\ell_{\mu\nu}, \hat{\tilde{G}}^\ell_{\mu\nu}, \hat{\tilde{\tilde{G}}}^\ell_{\mu\nu}, -B^\ell_{\mu,\nu}, -D^\ell_{\mu,\nu}\}$. In the next section we show the resulting formula for the moment generating function and take the $N \to \infty$ limit to derive our DMFT equations.

## B.4 DMFT Action

After inserting the Dirac-Delta functions to enforce the definitions of the order parameters, we derive the following moment generating functional in terms of $\boldsymbol{q} = \{\Phi, \hat{\Phi}, G, \hat{G}, \tilde{G}, \hat{\tilde{G}}, \tilde{\tilde{G}}, \hat{\tilde{\tilde{G}}}, A, B, C, D, j, k, n, p\}$

$$Z = \int \prod_{\ell,\boldsymbol{\mu},\boldsymbol{\nu}} \frac{d\Phi^\ell_{\mu\nu} d\hat{\Phi}^\ell_{\mu\nu}}{2\pi N^{-1}} \frac{dG^\ell_{\mu\nu} d\hat{G}^\ell_{\mu\nu}}{2\pi N^{-1}} \frac{d\tilde{G}^\ell_{\mu\nu} d\hat{\tilde{G}}^\ell_{\mu\nu}}{2\pi N^{-1}} \frac{d\tilde{\tilde{G}}^\ell_{\mu\nu} d\hat{\tilde{\tilde{G}}}^\ell_{\mu\nu}}{2\pi N^{-1}} \frac{dA^\ell_{\mu\nu} dB^\ell_{\mu\nu}}{2\pi N^{-1}} \frac{dC^\ell_{\mu\nu} dD^\ell_{\mu\nu}}{2\pi N^{-1}} \exp\left(NS[\boldsymbol{q}]\right)$$

where $S[\boldsymbol{q}]$ is the $O_N(1)$ DMFT action which takes the form

$$S[\boldsymbol{q}] = \sum_{\ell\boldsymbol{\mu}\boldsymbol{\nu}} \left[ \Phi^\ell_{\mu,\nu} \hat{\Phi}^\ell_{\mu,\nu} + G^\ell_{\mu\nu} \hat{G}^\ell_{\mu\nu} + \tilde{G}^\ell_{\mu\nu} \hat{\tilde{G}}^\ell_{\mu\nu} + \tilde{\tilde{G}}^\ell_{\mu\nu} \hat{\tilde{\tilde{G}}}^\ell_{\mu\nu} - A^\ell_{\mu\nu} B^\ell_{\mu\nu} - C^\ell_{\mu\nu} D^\ell_{\mu\nu} \right]$$
$$+ \frac{1}{N} \sum_{i=1}^N \sum_{\ell=1}^L \ln \mathcal{Z}^\ell_i[\boldsymbol{q}]. \tag{23}$$

The single-site moment generating functionals (MGF) $\mathcal{Z}^\ell_i$ involve only the integrals with sources $\{j^\ell_i, k^\ell_i, n^\ell_i, p^\ell_i\}$ for neuron $i \in [N]$ in layer $\ell$. For a given set of order parameters $\boldsymbol{q}$ at zero source, these functionals become identical across all neuron sites $i$. Concretely, for any $\ell \in [L], i \in [N]$, the single site MGF takes the form

$$\mathcal{Z}^\ell_i = \int \prod_{\boldsymbol{\mu}} \frac{d\chi^\ell_{\boldsymbol{\mu}} d\hat{\chi}^\ell_{\boldsymbol{\mu}}}{2\pi} \frac{d\xi^\ell_{\boldsymbol{\mu}} d\hat{\xi}^\ell_{\boldsymbol{\mu}}}{2\pi} \frac{d\zeta^\ell_{\boldsymbol{\mu}} d\hat{\zeta}^\ell_{\boldsymbol{\mu}}}{2\pi} \frac{d\tilde{\zeta}^\ell_{\boldsymbol{\mu}} d\hat{\tilde{\zeta}}^\ell_{\boldsymbol{\mu}}}{2\pi} \tag{24}$$

$$\exp\left( -\frac{1}{2} \sum_{\boldsymbol{\mu},\boldsymbol{\nu}} \left[ \hat{\chi}^{\ell+1}_{\boldsymbol{\mu}} \hat{\chi}^{\ell+1}_{\boldsymbol{\nu}} \Phi^\ell_{\mu,\nu} + \hat{\xi}^\ell_{\boldsymbol{\mu}} \hat{\xi}^\ell_{\boldsymbol{\nu}} G^{\ell+1}_{\mu\nu} + \hat{\tilde{\zeta}}^\ell_{\boldsymbol{\mu}} \hat{\tilde{\zeta}}^\ell_{\boldsymbol{\nu}} \tilde{\tilde{G}}^{\ell+1}_{\mu\nu} \right] \right)$$

$$\exp\left( -\frac{1}{2} \sum_{\boldsymbol{\mu}\boldsymbol{\nu}} \left[ \hat{\zeta}^\ell_{\boldsymbol{\mu}} \hat{\zeta}^\ell_{\boldsymbol{\nu}} \tilde{\tilde{G}}^{\ell+1}_{\mu,\nu} + 2\hat{\xi}^\ell_{\boldsymbol{\mu}} \hat{\zeta}^\ell_{\boldsymbol{\nu}} \tilde{G}^{\ell+1}_{\mu,\nu} \right] \right)$$

$$\exp\left( -\sum_{\boldsymbol{\mu}\boldsymbol{\nu}} \left[ \hat{\Phi}^\ell_{\mu\nu} \phi(h^\ell_{\boldsymbol{\mu}}) \phi(h^\ell_{\boldsymbol{\nu}}) + \hat{G}^\ell_{\mu\nu} g^\ell_{\boldsymbol{\mu}} g^\ell_{\boldsymbol{\nu}} + \hat{\tilde{G}}^\ell_{\mu\nu} g^\ell_{\boldsymbol{\mu}} \tilde{g}^\ell_{\boldsymbol{\nu}} + \hat{\tilde{\tilde{G}}}^\ell_{\mu\nu} \tilde{g}^\ell_{\boldsymbol{\mu}} \tilde{g}^\ell_{\boldsymbol{\nu}} \right] \right)$$

$$\exp\left( -i \sum_{\boldsymbol{\mu}\boldsymbol{\nu}} \left[ \hat{\chi}^{\ell+1}_{\boldsymbol{\mu}} g^{\ell+1}_{\boldsymbol{\nu}} A^\ell_{\mu\nu} + \hat{\chi}^\ell_{\boldsymbol{\mu}} \tilde{g}^{\ell+1}_{\boldsymbol{\nu}} C^\ell_{\mu\nu} + \phi(h^\ell_{\boldsymbol{\mu}}) \hat{\xi}^\ell_{\boldsymbol{\nu}} B^\ell_{\mu\nu} + \phi(h^\ell_{\boldsymbol{\mu}}) \hat{\zeta}^\ell_{\boldsymbol{\nu}} D^\ell_{\mu\nu} \right] \right)$$

$$\exp\left( \sum_{\boldsymbol{\mu}} \left[ \chi^\ell_{\boldsymbol{\mu}} (j^\ell_{i,\boldsymbol{\mu}} + i\hat{\chi}^\ell_{\boldsymbol{\mu}}) + \xi^\ell_{\boldsymbol{\mu}} (k^\ell_{i,\boldsymbol{\mu}} + i\hat{\xi}^\ell_{\boldsymbol{\mu}}) + \zeta^\ell_{\boldsymbol{\mu}} (n^\ell_{i,\boldsymbol{\mu}} + i\hat{\zeta}^\ell_{\boldsymbol{\mu}}) + \tilde{\zeta}^\ell_{\boldsymbol{\mu}} (p^\ell_{i,\boldsymbol{\mu}} + i\hat{\tilde{\zeta}}^\ell_{\boldsymbol{\mu}}) \right] \right).$$

As promised, the only terms in $\mathcal{Z}_i$ which vary over site index $i$ are the sources $\{j, k, n, p\}$. To simplify our later saddle point equations, we will abstract the notation for the single site MGF, letting

$$\mathcal{Z}^\ell_i = \int \prod_{\boldsymbol{\mu}} \frac{d\chi_{\boldsymbol{\mu}} d\hat{\chi}_{\boldsymbol{\mu}}}{2\pi} \frac{d\xi_{\boldsymbol{\mu}} d\hat{\xi}_{\boldsymbol{\mu}}}{2\pi} \frac{d\zeta_{\boldsymbol{\mu}} d\hat{\zeta}_{\boldsymbol{\mu}}}{2\pi} \frac{d\tilde{\zeta}_{\boldsymbol{\mu}} d\hat{\tilde{\zeta}}_{\boldsymbol{\mu}}}{2\pi} \exp\left( -\mathcal{H}^\ell_i[\chi, \hat{\chi}, \xi, \hat{\xi}, \zeta, \hat{\zeta}, \tilde{\zeta}, \hat{\tilde{\zeta}}] \right) \tag{25}$$

where $\mathcal{H}^\ell_i$ is the single site effective Hamiltonian for neuron $i$ and layer $\ell$. Note that at zero source, $\mathcal{H}^\ell_i$ are identical for all $i \in [N]$.

### B.5 SADDLE POINT EQUATIONS

Letting the full collection of concatenated order parameters $\boldsymbol{q}$ be indexed by $b$. We now take the $N \to \infty$ limit, using the method of steepest descent

$$Z = \int \prod_b \frac{\sqrt{N} dq_b}{\sqrt{2\pi}} \exp\left(NS[\boldsymbol{q}]\right) \sim \exp\left(NS[\boldsymbol{q}^*]\right), \ \nabla S[\boldsymbol{q}]|_{\boldsymbol{q}^*} = 0, \ N \to \infty. \tag{26}$$

We see that the integral over $\boldsymbol{q}$ is exponentially dominated by the saddle point where $\nabla S[\boldsymbol{q}] = 0$. We thus need to solve these saddle point equations for the $\boldsymbol{q}^*$. To do this, we need to introduce some notation. Let $O(\chi, \hat{\chi}, \xi, \hat{\xi}, \zeta, \hat{\zeta}, \tilde{\zeta}, \hat{\tilde{\zeta}})$ be an arbitrary function of the single site stochastic processes. We define the $\ell$-th layer $i$-th single site average, denoted by $\left\langle O(\chi, \hat{\chi}, \xi, \hat{\xi}, \zeta, \hat{\zeta}, \tilde{\zeta}, \hat{\tilde{\zeta}}) \right\rangle_{\ell,i}$ as

$$\left\langle O(\chi, \hat{\chi}, \xi, \hat{\xi}, \zeta, \hat{\zeta}, \tilde{\zeta}, \hat{\tilde{\zeta}}) \right\rangle_{\ell,i} = \frac{1}{\mathcal{Z}_i^\ell} \int \prod_{\boldsymbol{\mu}} \frac{d\chi_{\boldsymbol{\mu}} d\hat{\chi}_{\boldsymbol{\mu}}}{2\pi} \frac{d\xi_{\boldsymbol{\mu}} d\hat{\xi}_{\boldsymbol{\mu}}}{2\pi} \frac{d\zeta_{\boldsymbol{\mu}} d\hat{\zeta}_{\boldsymbol{\mu}}}{2\pi} \frac{d\tilde{\zeta}_{\boldsymbol{\mu}} d\hat{\tilde{\zeta}}_{\boldsymbol{\mu}}}{2\pi}$$
$$\exp\left(-\mathcal{H}_i^\ell[\chi, \hat{\chi}, \xi, \hat{\xi}, \zeta, \hat{\zeta}, \tilde{\zeta}, \hat{\tilde{\zeta}}]\right) O(\chi, \hat{\chi}, \xi, \hat{\xi}, \zeta, \hat{\zeta}, \tilde{\zeta}, \hat{\tilde{\zeta}}) \tag{27}$$

which can be interpreted as an average over the Gibbs measure defined by energy $\mathcal{H}_i^\ell$. With this notation, we now set about computing the saddle point equations which define the primal order parameters $\{\Phi, G, \tilde{G}, \tilde{\tilde{G}}\}$.

$$\frac{\partial S}{\partial \hat{\Phi}_{\boldsymbol{\mu\nu}}^\ell} = \Phi_{\boldsymbol{\mu\nu}}^\ell - \frac{1}{N} \sum_{i=1}^N \left\langle \phi(h_{\boldsymbol{\mu}}^\ell)\phi(h_{\boldsymbol{\nu}}^\ell) \right\rangle_{\ell,i} = 0$$

$$\frac{\partial S}{\partial \hat{G}_{\boldsymbol{\mu\nu}}^\ell} = G_{\boldsymbol{\mu\nu}}^\ell - \frac{1}{N} \sum_{i=1}^N \left\langle g_{\boldsymbol{\mu}}^\ell g_{\boldsymbol{\nu}}^\ell \right\rangle_{\ell,i} = 0$$

$$\frac{\partial S}{\partial \hat{\tilde{G}}_{\boldsymbol{\mu\nu}}^\ell} = \tilde{G}_{\boldsymbol{\mu\nu}}^\ell - \frac{1}{N} \sum_{i=1}^N \left\langle g_{\boldsymbol{\mu}}^\ell \tilde{g}_{\boldsymbol{\nu}}^\ell \right\rangle_{\ell,i} = 0$$

$$\frac{\partial S}{\partial \hat{\tilde{\tilde{G}}}_{\boldsymbol{\mu\nu}}^\ell} = \tilde{\tilde{G}}_{\boldsymbol{\mu\nu}}^\ell - \frac{1}{N} \sum_{i=1}^N \left\langle \tilde{g}_{\boldsymbol{\mu}}^\ell \tilde{g}_{\boldsymbol{\nu}}^\ell \right\rangle_{\ell,i} = 0$$

We further compute the saddle point equations for the dual order parameters

$$\frac{\partial S}{\partial \Phi^\ell_{\mu\nu}} = \hat{\Phi}^\ell_{\mu\nu} - \frac{1}{2N} \sum_{i=1}^N \left\langle \hat{\chi}^{\ell+1}_{\mu} \hat{\chi}^{\ell+1}_{\nu} \right\rangle_{\ell+1,i} = 0$$

$$\frac{\partial S}{\partial G^\ell_{\mu\nu}} = \hat{G}^\ell_{\mu\nu} - \frac{1}{2N} \sum_{i=1}^N \left\langle \hat{\xi}^{\ell-1}_{\mu} \hat{\xi}^{\ell-1}_{\nu} \right\rangle_{\ell-1,i} = 0$$

$$\frac{\partial S}{\partial \tilde{G}^\ell_{\mu\nu}} = \hat{\tilde{G}}^\ell_{\mu\nu} - \frac{1}{N} \sum_{i=1}^N \left\langle \hat{\xi}^{\ell-1}_{\mu} \hat{\zeta}^{\ell-1}_{\nu} \right\rangle_{\ell-1,i} = 0$$

$$\frac{\partial S}{\partial \tilde{\tilde{G}}^\ell_{\mu\nu}} = \hat{\tilde{\tilde{G}}}^\ell_{\mu\nu} - \frac{1}{2N} \sum_{i=1}^N \left\langle [\hat{\zeta}^{\ell-1}_{\mu} \hat{\zeta}^{\ell-1}_{\nu} + \hat{\tilde{\zeta}}^{\ell-1}_{\mu} \hat{\tilde{\zeta}}^{\ell-1}_{\nu}] \right\rangle_{\ell-1,i} = 0$$

$$\frac{\partial S}{\partial A^\ell_{\mu\nu}} = -B^\ell_{\mu\nu} - \frac{i}{N} \sum_{i=1}^N \left\langle \hat{\chi}^{\ell+1}_{\mu} g^{\ell+1}_{\nu} \right\rangle_{\ell+1,i} = 0$$

$$\frac{\partial S}{\partial B^\ell_{\mu\nu}} = -A^\ell_{\mu\nu} - \frac{i}{N} \sum_{i=1}^N \left\langle \phi(h^\ell_{\mu}) \hat{\xi}^\ell_{\nu} \right\rangle_{\ell,i} = 0$$

$$\frac{\partial S}{\partial C^\ell_{\mu\nu}} = -D^\ell_{\mu\nu} - \frac{i}{N} \sum_{i=1}^N \left\langle \hat{\chi}^{\ell+1}_{\mu} \tilde{g}^{\ell+1}_{\nu} \right\rangle_{\ell+1,i} = 0$$

$$\frac{\partial S}{\partial D^\ell_{\mu\nu}} = -C^\ell_{\mu\nu} - \frac{i}{N} \sum_{i=1}^N \left\langle \phi(h^\ell_{\mu}) \hat{\zeta}^\ell_{\nu} \right\rangle_{\ell,i} = 0 \,. \tag{28}$$

The correlation functions involving real variables $\{h, g, \tilde{g}\}$ have a straightforward interpetation. However, it is not immediately clear what to do with terms involving the dual fields $\{\hat{\chi}, \hat{\xi}, \hat{\zeta}\}$. As a starting example, let's consider one of the terms for $B^{\ell-1}_{\mu\nu}$, namely $-i \left\langle \hat{\chi}^\ell_{\nu} g^\ell_{\nu} \right\rangle$. We make progress by inserting another fictitious source term $u^\ell_{\mu}$ and differentiating near zero source

$$-i \left\langle \hat{\chi}^\ell_{\nu} g^\ell_{\nu} \right\rangle_i = \lim_{\{u_\mu\} \to 0} \frac{\partial}{\partial u^\ell_{\nu}} \left\langle g^\ell_{\nu} \exp\left( -i \sum_{\nu'} u_{\nu'} \hat{\chi}^\ell_{\mu'} \right) \right\rangle_i . \tag{29}$$

Introducing a vectorization notation $\boldsymbol{u}^\ell = \mathrm{Vec}\{u^\ell_{\mu}\}_{\mu}$, $\hat{\boldsymbol{\chi}}^\ell = \mathrm{Vec}\{\hat{\chi}^\ell_{\mu}\}_{\mu}$ and $\boldsymbol{\Phi}^{\ell-1} = \mathrm{Mat}\{\Phi^{\ell-1}_{\mu,\nu}\}_{\mu,\nu}$, we can perform the internal integrals over $\hat{\boldsymbol{\chi}}^\ell$

$$\int \prod_{\boldsymbol{\mu}} \frac{d\hat{\chi}^\ell_{\boldsymbol{\mu}}}{\sqrt{2\pi}} \exp\left( -\frac{1}{2} \hat{\boldsymbol{\chi}}^{\ell\top} \boldsymbol{\Phi}^{\ell-1} \hat{\boldsymbol{\chi}}^\ell + i\hat{\boldsymbol{\chi}}^\ell \cdot (\boldsymbol{\chi}^\ell - \boldsymbol{u}^\ell - \boldsymbol{A}^{\ell-1} \boldsymbol{g}^\ell - \boldsymbol{C}^{\ell-1} \tilde{\boldsymbol{g}}^\ell) \right)$$

$$= \exp\left( -\frac{1}{2} (\boldsymbol{\chi}^\ell - \boldsymbol{u}^\ell - \boldsymbol{A}^{\ell-1} \boldsymbol{g}^\ell - \boldsymbol{C}^{\ell-1} \tilde{\boldsymbol{g}}^\ell)[\boldsymbol{\Phi}^{\ell-1}]^{-1} (\boldsymbol{\chi}^\ell - \boldsymbol{u}^\ell - \boldsymbol{A}^{\ell-1} \boldsymbol{g}^\ell - \boldsymbol{C}^{\ell-1} \tilde{\boldsymbol{g}}^\ell) \right)$$

$$\exp\left( -\frac{1}{2} \ln \det \boldsymbol{\Phi}^{\ell-1} \right) . \tag{30}$$

We thus need to compute a derivative of the above function with respect to $\boldsymbol{u}^\ell$ at $\boldsymbol{u}^\ell = 0$, which gives

$$-i \left\langle \hat{\boldsymbol{\chi}}^\ell \boldsymbol{g}^{\ell\top} \right\rangle_i = [\boldsymbol{\Phi}^{\ell-1}]^{-1} \left\langle (\boldsymbol{\chi}^\ell - \boldsymbol{A}^{\ell-1} \boldsymbol{g}^\ell - \boldsymbol{C}^{\ell-1} \tilde{\boldsymbol{g}}^\ell) \boldsymbol{g}^{\ell\top} \right\rangle_i . \tag{31}$$

From the above reasoning, we can also easily obtain $\hat{\Phi}^{\ell-1}$ using

$$
\begin{aligned}
\left\langle \hat{\chi}^\ell \hat{\chi}^{\ell\top} \right\rangle &= -\frac{\partial^2}{\partial u^\ell \partial u^{\ell\top}}|_{u=0} \left\langle \exp\left(-iu^\ell \cdot \hat{\chi}^\ell\right) \right\rangle \\
&= -\int d\chi^\ell ... \frac{\partial^2}{\partial u^\ell \partial u^\top}|_{u=0} \\
&\times \exp\left(-\frac{1}{2}(\chi^\ell - u^\ell - A^{\ell-1}g^\ell - C^{\ell-1}\tilde{g}^\ell)[\Phi^{\ell-1}]^{-1}(\chi^\ell - u^\ell - A^{\ell-1}g^\ell - C^{\ell-1}\tilde{g}^\ell) - ...\right) \\
&= [\Phi^{\ell-1}]^{-1} - [\Phi^{\ell-1}]^{-1} \left\langle (\chi^\ell - A^{\ell-1}g^\ell - C^{\ell-1}\tilde{g}^\ell)(\chi^\ell - A^{\ell-1}g^\ell - C^{\ell-1}\tilde{g}^\ell)^\top \right\rangle [\Phi^{\ell-1}]^{-1}.
\end{aligned}
\tag{32}
$$

Performing a similar analysis, we insert source fields $r_+ = \begin{bmatrix} r^\ell \\ v^\ell \end{bmatrix}$ for $\hat{\xi}^\ell_+ = \begin{bmatrix} \hat{\xi}^\ell \\ \hat{\zeta}^\ell \end{bmatrix}$ and define $G^{\ell+1}_+ = \begin{bmatrix} G^{\ell+1} & \tilde{G}^{\ell+1} \\ \tilde{G}^{\ell+1\top} & \tilde{\tilde{G}}^{\ell+1} \end{bmatrix}$, and $B^\ell_+ = \begin{bmatrix} B^\ell & D^\ell \end{bmatrix}$ and then we can compute the necessary averages using the same technique

$$
-i\left\langle \phi(h^\ell)\hat{\xi}^{\ell\top}_+ \right\rangle = \frac{\partial}{\partial r^\ell_+}|_{r^\ell_+=0} \left\langle \phi(h^\ell) \exp\left(-ir^\ell_+ \cdot \hat{\xi}^\ell_+\right) \right\rangle = \left\langle \phi(h^\ell)(\xi^\ell_+ - B^{\ell\top}_+ \phi(h^\ell))^\top \right\rangle
$$

$$
\begin{aligned}
\left\langle \hat{\xi}^\ell_+ \hat{\xi}^{\ell\top}_+ \right\rangle &= -\frac{\partial^2}{\partial r^\ell_+ \partial r^{\ell\top}_+}|_{r^\ell_+=0} \left\langle \exp\left(-ir^\ell_+ \cdot \hat{\xi}^\ell_+\right) \right\rangle \\
&= [G^{\ell+1}_+]^{-1} - [G^{\ell+1}_+]^{-1} \left\langle (\xi^\ell_+ - B^{\ell\top}_+ \phi(h^\ell))(\xi^\ell_+ - B^{\ell\top}_+ \phi(h^\ell))^\top \right\rangle [G^{\ell+1}_+]^{-1}.
\end{aligned}
\tag{33}
$$

We now have formulas for all the necessary averages entirely in terms of the primal fields $\{\chi, \xi, \zeta, \tilde{\zeta}\}$.

## B.6 Linearizing with the Hubbard Trick

Now, using the fact that in the $N \to \infty$ limit $q$ concentrates around $q^*$, we will simplify our single site stochastic processes so we can obtain a final formula for $\{A, B, C, D, \hat{\Phi}, \hat{G}, \hat{\tilde{G}}, \hat{\tilde{\tilde{G}}}\}$. To do so, we utilize the Hubbard-Stratonovich identity

$$
\exp\left(-\frac{\sigma^2}{2}k^2\right) = \int \frac{du}{\sqrt{2\pi\sigma^2}} \exp\left(-\frac{1}{2\sigma^2}u^2 - iku\right),
\tag{34}
$$

which is merely a consequence of the Fourier transform of the Gaussian distribution. This is often referred to as "linearizing" the action since the a term quadratic in $k$ was replaced with an average of an action which is linear in $k$. In our setting, we perform this trick on a collection of variables which appear in the quadratic forms of our single site MGFs $\mathcal{Z}^\ell_i$. For example, for the $\hat{\chi}^{\ell+1}$ fields, we have

$$
\exp\left(-\frac{1}{2}\sum_{\mu\nu}\hat{\chi}^{\ell+1}_\mu \hat{\chi}^{\ell+1}_\nu \Phi^\ell_{\mu\nu}\right) = \left\langle \exp\left(-i\sum_\mu \hat{\chi}^{\ell+1}_\mu u^{\ell+1}_\mu\right) \right\rangle_{\{u^{\ell+1}_\mu\} \sim \mathcal{N}(0, \Phi^\ell)}.
\tag{35}
$$

Similarly, we perform a joint decomposition for the $\{\hat{\xi}^\ell, \hat{\zeta}^\ell\}$ fields which gives

$$
\exp\left(-\frac{1}{2}\sum_{\mu\nu}\left[\hat{\xi}^\ell_\mu \hat{\xi}^\ell_\nu G^{\ell+1}_{\mu,\nu} + 2\hat{\xi}^\ell_\mu \hat{\zeta}^\ell_\nu \tilde{G}^{\ell+1}_{\mu\nu} + \hat{\zeta}^\ell_\mu \hat{\zeta}^\ell_\nu \tilde{\tilde{G}}^{\ell+1}_{\mu\nu}\right]\right)
\tag{36}
$$

$$
= \left\langle \exp\left(-i\sum_\mu [r^\ell_\mu \hat{\xi}^\ell_\mu + v^\ell_\mu \hat{\zeta}^\ell_\mu]\right) \right\rangle_{\{r^\ell_\mu, v^\ell_\mu\} \sim \mathcal{N}(0, G^{\ell+1}_+)}, \quad G^{\ell+1}_+ = \begin{bmatrix} G^{\ell+1} & \tilde{G}^{\ell+1} \\ \tilde{G}^{\ell+1} & \tilde{\tilde{G}}^{\ell+1} \end{bmatrix}.
$$

We thus see that the Gaussian sources $\{r^\ell_\mu\}_\mu$ and $\{v^\ell_\mu\}_\mu$ are mean zero with correlation given by $\Sigma^{\ell+1}$. Now that we have linearized the quadratic components involving each of the dual fields

$\{\hat{\chi}, \hat{\xi}, \hat{\zeta}\}$, we now perform integration over these variables, giving

$$
\int \prod_{\boldsymbol{\mu}} \frac{d\hat{\chi}_{\boldsymbol{\mu}}^{\ell}}{2\pi} \exp\left(i \sum_{\boldsymbol{\mu}} \hat{\chi}_{\boldsymbol{\mu}}^{\ell} \left(\chi_{\boldsymbol{\mu}}^{\ell} - u_{\boldsymbol{\mu}}^{\ell} - \sum_{\nu} A_{\boldsymbol{\mu}\nu}^{\ell-1} g_{\nu}^{\ell} - \sum_{\nu} C_{\boldsymbol{\mu}\nu}^{\ell-1} \tilde{g}_{\nu}^{\ell}\right)\right)
$$

$$
= \prod_{\boldsymbol{\mu}} \delta\left(\chi_{\boldsymbol{\mu}}^{\ell} - u_{\boldsymbol{\mu}}^{\ell} - \sum_{\nu} A_{\boldsymbol{\mu}\nu}^{\ell-1} g_{\nu}^{\ell} - \sum_{\nu} C_{\boldsymbol{\mu}\nu}^{\ell-1} \tilde{g}_{\nu}^{\ell}\right)
$$

$$
\int \prod_{\boldsymbol{\mu}} \frac{d\hat{\xi}_{\boldsymbol{\mu}}^{\ell}}{2\pi} \exp\left(i \sum_{\boldsymbol{\mu}} \hat{\xi}_{\boldsymbol{\mu}}^{\ell} \left(\xi_{\boldsymbol{\mu}}^{\ell} - r_{\boldsymbol{\mu}}^{\ell} - \sum_{\nu} B_{\nu\boldsymbol{\mu}}^{\ell} \phi(h_{\nu}^{\ell})\right)\right) = \prod_{\boldsymbol{\mu}} \delta\left(\xi_{\boldsymbol{\mu}}^{\ell} - r_{\boldsymbol{\mu}}^{\ell} - \sum_{\nu} B_{\nu\boldsymbol{\mu}}^{\ell} \phi(h_{\nu}^{\ell})\right)
$$

$$
\int \prod_{\boldsymbol{\mu}} \frac{d\hat{\zeta}_{\boldsymbol{\mu}}^{\ell}}{2\pi} \exp\left(i \sum_{\boldsymbol{\mu}} \hat{\zeta}_{\boldsymbol{\mu}}^{\ell} \left(\zeta_{\boldsymbol{\mu}}^{\ell} - v_{\boldsymbol{\mu}}^{\ell} - \sum_{\nu} D_{\nu\boldsymbol{\mu}}^{\ell} \phi(h_{\nu}^{\ell})\right)\right) = \prod_{\boldsymbol{\mu}} \delta\left(\zeta_{\boldsymbol{\mu}}^{\ell} - v_{\boldsymbol{\mu}}^{\ell} - \sum_{\nu} D_{\nu\boldsymbol{\mu}}^{\ell} \phi(h_{\nu}^{\ell})\right).
$$

$$(37)$$

This reveals the following set of identities

$$
\chi_{\boldsymbol{\mu}}^{\ell} = u_{\boldsymbol{\mu}}^{\ell} + \sum_{\nu} A_{\boldsymbol{\mu}\nu}^{\ell-1} g_{\nu}^{\ell} + \sum_{\nu} C_{\boldsymbol{\mu}\nu}^{\ell-1} \tilde{g}_{\nu}^{\ell}
$$

$$
\xi_{\boldsymbol{\mu}}^{\ell} = r_{\boldsymbol{\mu}}^{\ell} + \sum_{\nu} B_{\nu\boldsymbol{\mu}}^{\ell} \phi(h_{\nu}^{\ell}) \,, \; \zeta_{\boldsymbol{\mu}}^{\ell} = v_{\boldsymbol{\mu}}^{\ell} + \sum_{\nu} D_{\nu\boldsymbol{\mu}}^{\ell-1} \phi(h_{\nu}^{\ell}).
$$

$$(38)$$

Since we know by construction that $\boldsymbol{u}^{\ell} = \boldsymbol{\chi}^{\ell} - \boldsymbol{A}^{\ell-1} \boldsymbol{g}^{\ell} - \boldsymbol{C}^{\ell-1} \tilde{\boldsymbol{g}}^{\ell}$ is a zero mean Gaussian with covariance $\boldsymbol{\Phi}^{\ell-1}$, we can simplify our expressions for $\boldsymbol{B}^{\ell-1}$ and $\hat{\boldsymbol{\Phi}}^{\ell-1}$ using Stein's Lemma

$$
\boldsymbol{B}^{\ell-1} = \frac{1}{N} \sum_{i=1}^{N} [\boldsymbol{\Phi}^{\ell-1}]^{-1} \left\langle \boldsymbol{u}^{\ell} \boldsymbol{g}^{\ell\top} \right\rangle_i = \frac{1}{N} \sum_{i=1}^{N} \left\langle \frac{\partial \boldsymbol{g}^{\ell\top}}{\partial \boldsymbol{u}^{\ell}} \right\rangle_i
$$

$$
\boldsymbol{D}^{\ell-1} = \frac{1}{N} \sum_{i=1}^{N} [\boldsymbol{\Phi}^{\ell-1}]^{-1} \left\langle \boldsymbol{u}^{\ell} \tilde{\boldsymbol{g}}^{\ell\top} \right\rangle_i = \frac{1}{N} \sum_{i=1}^{N} \left\langle \frac{\partial \tilde{\boldsymbol{g}}^{\ell\top}}{\partial \boldsymbol{u}^{\ell}} \right\rangle_i
$$

$$(39)$$

$$
\hat{\boldsymbol{\Phi}}^{\ell-1} = \frac{1}{2} [\boldsymbol{\Phi}^{\ell-1}]^{-1} - \frac{1}{2N} \sum_{i=1}^{N} [\boldsymbol{\Phi}^{\ell-1}]^{-1} \left\langle \boldsymbol{u}^{\ell} \boldsymbol{u}^{\ell\top} \right\rangle_i [\boldsymbol{\Phi}^{\ell-1}]^{-1} = 0.
$$

Similarly, using the Gaussianity of $\boldsymbol{r}^{\ell}, \boldsymbol{\zeta}^{\ell}, \hat{\boldsymbol{\zeta}}^{\ell}$ we have

$$
\boldsymbol{A}^{\ell} = \frac{1}{N} \sum_{i=1}^{N} \left\langle \frac{\partial \phi(\boldsymbol{h}^{\ell})}{\partial \boldsymbol{r}^{\ell\top}} \right\rangle_i \,, \; \boldsymbol{C}^{\ell} = \frac{1}{N} \sum_{i=1}^{N} \left\langle \frac{\partial \phi(\boldsymbol{h}^{\ell})}{\partial \boldsymbol{v}^{\ell\top}} \right\rangle_i \,, \; \hat{\boldsymbol{G}}^{\ell+1} = \hat{\tilde{\boldsymbol{G}}}^{\ell+1} = \hat{\tilde{\tilde{\boldsymbol{G}}}}^{\ell+1} = 0.
$$

## B.7 FINAL DMFT EQUATIONS

We now take the limit of zero source $\boldsymbol{j}^{\ell}, \boldsymbol{k}^{\ell}, \boldsymbol{n}^{\ell}, \boldsymbol{p}^{\ell} \to 0$. In this limit, all single site averages $\langle\rangle_i$ become identical so we can simplify the expressions for the order parameters. To "symmetrize" the equations we will also make the substitution $\boldsymbol{B} \to \boldsymbol{B}^{\top}, \boldsymbol{D} \to \boldsymbol{D}^{\top}$. Next, we also rescale all of the response functions $\{A^{\ell}, B^{\ell}, C^{\ell}, D^{\ell}\}$ by $\gamma_0^{-1}$ so that they are $O_{\gamma_0}(1)$ at small $\gamma_0$. This gives us the following set of equations for the order parameters

$$
\Phi_{\boldsymbol{\mu}\nu}^{\ell}(t,s) = \left\langle \phi(h_{\boldsymbol{\mu}}^{\ell}(t)) \phi(h_{\nu}^{\ell}(s)) \right\rangle \,, \; G_{\boldsymbol{\mu}\nu}^{\ell}(t,s) = \left\langle g_{\boldsymbol{\mu}}^{\ell}(t) g_{\nu}^{\ell}(s) \right\rangle \,, \; \tilde{G}_{\boldsymbol{\mu}\nu}^{\ell}(t,s) = \left\langle g_{\boldsymbol{\mu}}^{\ell}(t) \tilde{g}_{\nu}^{\ell}(s) \right\rangle
$$

$$
\tilde{\tilde{G}}_{\boldsymbol{\mu}\nu}^{\ell}(t,s) = \left\langle \tilde{g}_{\boldsymbol{\mu}}^{\ell}(t) \tilde{g}_{\nu}^{\ell}(s) \right\rangle \,, \; A_{\boldsymbol{\mu}\nu}^{\ell}(t,s) = \gamma_0^{-1} \left\langle \frac{\delta \phi(h_{\boldsymbol{\mu}}^{\ell}(t))}{\delta r_{\nu}^{\ell}(s)} \right\rangle \,, \; C_{\boldsymbol{\mu}\nu}^{\ell}(t,s) = \gamma_0^{-1} \left\langle \frac{\delta \phi(h_{\boldsymbol{\mu}}^{\ell}(t))}{\delta v_{\nu}^{\ell}(s)} \right\rangle
$$

$$
B_{\boldsymbol{\mu}\nu}^{\ell}(t,s) = \gamma_0^{-1} \left\langle \frac{\delta g_{\boldsymbol{\mu}}^{\ell+1}(t)}{\delta u_{\nu}^{\ell+1}(s)} \right\rangle \,, \; D_{\boldsymbol{\mu}\nu}^{\ell}(t,s) = \gamma_0^{-1} \left\langle \frac{\delta \tilde{g}_{\boldsymbol{\mu}}^{\ell+1}(t)}{\delta u_{\nu}^{\ell+1}(s)} \right\rangle.
$$

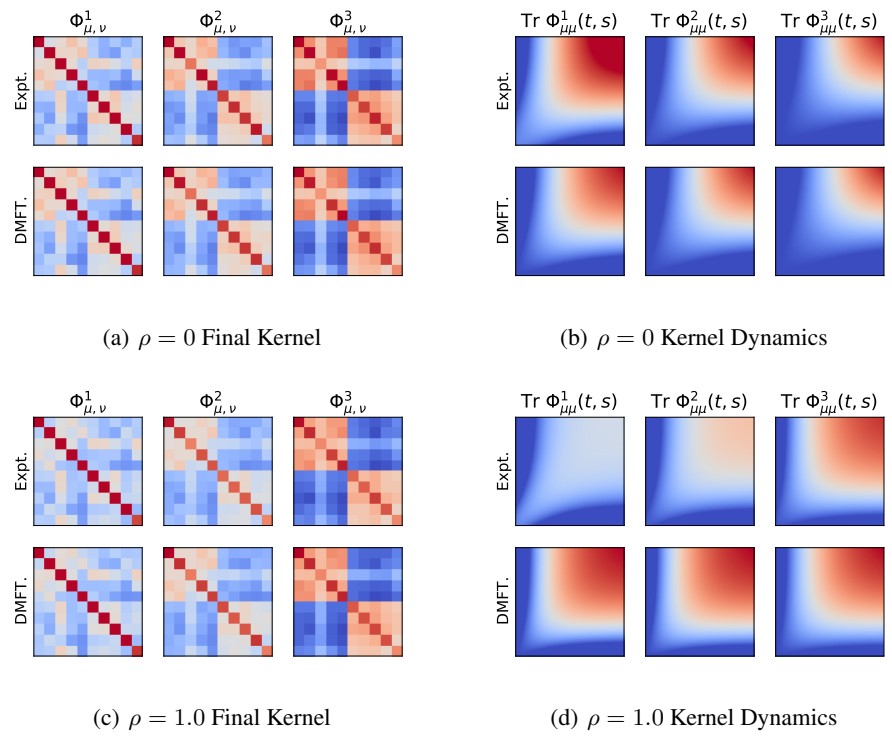

(a) $\rho = 0$ Final Kernel

(b) $\rho = 0$ Kernel Dynamics

(c) $\rho = 1.0$ Final Kernel

(d) $\rho = 1.0$ Kernel Dynamics

Figure 6: Feature kernels $\Phi^\ell$ and their dynamics predicted by solving full set of saddle point equations equation 40 for $\rho$-FA in depth 3 tanh network with $\gamma_0 = 1.0$. Solving deep nonlinear $\rho$-FA requires sampling the full triplet of Gaussian sources $\{u^\ell, r^\ell, v^\ell\}$ for each layer and computing all four response functions $\{A^\ell, B^\ell, C^\ell, D^\ell\}$. DMFT theoretical predictions are compared to a width $N = 3000$ neural network.

For the fields $h_\mu^\ell(t), z_\mu^\ell(t), \tilde{z}_\mu^\ell(t)$, we have the following equations

$$h_\mu^\ell(t) = u_\mu^\ell(t) + \gamma_0 \int_0^t ds \sum_{\nu=1}^P [A_{\mu\nu}^{\ell-1}(t,s) g_\nu^\ell(s) + C_{\mu\nu}^{\ell-1}(t,s) \tilde{g}_\mu^\ell(s) + \Delta_\nu(s) \Phi_{\mu\nu}^{\ell-1}(t,s) \tilde{g}_\nu^\ell(s)]$$

$$z_\mu^\ell(t) = r_\mu^\ell(t) + \gamma_0 \int_0^t ds \sum_{\nu=1}^P [B_{\mu\nu}^\ell(t,s) \phi(h_\nu^\ell(s)) + \Delta_\nu(s) \tilde{G}_{\mu\nu}^{\ell+1}(t,s) \phi(h_\nu^\ell(s))]$$

$$\tilde{g}_\mu^\ell(t) = \begin{cases} \dot\phi(h_\mu^\ell(t)) z_\mu^\ell(t) & \text{GD} \\ \dot\phi(h_\mu^\ell(t)) \left[\sqrt{1-\rho^2}\tilde\zeta_\mu^\ell(t) + \rho v_\mu^\ell(t) + \rho\gamma_0 \int_0^t ds \sum_{\nu=1}^P D_{\mu\nu}^\ell(t,s)\phi(h_\nu^\ell(s))\right] & \rho\text{-FA} \\ \dot\phi(h_\mu^\ell(t)) \tilde{z}^\ell , \ \tilde{z}^\ell \sim \mathcal{N}(0,1) & \text{DFA} \\ \dot\phi(m_\mu^\ell(t)) z_\mu^\ell(t) & \text{GLN} \\ \Delta_\mu(t) \phi(h_\mu^\ell(t)) & \text{Hebb} \end{cases}$$

$$\{u_\mu^\ell(t)\} \sim \mathcal{GP}(0, \mathbf{\Phi}^{\ell-1}) , \ \{r_\mu^\ell(t), v_\mu^\ell(t)\} \sim \mathcal{GP}(0, \mathbf{G}_+^{\ell+1}) , \ \{\tilde\zeta_\mu^\ell(t)\} \sim \mathcal{GP}(0, \tilde{\tilde{\mathbf{G}}}^{\ell+1})$$

$$\mathbf{G}_+^{\ell+1} = \begin{bmatrix} \mathbf{G}^{\ell+1} & \tilde{\mathbf{G}}^{\ell+1} \\ \tilde{\mathbf{G}}^{\ell+1,\top} & \tilde{\tilde{\mathbf{G}}}^{\ell+1} \end{bmatrix}. \tag{40}$$

## C  EXTENSION TO OTHER ARCHITECTURES AND OPTIMIZERS

In this section, we consider the effect of changing architectural details (multiple output channels and convolutional structure) and also optimization choices (momentum, regularization).

## C.1    MULTIPLE OUTPUT CLASSES

Similar to pre-existing work on the GD case (Bordelon & Pehlevan, 2022), our new generalized DMFT can be easily extended to $C$ output channels, provided the number of channels $C$ is not simultaneously taken to infinity with network width $N$. We note that the outputs of the network are now vectors $\boldsymbol{f}_\mu \in \mathbb{R}^C$ and that each eNTK entry is now a $C \times C$ matrix $\boldsymbol{K}_{\mu\nu}(t,s) \in \mathbb{R}^{C \times C}$. The relevant true gradient fields are vectors $\boldsymbol{g}_{c,\mu}^\ell = \frac{\partial f_{c,\mu}}{\partial \boldsymbol{h}_\mu^\ell}$. We construct pseudo-gradients $\tilde{\boldsymbol{g}}_{c,\mu}^\ell$ as before using each of our learning rules. The gradient-pseudogradient kernel $\boldsymbol{G}_{\mu\nu}^\ell \in \mathbb{R}^{C \times C}$ is $\tilde{G}_{c,c',\mu\nu}^\ell = \frac{1}{N} \boldsymbol{g}_{c,\mu}^\ell \cdot \boldsymbol{g}_{c',\nu}^\ell$. The eNTK $\boldsymbol{K}_{\mu\nu} = \sum_\ell \tilde{\boldsymbol{G}}_{\mu\nu}^{\ell+1} \Phi_{\mu\nu}^\ell$ can be used to derive the function dynamics

$$\frac{\partial \boldsymbol{f}_\mu}{\partial t} = \sum_\nu \boldsymbol{K}_{\mu\nu} \boldsymbol{\Delta}_\mu \ , \ \boldsymbol{\Delta}_\mu = -\frac{\partial \mathcal{L}}{\partial \boldsymbol{f}_\mu}. \tag{41}$$

At infinite width $N \to \infty$, the field dynamics for $h_\mu^\ell(t) \in \mathbb{R}$, $\boldsymbol{g}_\mu^\ell(t) \in \mathbb{R}^C$ satisfy

$$h_\mu^\ell(t) = u_\mu^\ell(t) + \gamma_0 \int_0^t ds \sum_{\nu=1}^P \left[ \tilde{\boldsymbol{g}}_\nu^\ell(s) \cdot \boldsymbol{\Delta}_\nu(s) \Phi_{\mu\nu}^{\ell-1}(t,s) + \boldsymbol{A}_{\mu\nu}^{\ell-1}(t,s) \cdot \boldsymbol{g}_\nu^\ell(s) + \boldsymbol{C}_{\mu\nu}^{\ell-1} \cdot \tilde{\boldsymbol{g}}_\nu^\ell(s) \right]$$

$$\boldsymbol{z}_\mu^\ell(t) = \boldsymbol{r}_\mu^\ell(t) + \gamma_0 \int_0^t ds \sum_{\nu=1}^P \phi(h_\nu^\ell(s)) \left[ \tilde{\boldsymbol{G}}_{\mu\nu}^{\ell+1}(t,s) \boldsymbol{\Delta}_\nu(s) + \boldsymbol{B}_{\mu\nu}^\ell(t,s) \right], \tag{42}$$

where $\boldsymbol{A}_{\mu\nu}^\ell(t,s) = \gamma_0 \left\langle \frac{\delta \phi(h_\mu^\ell(t))}{\delta \boldsymbol{r}_\nu^\ell(s)} \right\rangle \in \mathbb{R}^C$, $\boldsymbol{A}_{\mu\nu}^\ell(t,s) = \gamma_0 \left\langle \frac{\delta \phi(h_\mu^\ell(t))}{\delta \boldsymbol{v}_\nu^\ell(s)} \right\rangle \in \mathbb{R}^C$, and $\boldsymbol{B}_{\mu\alpha}(t,s) = \frac{\partial \boldsymbol{g}_\mu^{\ell+1}(t)}{\partial u_\nu^\ell(s)} \in \mathbb{R}^C$. The feature kernels are the same as before $\Phi_{\mu\nu}^\ell(t,s) = \langle \phi(h_\mu^\ell(t)) \phi(h_\nu^\ell(s)) \rangle$ while the gradient-pseudogradient kernel is $\tilde{\boldsymbol{G}}_{\mu\nu}^\ell(t,s) = \langle \boldsymbol{g}_\mu^\ell(t) \tilde{\boldsymbol{g}}_\nu^{\ell\top}(s) \rangle \in \mathbb{R}^{C \times C}$. The pseudogradient fields $\tilde{\boldsymbol{g}}^\ell$ are defined analogously for each learning rule as in the single class setting.

$$\tilde{\boldsymbol{g}}_\mu^\ell(t) = \begin{cases} \dot{\phi}(h_\mu^\ell(t)) \boldsymbol{z}_\mu^\ell(t) & \text{GD} \\ \dot{\phi}(h_\mu^\ell(t)) \left[ \rho \boldsymbol{v}_\mu^\ell(t) + \sqrt{1-\rho^2} \tilde{\boldsymbol{\zeta}}_\mu^\ell(t) + \rho \gamma_0 \int_0^t ds \sum_{\nu=1}^P \boldsymbol{D}_\nu^\ell(t,s) \phi(h_\nu^\ell(s)) \right] & \rho\text{-FA} \\ \dot{\phi}(h_\mu^\ell(t)) \tilde{\boldsymbol{z}}^\ell \ , \ \tilde{\boldsymbol{z}}^\ell \sim \mathcal{N}(0, \boldsymbol{I}) & \text{DFA} \\ \dot{\phi}(m_\mu^\ell(t)) \boldsymbol{z}_\mu^\ell(t) & \text{GLN} \\ \boldsymbol{1} \boldsymbol{\Delta}_\mu(t) \phi(h_\mu^\ell(t)) & \text{Hebb} \end{cases} \tag{43}$$

## C.2    CNN

The DMFT described for each of these learning rules can also be extended to CNNs with infinitely many channels. Following the work of Bordelon & Pehlevan (2022) Appendix G on the GD DMFT limit for CNNs, we let $W_{ij,\mathfrak{a}}^\ell$ represent the value of the filter at spatial displacement $\mathfrak{a}$ from the center of the filter, which maps relates activity at channel $j$ of layer $\ell$ to channel $i$ of layer $\ell+1$. The fields $h_{\mu,i,\mathfrak{a}}^\ell$ satisfy the recursion

$$h_{\mu,i,\mathfrak{a}}^{\ell+1} = \frac{1}{\sqrt{N}} \sum_{j=1}^N \sum_{\mathfrak{b} \in \mathcal{S}^\ell} W_{ij,\mathfrak{b}}^\ell \phi(h_{\mu,j,\mathfrak{a}+\mathfrak{b}}^\ell) \ , \ i \in [N], \tag{44}$$

where $\mathcal{S}^\ell$ is the spatial receptive field at layer $\ell$. For example, a $(2k+1) \times (2k+1)$ convolution will have $\mathcal{S}^\ell = \{(i,j) \in \mathbb{Z}^2 : -k \le i \le k, -k \le j \le k\}$. The output function is obtained from the last layer is defined as $f_\mu = \frac{1}{\gamma_0 N} \sum_{i=1}^N \sum_{\mathfrak{a}} w_{i,\mathfrak{a}}^L \phi(h_{\mu,i,\mathfrak{a}}^L)$. The true gradient fields have the same definition as before $\boldsymbol{g}_{\mu,\mathfrak{a}}^\ell = \gamma_0 N \frac{\partial f_\mu}{\partial \boldsymbol{h}_{\mu,\mathfrak{a}}^\ell} \in \mathbb{R}^N$, which as before enjoy the following recursion

$$\boldsymbol{g}_{\mu,\mathfrak{a}}^\ell = \gamma_0 N \sum_{\mathfrak{b}} \frac{\partial f_\mu}{\partial \boldsymbol{h}_{\mu,\mathfrak{b}}^{\ell+1}} \cdot \frac{\partial \boldsymbol{h}_{\mu,\mathfrak{b}}^{\ell+1}}{\partial \boldsymbol{h}_{\mu,\mathfrak{a}}^\ell} = \dot{\phi}(\boldsymbol{h}_{\mu,\mathfrak{a}}^\ell) \odot \left[ \frac{1}{\sqrt{N}} \sum_{j=1}^N \sum_{\mathfrak{b} \in \mathcal{S}^\ell} \boldsymbol{W}_\mathfrak{b}^{\ell\top} \boldsymbol{g}_{\mu,\mathfrak{a}-\mathfrak{b}}^{\ell+1} \right]. \tag{45}$$

We consider the following learning dynamics for the filters

$$\frac{d}{dt}\boldsymbol{W}_{\mathfrak{b}}^{\ell} = \frac{\gamma_0}{\sqrt{N}} \sum_{\mu,\mathfrak{c}} \Delta_\mu \tilde{\boldsymbol{g}}_{\mu,\mathfrak{c}}^{\ell+1} \phi(\boldsymbol{h}_{\mu,\mathfrak{c}+\mathfrak{b}}^{\ell})^\top \tag{46}$$

where as before $\tilde{\boldsymbol{g}}^\ell$ is determined by the learning rule. The relevant kernel order parameters now have spatial indices. For instance the feature kernel at each layer has form $\Phi_{\mu,\nu,\mathfrak{ab}}^\ell = \frac{1}{N}\phi(\boldsymbol{h}_{\mu,\mathfrak{a}}^\ell(t)) \cdot \phi(\boldsymbol{h}_{\nu,\mathfrak{b}}^\ell(s))$. At the infinite width $N \to \infty$, the order parameters and field dynamics have the form

$$h_{\mu,\mathfrak{a}}^\ell(t) = u_{\mu,\mathfrak{a}}^\ell(t) + \gamma_0 \int_0^t ds \sum_{\nu,\mathfrak{b},\mathfrak{c}} \Delta_\nu(s) \Phi_{\mu\nu,\mathfrak{a}+\mathfrak{b},\mathfrak{b}+\mathfrak{c}}^{\ell-1}(t,s) \tilde{g}_{\nu,\mathfrak{c}}^\ell(s) \tag{47}$$

$$+ \gamma_0 \int_0^t ds \sum_{\nu,\mathfrak{b}} [A_{\mu\nu,\mathfrak{ab}}^{\ell-1}(t,s) g_{\nu,\mathfrak{b}}^\ell(s) + C_{\mu\nu,\mathfrak{ab}}^{\ell-1}(t,s) \tilde{g}_{\nu\mathfrak{b}}^\ell(s)]$$

$$z_{\mu,\mathfrak{a}}^\ell(t) = r_{\mu,\mathfrak{a}}^\ell(t) + \gamma_0 \int_0^t ds \sum_{\nu,\mathfrak{b},\mathfrak{c}} \tilde{G}_{\mu\nu,\mathfrak{a}-\mathfrak{b},\mathfrak{c}-\mathfrak{b}}^{\ell+1}(t,s) \phi(h_{\nu,\mathfrak{c}}^\ell(s))$$

$$+ \gamma_0 \int_0^t ds \sum_{\nu,\mathfrak{b}} B_{\mu\nu,\mathfrak{ab}}^\ell(t,s) \phi(h_{\nu,\mathfrak{b}}^\ell(s)) \tag{48}$$

where correlation and response functions have the usual definitions

$$\Phi_{\mu\alpha,\mathfrak{ab}}^\ell(t,s) = \langle \phi(h_{\mu\mathfrak{a}}^\ell(t)) \phi(h_{\alpha\mathfrak{b}}^\ell(s)) \rangle \;,\; G_{\mu\alpha,\mathfrak{ab}}^\ell(t,s) = \langle g_{\mu\mathfrak{a}}^\ell(t) g_{\alpha\mathfrak{b}}^\ell(s) \rangle \;,\; \tilde{G}_{\mu\nu,\mathfrak{ab}}^\ell(t,s) = \langle g_{\mu\mathfrak{a}}^\ell(t) \tilde{g}_{\alpha\mathfrak{b}}^\ell(s) \rangle$$

$$A_{\mu\alpha,\mathfrak{ab}}^\ell(t,s) = \frac{1}{\gamma_0} \left\langle \frac{\delta\phi(h_{\mu\mathfrak{a}}^\ell(t))}{\delta r_{\alpha\mathfrak{b}}^\ell(s)} \right\rangle \;,\; B_{\mu\alpha,\mathfrak{ab}}^\ell(t,s) = \frac{1}{\gamma_0} \left\langle \frac{\delta g_{\mu\mathfrak{a}}^{\ell+1}(t)}{\delta u_{\alpha\mathfrak{b}}^{\ell+1}(s)} \right\rangle \;. \tag{49}$$

## C.3 L2 Regularization (Weight Decay)

L2 regularization on the weights $\boldsymbol{W}^\ell$ (weight decay) can also be modeled within DMFT. We start by looking at the weight dynamics

$$\frac{d}{dt}\boldsymbol{W}^\ell = \frac{\gamma_0}{\sqrt{N}} \sum_{\mu=1}^P \Delta_\mu \tilde{\boldsymbol{g}}_\mu^{\ell+1} \phi(\boldsymbol{h}_\mu^\ell)^\top - \lambda \boldsymbol{W}^\ell$$

$$\implies \boldsymbol{W}^\ell(t) = e^{-\lambda t} \boldsymbol{W}^\ell(0) + \frac{\gamma_0}{\sqrt{N}} \int_0^t ds \, e^{-\lambda(t-s)} \sum_\mu \Delta_\mu(s) \tilde{\boldsymbol{g}}_\mu^{\ell+1}(s) \phi(\boldsymbol{h}_\mu^\ell(s))^\top \tag{50}$$

In the second line we used an integrating factor $e^{\lambda t}$. We can thus arrive at the following feature dynamics in the DMFT limit

$$h_\mu^\ell(t) = e^{-\lambda t} \chi_\mu^\ell(t) + \gamma_0 \int_0^t ds \, e^{-\lambda(t-s)} \sum_\nu \Delta_\nu(s) \Phi_{\mu\nu}^{\ell-1}(s) \tilde{g}_\nu^\ell(s)$$

$$z_\mu^\ell(t) = e^{-\lambda t} \xi_\mu^\ell(t) + \gamma_0 \int_0^t ds \, e^{-\lambda(t-s)} \sum_\nu \Delta_\nu(s) \tilde{G}_{\mu\nu}^{\ell+1}(s) \phi(h_\nu^\ell(s)).$$

The $\tilde{g}$ dynamics are also modified appropriately with factors of $e^{-\lambda t}$ and $e^{-\lambda(t-s)}$ for each of our learning rules. We see that the contribution from the initial conditions $\chi, \xi$ are suppressed at late times while the feature learning update which is $O(\gamma_0/\lambda)$ in the first layer dominates scale of the final features.

## C.4 Momentum

Momentum uses a low-pass filtered version of the gradients to update the weights (Goh, 2017). A continuous time limit of momentum dynamics on the trainable parameters $\{\boldsymbol{W}^\ell\}$ would give the

following differential equations

$$\frac{\partial}{\partial t}\boldsymbol{W}^\ell(t) = \boldsymbol{Q}^\ell(t)$$

$$\tau\frac{d}{dt}\boldsymbol{Q}^\ell(t) = -\boldsymbol{Q}^\ell + \frac{\gamma_0}{\sqrt{N}}\sum_\mu \Delta_\mu(t)\tilde{\boldsymbol{g}}_\mu^{\ell+1}(t)\phi(\boldsymbol{h}_\mu^\ell(t))^\top. \tag{51}$$

We write the expression this way so that the small time constant $\tau \to 0$ limit corresponds to classic gradient descent. Integration of the $\boldsymbol{Q}^\ell(t)$ dynamics gives the following integral expression for $\boldsymbol{W}^\ell$

$$\boldsymbol{W}^\ell(t) = \boldsymbol{W}^\ell(0) + \frac{\gamma_0}{\sqrt{N}\tau}\int_0^t dt' \int_0^{t'} dt'' e^{-(t'-t'')/\tau}\sum_\mu \Delta_\mu(t'')\tilde{\boldsymbol{g}}_\mu^{\ell+1}(t'')\phi(\boldsymbol{h}_\mu^\ell(t''))^\top. \tag{52}$$

These weight dynamics give rise to the following field evolution

$$h_\mu^{\ell+1}(t) = \chi_\mu^{\ell+1}(t) + \frac{\gamma_0}{\tau}\int_0^t dt' \int_0^{t'} dt'' e^{-(t'-t'')/\tau}\sum_\nu \Delta_\nu(t'')\tilde{g}_\nu^{\ell+1}(t'')\Phi_{\mu\nu}^\ell(t,t'')$$

$$z_\mu^\ell(t) = \xi_\mu^\ell(t) + \frac{\gamma_0}{\tau}\int_0^t dt' \int_0^{t'} dt'' e^{-(t'-t'')/\tau}\sum_\nu dt'' \Delta_\alpha(t'')\tilde{G}_{\mu\nu}^{\ell+1}(t,t'')\phi(h_\nu^\ell(t'')). \tag{53}$$

We see that in the $\tau \to 0$ limit, the $t''$ integral is dominated by the contribution at $t'' \sim t'$ recovering usual gradient descent dynamics. For $\tau \gg 0$, we see that the integral accumulates additional contributions from the past values of fields and kernels.

## D  LAZY LIMITS

In this section we discuss the lazy $\gamma_0 \to 0$ limit. In this limit we see that $h^\ell(t) = u^\ell(t)$ and $z^\ell(t) = r^\ell(t)$ for all time $t$. Since the input data gram matrix $\Phi_{\mu\nu}^0 = \frac{1}{D}\boldsymbol{x}_\mu \cdot \boldsymbol{x}_\nu$ is a constant in time the sources in the first hidden layer $u_\mu^1$ are constant in time. Consequently, the first layer feature kernel is constant in time since

$$\Phi_{\mu\nu}^1(t,s) = \langle \phi(h_\mu^1(t))\phi(h_\nu^1(s))\rangle = \langle \phi(u_\mu^1)\phi(u_\nu^1)\rangle_{\boldsymbol{u}^1 \sim \mathcal{N}(0,\boldsymbol{\Phi}^0)}. \tag{54}$$

Now, we see that this argument can proceed inductively. Since $\boldsymbol{\Phi}^1$ is time-independent, the second layer fields $\boldsymbol{h}^2 = \boldsymbol{u}^2 \sim \mathcal{N}(0,\boldsymbol{\Phi}^1)$ are also constant in time, implying $\boldsymbol{\Phi}^2$ is constant in time. This argument is repeated for all layer $\ell \in [L]$. Similarly, we can analyze the backward pass fields $\boldsymbol{z}^\ell$. Since $\boldsymbol{z}^L \sim \mathcal{N}(0,\boldsymbol{G}^{L+1})$ are constant, then $\boldsymbol{z}^\ell$ are time-independent for all $\ell$. It thus suffices to compute the static kernels $\{\Phi^\ell, G^\ell, \tilde{G}^\ell\}$ at initialization

$$\boldsymbol{\Phi}^\ell = \langle \phi(\boldsymbol{u}^\ell)\phi(\boldsymbol{u}^\ell)^\top\rangle_{\boldsymbol{u}^\ell \sim \mathcal{N}(0,\boldsymbol{\Phi}^{\ell-1})}$$

$$\boldsymbol{G}^\ell = \langle [\dot{\phi}(\boldsymbol{u}^\ell)\odot \boldsymbol{r}^\ell][\dot{\phi}(\boldsymbol{u}^\ell)\odot \boldsymbol{r}^\ell]^\top\rangle_{\boldsymbol{u}^\ell \sim \mathcal{N}(0,\boldsymbol{\Phi}^{\ell-1}),\boldsymbol{r}^\ell \sim \mathcal{N}(0,\boldsymbol{G}^{\ell+1})}$$

$$= \boldsymbol{G}^{\ell+1}\odot \dot{\boldsymbol{\Phi}}^\ell \ , \ \dot{\boldsymbol{\Phi}}^\ell = \langle \dot{\phi}(\boldsymbol{u}^\ell)\dot{\phi}(\boldsymbol{u}^\ell)^\top\rangle_{\boldsymbol{u}^\ell \sim \mathcal{N}(0,\boldsymbol{\Phi}^{\ell-1})}. \tag{55}$$

where in the last line we utilized the independence of $\boldsymbol{u}^\ell, \boldsymbol{r}^\ell$. These above equations give a forward pass recursion for the $\boldsymbol{\Phi}^\ell$ kernels and the backward pass recursion for $\boldsymbol{G}^\ell$. Lastly, depending on the learning rule, we arrive at the following definitions for $\tilde{\boldsymbol{G}}^\ell$ for $\ell \in \{1,...,L\}$

$$\tilde{\boldsymbol{G}}^\ell = \langle [\dot{\phi}(\boldsymbol{u}^\ell)\odot \boldsymbol{r}^\ell]\tilde{\boldsymbol{g}}^{\ell\top}\rangle = \begin{cases} \boldsymbol{G}^{\ell+1}\odot \dot{\boldsymbol{\Phi}}^\ell & \text{GD} \\ \rho\tilde{\boldsymbol{G}}^{\ell+1}\odot \dot{\boldsymbol{\Phi}}^\ell & \rho\text{-FA} \\ 0 & \text{DFA,Hebb} \\ \boldsymbol{G}^{\ell+1}\odot \langle \dot{\phi}(\boldsymbol{m}^\ell)\dot{\phi}(\boldsymbol{m}^\ell)^\top\rangle & \text{GLN} \end{cases} \tag{56}$$

Using these results for $\tilde{\boldsymbol{G}}$, we can compute the initial eNTK $\boldsymbol{K} = \sum_{\ell=0}^L \tilde{\boldsymbol{G}}^{\ell+1}\odot \boldsymbol{\Phi}^\ell$ which governs prediction dynamics.

### D.1 LAZY LIMIT PERFORMANCES ON REALISTIC TASKS

We note that, while the DMFT equations on $P$ datapoints and $T$ timesteps require $O(P^3 T^3)$ time complexity to solve in the rich regime, the lazy limit gives neural network predictions in $O(P^3)$ time, since the predictor can be obtained by solving a linear system of $P$ equations. The performance of these lazy limit kernels on realistic tasks would match the performances reported by Lee et al. (2020). Specifically, GD and $\rho = 1$ FA would match the test accuracy reported for "infinite width GD", while $\rho = 0$ FA, DFA, and Hebbian rules would match "infinite width Bayesian" networks in Figure 1 of Lee et al. (2020).

## E   DEEP LINEAR NETWORKS

In deep linear networks, the DMFT equations close without needing any numerical sampling procedure, as was shown in prior work on the GD case (Yang & Hu, 2021; Bordelon & Pehlevan, 2022). The key observation is that for all of the following learning rules, the fields $\{\boldsymbol{h}, \boldsymbol{g}, \tilde{\boldsymbol{g}}\}$ are linear combinations of the Gaussian sources $\{u, r, v\}$, and are thus Gaussian themselves. Concretely, we introduce a vector notation $\boldsymbol{h}^\ell = \mathrm{Vec}\{h_\mu^\ell(t)\}$ and $\boldsymbol{g}^\ell = \mathrm{Vec}\{g_\mu^\ell(t)\}$, etc. We have in each layer

$$\boldsymbol{h}^\ell = \boldsymbol{R}_{h,u}\boldsymbol{u}^\ell + \boldsymbol{R}_{h,r}\boldsymbol{r}^\ell + \boldsymbol{R}_{h,v}\boldsymbol{v}^\ell + \boldsymbol{R}_{h,\tilde{\zeta}}\tilde{\boldsymbol{\zeta}}^\ell$$

$$\boldsymbol{g}^\ell = \boldsymbol{R}_{g,u}\boldsymbol{u}^\ell + \boldsymbol{R}_{g,r}\boldsymbol{r}^\ell + \boldsymbol{R}_{g,v}\boldsymbol{v}^\ell + \boldsymbol{R}_{g,\tilde{\zeta}}\tilde{\boldsymbol{\zeta}}^\ell$$

$$\tilde{\boldsymbol{g}}^\ell = \boldsymbol{R}_{\tilde{g},u}\boldsymbol{u}^\ell + \boldsymbol{R}_{\tilde{g},r}\boldsymbol{r}^\ell + \boldsymbol{R}_{\tilde{g},v}\boldsymbol{v}^\ell + \boldsymbol{R}_{\tilde{g},\tilde{\zeta}}\tilde{\boldsymbol{\zeta}}^\ell$$

where the matrices $\boldsymbol{R}$ depend on the learning rule and the data. The necessary kernels $\boldsymbol{H}^\ell = \langle \boldsymbol{h}^\ell \boldsymbol{h}^{\ell\top} \rangle$ can thus be closed algebraically since all of the correlation statistics of the sources $\{u, r, v\}$ have known two-point correlation statistics.

### E.1   LINEAR NETWORK TRAINED WITH GD

The $\boldsymbol{R}$ matrices for GD were provided in (Bordelon & Pehlevan, 2022). We start by noting the following DMFT equations for $\boldsymbol{h}^\ell, \boldsymbol{g}^\ell$

$$\boldsymbol{h}^\ell = \boldsymbol{u}^\ell + \gamma_0(\boldsymbol{A}^{\ell-1} + \boldsymbol{H}_{\boldsymbol{\Delta}}^{\ell-1})\boldsymbol{g}^\ell \ , \ \boldsymbol{g}^\ell = \boldsymbol{r}^\ell + \gamma_0(\boldsymbol{B}^\ell + \boldsymbol{G}_{\boldsymbol{\Delta}}^{\ell+1})\boldsymbol{h}^\ell \tag{57}$$

where $[\boldsymbol{H}_{\boldsymbol{\Delta}}^{\ell-1}]_{\mu\nu,ts} = H_{\mu\nu}^\ell(t,s)\Delta_\nu(s)$. Isolating the dependence of these equations on $\boldsymbol{u}$ and $\boldsymbol{r}$, we have

$$\left[\boldsymbol{I} - \gamma_0^2(\boldsymbol{A}^{\ell-1} + \boldsymbol{H}_{\boldsymbol{\Delta}}^{\ell-1})(\boldsymbol{B}^\ell + \boldsymbol{G}_{\boldsymbol{\Delta}}^{\ell+1})\right]\boldsymbol{h}^\ell = \boldsymbol{u}^\ell + \gamma_0^2(\boldsymbol{A}^{\ell-1} + \boldsymbol{H}_{\boldsymbol{\Delta}}^{\ell-1})(\boldsymbol{B}^\ell + \boldsymbol{G}_{\boldsymbol{\Delta}}^{\ell+1})\boldsymbol{r}^\ell$$

$$\left[\boldsymbol{I} - \gamma_0^2(\boldsymbol{B}^\ell + \boldsymbol{G}_{\boldsymbol{\Delta}}^{\ell+1})(\boldsymbol{A}^{\ell-1} + \boldsymbol{H}_{\boldsymbol{\Delta}}^{\ell-1})\right]\boldsymbol{g}^\ell = \boldsymbol{r}^\ell + \gamma_0^2(\boldsymbol{B}^\ell + \boldsymbol{G}_{\boldsymbol{\Delta}}^{\ell+1})(\boldsymbol{A}^{\ell-1} + \boldsymbol{H}_{\boldsymbol{\Delta}}^{\ell-1})\boldsymbol{r}^\ell. \tag{58}$$

These equations can easily be closed for $\boldsymbol{H}^\ell$ and $\boldsymbol{G}^\ell$.

### E.2   $\rho$-ALIGNED FEEDBACK ALIGNMENT

In $\rho$-FA we define the following pseudo-gradient fields

$$\tilde{\boldsymbol{g}}^\ell = \sqrt{1-\rho^2}\tilde{\boldsymbol{\zeta}}^\ell + \rho\boldsymbol{v}^\ell + \rho\gamma_0\boldsymbol{D}^\ell\boldsymbol{h}^\ell \tag{59}$$

Next, we note that, at initialization, the $\tilde{\boldsymbol{G}}^\ell$ can be computed recursively

$$\tilde{\boldsymbol{G}}^\ell = \rho\tilde{\boldsymbol{G}}^{\ell+1} \tag{60}$$

We note that $\frac{\partial}{\partial \boldsymbol{r}^1}\boldsymbol{h}^1 = 0$ which implies $\boldsymbol{A}^1 = 0$. Similarly we have $\frac{\partial}{\partial \boldsymbol{r}^2}\boldsymbol{h}^2 = 0$. Thus $\boldsymbol{A}^2 = 0$. Proceeding inductively, we find $\boldsymbol{A}^\ell = 0$. Similarly, we note that $\frac{\partial \tilde{\boldsymbol{g}}^L}{\partial \boldsymbol{u}^L} = 0$ so $\boldsymbol{D}^{L-1} = 0$. Inductively, we have $\boldsymbol{D}^\ell = 0$ for all $\ell$. Using these facts, we thus find the following equations

$$\boldsymbol{h}^\ell = \boldsymbol{u}^\ell + \gamma_0(\boldsymbol{C}^{\ell-1} + \boldsymbol{H}_{\boldsymbol{\Delta}}^{\ell-1})\tilde{\boldsymbol{g}}^\ell \tag{61}$$

$$\boldsymbol{g}^\ell = \boldsymbol{r}^\ell + \gamma_0(\boldsymbol{B}^\ell + \boldsymbol{G}_{\boldsymbol{\Delta}}^{\ell+1})\boldsymbol{h}^\ell \tag{62}$$

$$\tilde{\boldsymbol{g}}^\ell = \sqrt{1-\rho^2}\tilde{\boldsymbol{\zeta}}^\ell + \rho\boldsymbol{v}^\ell \tag{63}$$

We can close these equations for $\boldsymbol{H}^\ell$ and $\tilde{\boldsymbol{G}}^\ell$

$$\boldsymbol{H}^\ell = \boldsymbol{H}^{\ell-1} + \gamma_0^2 \left(\boldsymbol{C}^{\ell-1} + \boldsymbol{H}_{\boldsymbol{\Delta}}^{\ell-1}\right) \tilde{\boldsymbol{G}}^{\ell+1} \left(\boldsymbol{C}^{\ell-1} + \boldsymbol{H}_{\boldsymbol{\Delta}}^{\ell-1}\right)^\top$$
$$\tilde{\boldsymbol{G}}^\ell = \rho \tilde{\boldsymbol{G}}^{\ell+1} + \gamma_0^2 \left(\boldsymbol{B}^\ell + \boldsymbol{G}_{\boldsymbol{\Delta}}^{\ell+1}\right) \left(\boldsymbol{C}^{\ell-1} + \boldsymbol{H}_{\boldsymbol{\Delta}}^{\ell-1}\right) \tilde{\boldsymbol{G}}^{\ell+1}$$
$$\boldsymbol{C}^\ell = \rho \left(\boldsymbol{C}^{\ell-1} + \boldsymbol{H}_{\boldsymbol{\Delta}}^{\ell-1}\right) , \; \boldsymbol{B}^\ell = \boldsymbol{B}^{\ell+1} + \boldsymbol{G}_{\boldsymbol{\Delta}}^{\ell+2}. \tag{64}$$

The matrices $\tilde{\tilde{\boldsymbol{G}}}^\ell = \boldsymbol{1}\boldsymbol{1}^\top$ are all rank one. Thus it suffices to compute the vectors $\boldsymbol{c}^\ell = \left(\boldsymbol{C}^{\ell-1} + \boldsymbol{H}_{\boldsymbol{\Delta}}^{\ell-1}\right)\boldsymbol{1}$. Further, it suffices to consider $\boldsymbol{d}^\ell = \tilde{\boldsymbol{G}}^\ell \boldsymbol{1}/|\boldsymbol{1}|^2$. With this formalism we have

$$\boldsymbol{H}^\ell = \boldsymbol{H}^{\ell-1} + \gamma_0^2 \boldsymbol{c}^\ell \boldsymbol{c}^{\ell\top} , \; \boldsymbol{d}^\ell = \rho \boldsymbol{d}^{\ell+1} + \gamma_0^2 (\boldsymbol{B}^\ell + \boldsymbol{G}_{\boldsymbol{\Delta}}^{\ell+1}) \boldsymbol{c}^\ell. \tag{65}$$

The analysis for DFA and Hebb rules is very similar.

# F   EXACTLY SOLVEABLE 2 LAYER LINEAR MODEL

## F.1   GRADIENT FLOW

Based on the prior results from (Bordelon & Pehlevan, 2022), the $H_y = \boldsymbol{y}^\top \boldsymbol{H} \boldsymbol{y}/|\boldsymbol{y}|^2$ dynamics for GD are coupled to the dynamics for the error $\Delta(t) = \frac{1}{|\boldsymbol{y}|} \boldsymbol{y} \cdot \boldsymbol{\Delta}(t)$ have the form

$$\frac{d}{dt} H_y(t) = 2\gamma_0^2 (y - \Delta)\Delta , \; \frac{d}{dt}\Delta = -2H_y \Delta. \tag{66}$$

These dynamics have the conservation law $\frac{d}{dt} H_y^2 = \gamma_0^2 \frac{d}{dt}(y - \Delta)^2$. Integrating this conservation law from time 0 to time $t$, we find $H_y(t)^2 = 1 + \gamma_0^2 (y - \Delta(t))^2$. We can therefore solve a single ODE for $\Delta(t)$, giving the following simplified dynamics

$$\frac{d}{dt}\Delta = -2\sqrt{1 + \gamma_0^2 (y - \Delta)^2}\Delta , \; H_y = \sqrt{1 + \gamma_0^2 (y - \Delta)^2}. \tag{67}$$

These dynamics interpolate between exponential convergence (at small $\gamma_0$) and a logistic convergence (at large $\gamma_0$) of $\Delta(t)$ to zero. Since $\Delta \to 0$ at late time, the final value of the kernel alignment is $H_y = \sqrt{1 + \gamma_0^2 y^2}$.

## F.2   $\rho$-ALIGNED FA

For the two layer linear network, the $\rho$-FA field dynamics are

$$\frac{d}{dt} h_\mu(t) = \gamma_0 \sum_\nu \tilde{g}_\nu(t) \Delta_\nu(s) K_{\mu\nu}^x , \; \frac{d}{dt} g_\mu(t) = \gamma_0 \sum_\nu \Delta_\nu(t) h_\nu(t). \tag{68}$$

FA we have $\tilde{g}_\mu(t) = \tilde{g} \sim \mathcal{N}(0,1)$ which is a constant standard normal. We let $a_\mu(t) = \langle \tilde{g} h_\mu(t) \rangle$. The dynamics for $H_{\mu\nu}$ and $a_\mu$ are coupled

$$\frac{d}{dt} H_{\mu\nu} = \gamma_0 a_\nu(t) \sum_\nu \Delta_\nu(t) K_{\mu\nu}^x + \gamma_0 a_\mu(t) \sum_\nu \Delta_\nu(t) K_{\mu\nu}^x$$

$$\frac{d}{dt} a_\mu(t) = \gamma_0 \sum_\nu \Delta_\nu K_{\mu\nu}^x$$

$$\frac{d}{dt} \tilde{G}(t) = \gamma_0 \sum_\mu \Delta_\mu(t) a_\mu(t) , \; \frac{d}{dt}\Delta_\mu(t) = -\sum_\nu [H_{\mu\nu}(t) + \tilde{G}(t) K_{\mu\nu}^x]\Delta_\nu(t). \tag{69}$$

Whitening the dataset $\boldsymbol{K}^x = \boldsymbol{I}$ and projecting all dynamics on $\hat{\boldsymbol{y}}$ subspace gives the reduced dynamics

$$\frac{d}{dt} H = 2\gamma_0 a\Delta , \; \frac{d}{dt} a = \gamma_0 \Delta , \; \frac{d}{dt}\tilde{G} = \gamma_0 \Delta a , \; \frac{d}{dt}\Delta = -[H + \tilde{G}]\Delta. \tag{70}$$

From these dynamics we identify the following set of conservation laws

$$2\frac{d}{dt}\tilde{G} = \frac{d}{dt}a^2 = \frac{d}{dt}H$$
$$\implies 2\tilde{G} - 2\rho = a^2 = H - 1. \tag{71}$$

Writing everything in terms of $\Delta, a$ we have

$$\frac{d}{dt}a = \gamma_0 \Delta \;,\; \frac{d}{dt}\Delta = -\left[\frac{3}{2}a^2 + (1+\rho)\right]\Delta = -\gamma_0^{-1}\frac{d}{dt}\left[\frac{1}{2}a^3 + (1+\rho)a\right]$$

Integrating both sides of this equation from 0 to $t$ gives $\Delta = y - \gamma_0^{-1}\left[\frac{1}{2}a^3 + (1+\rho)a\right]$. Thus, the $a$ dynamics now one dimensional, giving

$$\frac{d}{dt}a = \gamma_0 y - \frac{1}{2}a^3 - (1+\rho)a. \tag{72}$$

When run from initial condition $a = 0$, this will converge to the smallest positive root of the cubic equation $\frac{1}{2}a^3 + (1+\rho)a = \gamma_0 y$. This implies that, for small $\gamma_0$ we have $a \sim \frac{\gamma_0 y}{1+\rho}$ so that $\Delta H = 2\Delta\tilde{G} \sim \frac{\gamma_0^2 y^2}{(1+\rho)^2}$ and so that larger initial alignment $\rho$ leads to smaller changes in the feature kernel and pseudo-gradient alignment kernel. At large $\gamma_0 y$, we have that $a \sim (2\gamma_0 y)^{1/3}$ so that $\Delta H = 2\Delta\tilde{G} \sim (2\gamma_0 y)^{2/3}$.

### F.3 HEBB

For the Hebb rule, $\tilde{G}_\mu = \langle gh_\mu \rangle \Delta_\mu = \gamma_0 f_\mu \Delta_\mu = \gamma_0 (y_\mu - \Delta_\mu)\Delta_\mu$. Under the whitening assumption $K^x_{\mu\nu} = \delta_{\mu\nu}$, the dynamics decouples over samples

$$\frac{d}{dt}H_{\mu,\mu} = 2\gamma_0 H_{\mu\mu}\Delta_\mu^2 \;,\; \frac{d}{dt}\Delta_\mu = -[H_{\mu\mu} + \gamma_0(y_\mu - \Delta_\mu)\Delta_\mu]\Delta_\mu. \tag{73}$$

We see that $H_{\mu\mu}$ strictly increases. The possible fixed points for $\Delta_\mu$ are $\Delta_\mu = 0$ or $\Delta_\mu = \frac{1}{2}\left[y_\mu \pm \sqrt{y_\mu^2 + \gamma_0^{-1}H_{\mu\mu}}\right]$. One of these roots shares a sign with $y_\mu$ and has larger absolute value. The other root has the opposite sign from $y_\mu$. From the initial condition $\Delta_\mu = y_\mu$ and $H_{\mu\mu} = 1$, $\Delta_\mu$ is initially approaching decreasing in absolute value so that $|\Delta_\mu| \in (0, |y_\mu|)$ and will have the same sign as $y_\mu$. In this regime $\frac{d}{dt}|\Delta_\mu| < 0$. Thus, the system will eventually reach the fixed point at $\Delta_\mu = 0$, rather than increasing in magnitude to the root which shares a sign with $y_\mu$ or continuing to the root with the opposite sign as $y_\mu$.

## G  DISCUSSION OF MODIFIED HEBB RULE

We chose to modify the traditional Hebb rule to include a weighing of each example by its instantaneous error. In this section we discuss this choice and provide a brief discussion of alternatives

- *Traditional Hebb Learning*: $\frac{d}{dt}W^\ell \propto \sum_\mu \phi(h^{\ell+1})\phi(h^\ell)^\top$. In the absence of regularization or normalization, this learning rule will continue to update the weights even once the task is fully learned, leading to divergences at infinite time $t \to \infty$.
- *Single Power of the Error*: $\frac{d}{dt}W^\ell \propto \sum_\mu \Delta_\mu \phi(h^{\ell+1})\phi(h^\ell)^\top$. While this rule may naively appear plausible, it can only learn training points with positive target values $y_\mu$ in a linear network if $\gamma_0 > 0$. Further this rule only gives Hebbian updates when $\Delta_\mu > 0$.
- *Two Powers of the Error*: $\frac{d}{dt}W^\ell \propto \sum_\mu \Delta_\mu^2 \phi(h^{\ell+1})\phi(h^\ell)^\top$. This was our error modified Hebb rule. We note that the update always has the correct sign for a Hebbian update and the updates stop when the network converges to zero error, preventing divergence of the features at late time.

## H  FINITE SIZE EFFECTS

We can reason about the fluctuations of $q$ around the saddle point $q^*$ at large but finite $N$ using a Taylor expansion of the DMFT action $S$ around the saddle point. This argument will show that

at large but finite $N$, we can treat $\boldsymbol{q}$ as fluctuating over initializations with mean $\boldsymbol{q}^*$ and variance $O(N^{-1})$. We will first illustrate the mechanics of this computation of an arbitrary observable with a scalar example before applying this to the DMFT.

## H.1 SCALAR EXAMPLE

Suppose we have a scalar variable $q$ with a distribution defined by Gibbs measure $\frac{e^{-NS[q]}}{\int dq e^{-NS[q]}}$ for action $S$. We consider averaging some arbitrary observable $\mathcal{O}(q)$ over this distribution

$$\langle \mathcal{O}(q) \rangle = \frac{\int dq \exp\left(-NS[q]\right) \mathcal{O}(q)}{\int dq \exp\left(-NS[q]\right)}. \tag{74}$$

We Taylor expand $S$ around its saddle point $q^*$ giving $S[q] = S[q^*] + \frac{1}{2}S''[q^*](q - q^*)^2 + \sum_{k=3}^{\infty} S^{(k)}[q^*](q - q^*)^k$. This gives

$$\langle O(q) \rangle = \frac{\int dq \exp\left(-N[\frac{1}{2}S''[q^*](q - q^*)^2 - \sum_{k=3}^{\infty} S^{(k)}[q^*](q - q^*)^k]\right) \mathcal{O}(q)}{\int dq \exp\left(-N[\frac{1}{2}S''[q^*](q - q^*)^2 - \sum_{k=3}^{\infty} S^{(k)}[q^*](q - q^*)^k]\right)}. \tag{75}$$

The $\exp(NS[q^*])$ terms canceled in both numerator and denominator. We let the variable $q - q^* = \frac{1}{\sqrt{N}}\delta$. After this change of variable, we have

$$\langle \mathcal{O}(q) \rangle = \frac{\int d\delta \exp\left(-\frac{1}{2}S''[q^*]\delta^2 - \sum_{k=3}^{\infty} N^{1-k/2}S^{(k)}[q^*]\delta^k\right) \mathcal{O}(q^* + N^{-1/2}\delta)}{\int d\delta \exp\left(-\frac{1}{2}S''[q^*]\delta^2 - \sum_{k=3}^{\infty} N^{1-k/2}S^{(k)}[q^*]\delta^k\right)}. \tag{76}$$

We note that all the higher order derivatives ($k \geq 3$) are suppressed by at least $N^{-1/2}$ compared to the quadratic term. Letting $U = \sum_{k=3}^{\infty} N^{1-k/2}S^{(k)}[q^*]\delta^k$ represent the *perturbed potential*, we can Taylor expand the exponential around the Gaussian unperturbed potential $\exp\left(-\frac{1}{2}\delta^2 S''[q^*]\right)$. We let $\langle \mathcal{O}(\delta) \rangle_0 = \mathbb{E}_{q \sim \mathcal{N}(0, S''[q^*]^{-1})} \mathcal{O}(\delta)$ represent an average over this unperturbed potential

$$\langle \mathcal{O}(q) \rangle = \frac{\int d\delta \exp\left(-\frac{1}{2}S''[q^*]\delta^2\right)[1 - U + \frac{1}{2}U^2 + ...]\mathcal{O}(q)}{\int d\delta \exp\left(-\frac{1}{2}S''[q^*]\delta^2\right)[1 - U + \frac{1}{2}U^2 + ...]} \tag{77}$$

$$= \frac{\langle \mathcal{O}(q) \rangle_0 - \langle \mathcal{O}(q)U \rangle_0 + \frac{1}{2}\langle \mathcal{O}(q)U^2 \rangle_0 + ...}{1 - \langle U \rangle_0 + \frac{1}{2}\langle U^2 \rangle_0 + ...}. \tag{78}$$

Truncating each series in numerator and denominator at a certain order in $1/N$ gives a Pade-Approximant to the full observable average (Bender et al., 1999). Alternatively, this can be expressed in terms of a cumulant expansion (Kardar, 2007)

$$\langle \mathcal{O} \rangle = \sum_{k=0}^{\infty} \frac{(-1)^k}{k!} \langle \mathcal{O}U^k \rangle_0^c, \tag{79}$$

where $\langle \mathcal{O}U^k \rangle_0^c$ are the *connected* correlations, or alternatively the *cumulants*. The first two connected correlations have the form

$$\langle \mathcal{O}U \rangle_0^c = \langle \mathcal{O}U \rangle_0 - \langle \mathcal{O} \rangle \langle U \rangle_0$$
$$\langle \mathcal{O}U^2 \rangle_0^c = \langle \mathcal{O}U^2 \rangle_0 - 2\langle \mathcal{O}U \rangle_0 \langle U \rangle_0 - \langle \mathcal{O} \rangle_0 \langle U^2 \rangle_0 + 2\langle \mathcal{O} \rangle \langle U \rangle_0^2. \tag{80}$$

Using Stein's lemma, we can now attempt to extract the leading $O(N^{-1})$ behavior from each of these terms. First, we will note the following useful identity which follows from Stein's Lemma

$$\langle \mathcal{O}(q)\delta^k \rangle = N^{k/2} \langle \mathcal{O}(q)(q - q^*)^k \rangle \tag{81}$$

$$= N^{k/2-1}[S'']^{-1}[(k-1)\langle \mathcal{O}(q)(q - q^*)^{k-2} \rangle + \langle \mathcal{O}'(q)(q - q^*)^{k-1} \rangle]$$

$$= (k-1)[S'']^{-1}\langle \mathcal{O}(q)\delta^{k-2} \rangle + \frac{1}{\sqrt{N}}[S'']^{-1}\langle \mathcal{O}'(q)\delta^{k-1} \rangle. \tag{82}$$

Using these this fact, we can find the first few correlation functions of interest

$$\langle \mathcal{O}(q)U \rangle = 3N^{-1}S^{(3)}[S'']^{-2} \langle \mathcal{O}'(q) \rangle_0 + 3N^{-1}S^{(4)}[S'']^{-2} \langle \mathcal{O}(q) \rangle_0 + O(N^{-2})$$

$$\langle \mathcal{O}(q)U^2 \rangle = 15N^{-1}[S^{(3)}]^2[S'']^{-3} \langle \mathcal{O}(q) \rangle + O(N^{-1}). \tag{83}$$

Thus, the leading order Pade-Approximant has the form

$$\langle \mathcal{O}(q) \rangle = \frac{\langle \mathcal{O} \rangle_0 - \frac{3}{N}S^{(3)}[S'']^{-2} \langle \mathcal{O}'(q) \rangle_0 - \frac{3}{N}S^{(4)}[S'']^{-2} \langle \mathcal{O}(q) \rangle_0 + \frac{15}{2N}[S^{(3)}]^2[S'']^{-3} \langle \mathcal{O}(q) \rangle}{1 - \frac{3}{N}S^{(3)}[S'']^{-2} - \frac{3}{N}S^{(4)}[S'']^{-2} + \frac{15}{2N}[S^{(3)}]^2[S'']^{-3}}. \tag{84}$$

### H.1.1 DMFT Action Expansion

The logic of the previous section can be extended to our DMFT. We first redefine the action as its negation $S \to -S$ to simplify the argument. Concretely, this action $S[q]$ defines a Gibbs measure over the order parameters $q$ which we can use to compute observable averages

$$\langle \mathcal{O}(q) \rangle = \frac{\int \exp(-NS[q]) \mathcal{O}(q)}{\int \exp(-NS[q])} \tag{85}$$

As before, one can Taylor expand the action around the saddle point $q^*$

$$S[q] \sim S[q^*] + \frac{1}{2}(q - q^*)\nabla^2 S[q^*](q - q^*) + ... \tag{86}$$

As before, the linear term vanishes since $\nabla_q S[q^*] = 0$ at the saddle point $q^*$. We again change variables to $\delta = \sqrt{N}(q - q^*)$ and express the average as

$$\langle \mathcal{O} \rangle = \frac{\int d\delta \exp\left(-\frac{1}{2}\delta^\top \nabla^2 S\delta + U(\delta)\right) \mathcal{O}(\delta)}{\int d\delta \exp\left(-\frac{1}{2}\delta^\top \nabla^2 S\delta + U(\delta)\right)}$$

$$= \sum_{k=0}^{\infty} \frac{(-1)^k}{k!} \langle \mathcal{O}U^k \rangle_0^c \tag{87}$$

where $\langle \rangle_0$ denotes a Gaussian average over $q \sim \mathcal{N}(q^*, \frac{1}{N}[\nabla^2 S]^{-1})$.

### H.2 Hessian Components of DMFT Action

To gain insight into the Hessian, we will first restrict our attention to the subset of Hessian entries related to $\Phi^\ell, \hat{\Phi}^\ell$. We again adopt a multi-index notation $\boldsymbol{\mu} = (\mu, \nu, t, s)$ so that $\Phi^\ell_{\boldsymbol{\mu}} = \Phi^\ell_{\mu\nu}(t, s)$

$$\frac{\partial^2 S}{\partial \Phi^\ell_{\boldsymbol{\mu}} \partial \Phi^{\ell'}_{\boldsymbol{\mu}'}} = 0$$

$$\frac{\partial^2 S}{\partial \Phi^\ell_{\boldsymbol{\mu}} \partial \hat{\Phi}^{\ell'}_{\boldsymbol{\mu}'}} = \delta_{\ell,\ell'}\delta_{\boldsymbol{\mu},\boldsymbol{\mu}'} - \delta_{\ell',\ell+1}\frac{\partial}{\partial \Phi^\ell_{\boldsymbol{\mu}}}\Phi^{\ell+1}_{\boldsymbol{\mu}'}$$

$$\frac{\partial^2 S}{\partial \hat{\Phi}^\ell_{\boldsymbol{\mu}} \partial \hat{\Phi}^{\ell'}_{\boldsymbol{\mu}'}} = \delta_{\ell,\ell'}\left[\langle \phi(h^\ell_\mu(t))\phi(h^\ell_\nu(s))\phi(h^\ell_{\mu'}(t'))\phi(h^\ell_{\nu'}(s')) \rangle - \Phi^\ell_{\boldsymbol{\mu}}\Phi^\ell_{\boldsymbol{\mu}'}\right].$$

The first equation follows from the fact that $\hat{\chi}$ has vanishing moments due to the normalization of the probability distribution induced by $\mathcal{Z}^\ell$. Similarly, for the $G, \hat{G}$ kernels we have

$$\frac{\partial^2 S}{\partial G^\ell_{\boldsymbol{\mu}} \partial G^{\ell'}_{\boldsymbol{\mu}'}} = 0$$

$$\frac{\partial^2 S}{\partial G^\ell_{\boldsymbol{\mu}} \partial \hat{G}^{\ell'}_{\boldsymbol{\mu}'}} = \delta_{\ell,\ell'}\delta_{\boldsymbol{\mu},\boldsymbol{\mu}'} - \delta_{\ell',\ell-1}\frac{\partial}{\partial G^\ell_{\boldsymbol{\mu}}}G^{\ell-1}_{\boldsymbol{\mu}'}$$

$$\frac{\partial^2 S}{\partial \hat{\Phi}^\ell_{\boldsymbol{\mu}} \partial \hat{\Phi}^{\ell'}_{\boldsymbol{\mu}'}} = \delta_{\ell,\ell'}\left[\langle g^\ell_\mu(t)g^\ell_\nu(s)g^\ell_{\mu'}(t')g^\ell_{\nu'}(s') \rangle - G^\ell_{\boldsymbol{\mu}}G^\ell_{\boldsymbol{\mu}'}\right].$$

Proceeding in a similar manner, we can compute all off-diagonal components such as $\frac{\partial^2 S}{\partial \Phi \partial \hat{G}}$ and $\frac{\partial^2 S}{\partial \hat{\Phi} \partial \hat{G}}$. Once all entries are computed, one can seek an inverse of the Hessian to obtain the covariance of the order parameters.

### H.3 SINGLE SAMPLE NEXT-TO-LEADING ORDER PERTURBATION THEORY

In order to obtain exact analytical expressions, we will consider $L$-hidden layer ReLU and linear neural networks in the lazy regime trained on a single sample with $K^x = \frac{|\boldsymbol{x}|^2}{D} = 1$. To ensure preservation of norm, we will use $\phi(h) = \sqrt{2}h\Theta(h)$ for ReLU and $\phi(h) = h$ for linear networks. First, we note that in either case, the infinite width saddle point equations give

$$\Phi^\ell = \left\langle \phi(h)^2 \right\rangle_{h \sim \mathcal{N}(0, \Phi^{\ell-1})} = \Phi^{\ell-1} \;,\; \Phi^0 = 1$$

$$G^\ell = \left\langle \dot{\phi}(h)^2 z^2 \right\rangle_{h,z} = G^{\ell+1} \;,\; G^{L+1} = 1$$

$$\implies \Phi^\ell = 1 \;,\; G^\ell = 1 \;,\; \forall \ell \in [1, ..., L]. \tag{88}$$

At large but finite width, the kernels therefore fluctuate around this typical mean value of $\Phi^\ell = 1$ and $G^\ell = 1$. We now compute the necessary ingredients to invert the Hessian

$$V_\phi = \left\langle \phi(h)^4 \right\rangle - \Phi^2 = \begin{cases} 5 & \text{ReLU} \\ 2 & \text{Linear} \end{cases}$$

$$V_g = \left\langle \dot{\phi}(h)^4 z^4 \right\rangle - G^2 = \begin{cases} 5 & \text{ReLU} \\ 2 & \text{Linear} \end{cases}. \tag{89}$$

Next, we compute the sensitivity of each layer's kernel to the previous layer

$$\frac{\partial}{\partial \Phi^\ell} \Phi^{\ell+1} = 1 \;,\; \frac{\partial}{\partial G^{\ell+1}} G^\ell = 1. \tag{90}$$

First, let's analyze the marginal covariance statistics for $\boldsymbol{\Phi} = \text{Vec}\{\Phi^\ell\}_{\ell=1}^L$ and $\hat{\boldsymbol{\Phi}} = \text{Vec}\{\hat{\Phi}^\ell\}_{\ell=1}^L$. We note that the DMFT action has Hessian components

$$\boldsymbol{H}_\Phi = \begin{bmatrix} \nabla^2_{\boldsymbol{\Phi}} S & \nabla^2_{\boldsymbol{\Phi}\hat{\boldsymbol{\Phi}}} S \\ \nabla^2_{\hat{\boldsymbol{\Phi}}\boldsymbol{\Phi}} S & \nabla^2_{\hat{\boldsymbol{\Phi}}} S \end{bmatrix} = \begin{bmatrix} 0 & \boldsymbol{U} \\ \boldsymbol{U}^\top & V_\phi \boldsymbol{I} \end{bmatrix} \;,\; \boldsymbol{U} = \begin{bmatrix} 1 & -1 & 0 & ... & 0 & 0 \\ 0 & 1 & -1 & ... & 0 & 0 \\ \vdots & \vdots & \ddots & \ddots & \vdots & \vdots \\ 0 & 0 & 0 & ... & 1 & -1 \\ 0 & 0 & 0 & ... & 0 & 1 \end{bmatrix}. \tag{91}$$

We seek a (physical) inverse $\boldsymbol{C}$ which has vanishing lower diagonal entry, indicating zero variance in the dual order parameters $\hat{\boldsymbol{\Phi}}$. This gives us the following linear equations

$$\boldsymbol{H}_\Phi \boldsymbol{C} = \begin{bmatrix} 0 & \boldsymbol{U} \\ \boldsymbol{U}^\top & V_\phi \boldsymbol{I} \end{bmatrix} \begin{bmatrix} \boldsymbol{C}_{11} & \boldsymbol{C}_{12} \\ \boldsymbol{C}_{12}^\top & 0 \end{bmatrix} = \begin{bmatrix} \boldsymbol{I} & 0 \\ 0 & \boldsymbol{I} \end{bmatrix}$$

$$\implies \boldsymbol{U}\boldsymbol{C}_{12}^\top = \boldsymbol{I} \;,\; \boldsymbol{U}^\top \boldsymbol{C}_{11} + V_\phi \boldsymbol{C}_{12}^\top = 0. \tag{92}$$

The relevant entry is $\boldsymbol{C}_{11} = -V_\phi [\boldsymbol{U}^\top]^{-1} \boldsymbol{U}^{-1}$. This matrix has the form

$$\boldsymbol{C}_{11} = -V_\phi \begin{bmatrix} 1 & 0 & 0 & ... & 0 \\ 1 & 1 & 0 & ... & 0 \\ 1 & 1 & 1 & ... & 0 \\ \vdots & \vdots & \ddots & \ddots & \vdots \\ 1 & 1 & 1 & ... & 1 \end{bmatrix} \begin{bmatrix} 1 & 1 & 1 & ... & 1 \\ 0 & 1 & 1 & ... & 1 \\ 0 & 0 & 1 & ... & 1 \\ \vdots & \vdots & \ddots & \ddots & \vdots \\ 0 & 0 & 0 & ... & 1 \end{bmatrix} = -V_\phi \begin{bmatrix} 1 & 1 & 1 & ... & 1 \\ 1 & 2 & 2 & ... & 2 \\ 1 & 2 & 3 & ... & 3 \\ \vdots & \vdots & \ddots & \ddots & \vdots \\ 1 & 2 & 3 & ... & L \end{bmatrix}. \tag{93}$$

Using the fact that the covariance is the negative of the Hessian inverse multiplied by $1/N$, we have the following covariance structure for $\{\Phi^\ell\}$

$$\text{Cov}(\Phi^\ell, \Phi^{\ell'}) = \frac{1}{N} V_\phi \min\{\ell, \ell'\}. \tag{94}$$

This result can be interpreted as the covariance of Brownian motion. Following an identical argument, we find

$$\text{Cov}(G^\ell, G^{\ell'}) = \frac{1}{N} V_g \min\{L + 1 - \ell, L + 1 - \ell'\}. \tag{95}$$

We verify these scalings against experiments below in Figure 7.

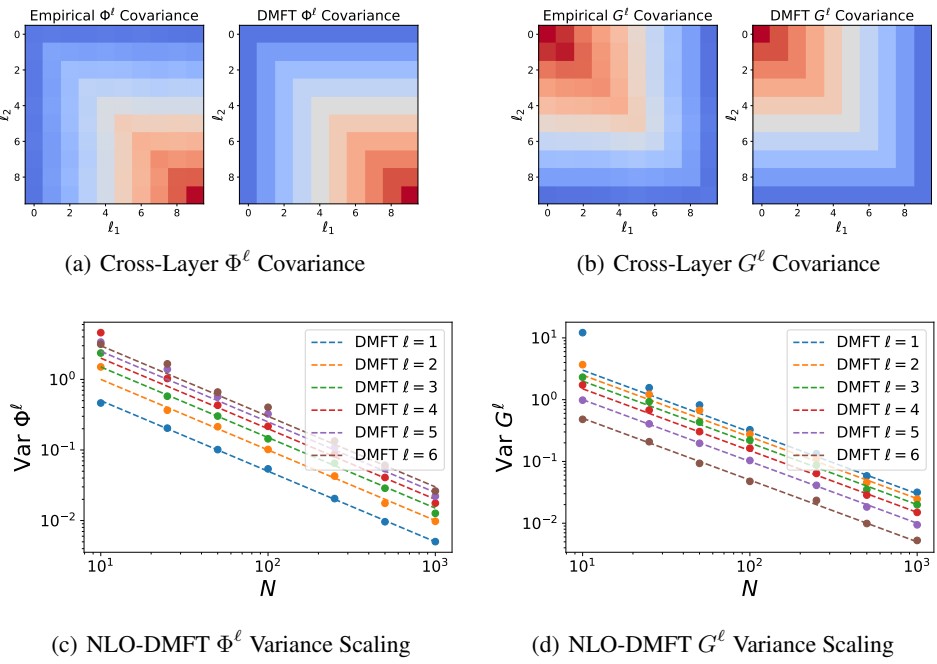

(a) Cross-Layer $\Phi^\ell$ Covariance

(b) Cross-Layer $G^\ell$ Covariance

(c) NLO-DMFT $\Phi^\ell$ Variance Scaling

(d) NLO-DMFT $G^\ell$ Variance Scaling

Figure 7: Verification of kernel fluctuations through next-to-leading-order (NLO) perturbation theory within DMFT formalism. (a) The cross layer covariance structure of $\{\Phi^\ell\}$ in a $L = 10$ hidden layer ReLU MLP. The empirical covariance was estimated by initializing a large number (500) of random networks and computing their $\Phi^\ell$ kernels. We see that variance for $\Phi^\ell$ increases as $\ell$ increases. (b) The cross-layer covariance structure of $\{G^\ell\}$. The variance of $G^\ell$ is larger for smaller $\ell$. (c) The predicted variance of $\Phi^\ell$ for different layer $\ell$ and widths $N$. All layers have variance scaling as $1/N$, consistent with NLO perturbation theory. (d) The scaling of $G^\ell$ variance.

