# OpenReview forum: "The Influence of Learning Rule on Representation Dynamics in Wide Neural Networks"
_ICLR.cc/2023/Conference — ICLR 2023 notable top 25%_

### Official Review · Reviewer_BwSf · 2022-10-24

**Confidence:** 4
**Correctness:** 3
**Technical Novelty And Significance:** 3
**Empirical Novelty And Significance:** 2
**Recommendation:** 8

**Clarity, Quality, Novelty And Reproducibility:**

The analytical results of this paper are an interesting tool to probe the learning dynamics of neural networks and the derivation is easily understandable and could be adapted with minimal change to different learning rules. While this computation is essentially the same as Bordelon & Pehlevan '22 here it's applied to a number of learning rules, so it's a significant contribution on its own.

The authors choose to call their analytical result dynamical mean field theory (DMFT in short). This name is slightly confusing as there are other works claiming to have DMFT for neural networks, where here DMFT is intended as an effective theory that maps the high dimensional dynamics of the weights to the low dimensional evolution of a set of order parameters. A modern derivation of this kind was done by Agoritsas et al. '17 (Arxiv:1710.04894) for the perceptron and then expanded by Mignacco et al. '20 (Arxiv:2006.06098). This approach was also recently proven rigorously by Celentano et al. '21 (Arxiv:2112.07572) and Gerbelot et al. '22 (Arxiv:2210.06591).
The main difference between this line of works and the paper at hand is in the treatment of the data, and while this is clear from the derivations it would be clarifying to point it out more explicitly in the main text.

The authors provide a code in the supplementary material reproducing the figures in the paper, and the numerical method proposed seems to converge without significant issues.

**Strength And Weaknesses:**

Strengths:
1) The paper is reasonably well-written and easy to follow.
2) The analytical procedure to obtain a theoretical characterisation of the dynamics is easily generalisable to different learning rules and architectures beyond what is shown in the paper.
3) The predictions of the theory are compared with experiments and the plots display for the most part quantitative agreement between theory and experiments.
4) The authors check numerically that the large width limit is approached polynomially fast in the width of the network.

Weaknesses:
1) The role of the data on the theory is not explicitly stated in the main test.
2) The authors propose a fixed point scheme to solve the self-consistent equations. Do you have a theoretical argument for its convergence and some bounds on the number of iterations required?
3) In many plots (for example figure 3a, blue line) there is a small disagreement between theory and experiments. What could be the reason for the discrepancy?
4) While the numerical checks of the finite width effects are very convincing, the theoretical argument for the finite width expansion seems not so transparent. Can you plot the finite width corrections to the theory?


**Summary Of The Paper:**

The paper studies the learning dynamics of neural networks in the mean field limit. The authors focus on gradient descent, Hebbian learning, feedback alignment and direct feedback alignment on networks of arbitrary depth, as well as gated linear networks. For these cases we are presented with a set of self-consistent stochastic differential equations to be solved numerically.
The self consistent stochastic equations simplify in the static limit and the linear network case and reduce to other results in the literature.
Specifically, the authors obtain an explicit characterisation of the dynamics of two layer linear networks.

**Summary Of The Review:**

The paper presents a set of self-consistent stochastic differential equations that describe the learning dynamics of deep networks for a number of learning rules. The paper is well-written and clearly details the analytical derivation as well as the numerics involved. The results are easily reproducible and a code is provided in the supplementary materials.

While the treatment of the dynamics and the phenomenology extracted from it are not completely novel it's a significant contribution that could be expanded in many directions

---

> ### Author Response · Authors · 2022-11-12
> **Response to Reviewer BwSf (Part 1)**
>
>
> We thank the reviewer for carefully reading our submission and providing useful suggestions to improve the paper. Below we try to address the main concerns and questions brought up in the review.
>
> *The role of the data on the theory is not explicitly stated in the main text.*
>
> We mention that the data gram matrix $K^x_{\mu\nu} = \frac{1}{D} x_\mu \cdot x_\nu$ gives the base case for $\Phi$ kernels $\Phi^0 = K^x$ below equation 4. To reiterate this and make it more clear in the main text results section, we added a line on page 4 below equation 5 (the main DMFT equations) and above Table 1 which states
>
> > "The dependence on data comes from the base case $\Phi^0_{\mu\nu}(t,s) = \frac{1}{D} x_\mu \cdot x_\nu$ and error signal $\Delta_\mu = - \frac{\partial L}{\partial f_\mu}$."
>
> *The authors propose a fixed point scheme to solve the self-consistent equations. Do you have a theoretical argument for its convergence and some bounds on the number of iterations required?*
>
> This is a great question! At the moment we have not been able to prove that the fixed point iteration converges, but have been thinking about ways to show that the iteration scheme is a contraction. If $k$ is the vector of order parameters and $k_{t+1} = F[k_t]$ is the fixed point iteration, then by **Banach Fixed-Point theorem** it would suffice to prove that $|F[k]-F[k']| < |k - k'|$. We have not yet figured out how to establish this for our DMFT solver since $F$ is a complicated set of integral operators and averaging procedure for the DMFT. One promising direction would be to assume the idealization of infinitely many Monte-carlo samples. We think this is a great open problem for future work.
>
> We added the following sentence at the end of the discussion
> > "Future work could provide theoretical convergence guarantees for our DMFT solver. "
>
> *In many plots (for example figure 3a, blue line) there is a small disagreement between theory and experiments. What could be the reason for the discrepancy?*
>
> At finite width and small $\gamma_0$, the initial condition of network predictions has large variance. DMFT predicts that all network outputs are identically zero at initialization. To get good agreement with the infinite width theory at small $\gamma_0$ one would need to increase the width of the network to enforce a good match in initial conditions. We note that the larger $\gamma_0$ experiments show good match in Figure 3a. This is shown quantitatively in Figure 3e where, for any width $N$ network, training with larger $\gamma_0$ gives kernel dynamics which more closely match the DMFT solutions.
>
> *While the numerical checks of the finite width effects are very convincing, the theoretical argument for the finite width expansion seems not so transparent. Can you plot the finite width corrections to the theory?*
>
> Yes, this is a good point. We added Appendix sections 8.2 and 8.3 where we provide an example of the mechanics of a next-to-leading order (NLO) perturbative calculation using the Hessian of the DMFT action.  In the new Figure 7, we plot the predictions of the $1/N$ NLO-DMFT perturbation theory against numerical experiments in a lazy deep ReLU network with one training sample. These idealizations allows us to analytically invert the Hessian of the DMFT action. We compute the covariance structure of $\Phi^\ell$ and $G^\ell$ on a simple sample, finding that
>
> $$\text{Cov}(\Phi^\ell,\Phi^{\ell'}) = \frac{1}{N} \min(\ell,\ell') \left[ \left< \phi(h)^4 \right> - \left< \phi(h)^2 \right>^2\right]$$
>
> This can be interpreted as a random walk / Brownian motion if the layer index is reinterpreted as time. Specifically for this single sample problem, we have
>
> $$\Phi^0 = 1 \ , \ \Phi^{\ell} = \Phi^{\ell-1} + \epsilon_\phi^\ell \ , \ \text{Var}(\epsilon_\phi^\ell) = \frac{1}{N}\left[ \left< \phi(h)^4 \right> - \left< \phi(h)^2 \right>^2\right]$$
>
> The $\epsilon^\ell$ are independent across layers so the variance of $\Phi^\ell$ accumulates on the forward pass (increasing $\ell$). Likewise for the $G^\ell$ kernels, we find
> $$\text{Cov}(G^\ell,G^{\ell'})=\frac{1}{N} \min( L+1-\ell, L+1-\ell' ) \left[ \left< g^4 \right> - \left< g^2 \right>^2 \right]$$
> which has interpretation
> $$ G^{L+1} = 1 \ , \ G^{\ell} = G^{\ell+1} +  \epsilon^{\ell}_g \ , \ \text{Var}(\epsilon^\ell_g) = \frac{1}{N} \left[ \left< g^4 \right> - \left< g^2 \right>^2 \right]$$
>
> We verify these analytical predictions against experiments in Figure 7 and find very precise agreement at sufficiently large $N$, (even $N \sim 20$ is wide enough to have finite size effects well predicted by the NLO perturbation theory).

---

> > ### Author Response · Authors · 2022-11-12
> > **Response to Reviewer BwSf (Part 2)**
> >
> >
> > *The authors choose to call their analytical result dynamical mean field theory (DMFT in short). This name is slightly confusing as there are other works claiming to have DMFT for neural networks, where here DMFT is intended as an effective theory that maps the high dimensional dynamics of the weights to the low dimensional evolution of a set of order parameters. A modern derivation of this kind was done by Agoritsas et al. '17 (Arxiv:1710.04894) for the perceptron and then expanded by Mignacco et al. '20 (Arxiv:2006.06098). This approach was also recently proven rigorously by Celentano et al. '21 (Arxiv:2112.07572) and Gerbelot et al. '22 (Arxiv:2210.06591). The main difference between this line of works and the paper at hand is in the treatment of the data, and while this is clear from the derivations it would be clarifying to point it out more explicitly in the main text.*
> >
> > Thank you for this point. We use DMFT exactly to mean "as an effective theory that maps the high dimensional dynamics of the weights to the low dimensional evolution of a set of order parameters" as the reviewer elegantly put. Our use of the term was inspired by the dynamical mean field theory of random recurrent networks, where a dynamical system with random weight matrix is characterized by a self-averaging correlation function (such as, https://journals.aps.org/prl/abstract/10.1103/PhysRevLett.61.259).
> >
> >
> > And we also agree that other works cited by the reviewer employ similar mathematical methods to describe the high dimensional dynamics of machine learning algorithms in terms of correlation and response functions. We further that one way our work differs is in the treatment of data. Our current proposed DMFT is about characterizing the typical dynamics over random initializations for a fixed dataset, as opposed to previous work which usually study typical behavior for high-dimensional random Gaussian data.
> >
> >
> > We added the following sentences to the related works section (top of last paragraph).
> >
> > > "Recent works have utilized DMFT techniques to analyze typical performance of algorithms trained on high-dimensional random data \citep{agoritsas2018out, mignacco2020dynamical, celentano2021high, gerbelot_2022}. In the present work, we do not average over random datasets, but rather over initial random weights and treat data as an input to the theory."

---

> > > ### Comment · Reviewer_BwSf · 2022-11-15
> > > **Score change**
> > >
> > > I thank the authors for the thorough response. All my concerns are addressed, and I change the score to 8

---

### Official Review · Reviewer_ZSh1 · 2022-10-24

**Confidence:** 3
**Correctness:** 4
**Technical Novelty And Significance:** 3
**Empirical Novelty And Significance:** 3
**Recommendation:** 8

**Clarity, Quality, Novelty And Reproducibility:**

# High-level remarks

- The paper is generally well-written, though some portions are hard to follow. The authors can probably improve the concerned lines of text easily. Overall, the paper is structured well.

- The authors include Jupyter notebooks with their experiments and the figure-generating code. This should make the work easy to reproduce.


# Detailed questions/suggestions

- In Equation (2), is there an extra gamma_0 that should not be there?
- In equation (3): why is there $\dot\phi$ in the line for GD? From the paragraph above equation (2), it seems that $\dot\phi$ is only used for GLNs.
- In equation (5), the authors use $z_\mu^l(t)$, which I believe is the pre-gradient signal from page 2 of https://arxiv.org/pdf/2205.09653 . Could the authors introduce this new symbol in their text as well, just before or after equation (5)?

- Figure (1) is for a 2-layer neural network. How well would DMFT approximate learning if the network were deeper?
- “Gausssian ” should be “Gaussian ”

- The end of this sentence in the appendix is missing: “this learning rule will continue to update the weights even once the task is fully learned, leading ”

- Equation (9) in Appendix B.1 needs a period at the end, as do a few other equations in the appendices.



**Strength And Weaknesses:**

# Strengths:

- The paper is relevant to a big portion of the ICLR community, and provides theory and insights that may result in follow-up research, and possible better learning rules.
- The papers release their code in the supplementary material. This includes code to generate the graphics included in the paper.
- The paper is well-written.

# Weaknesses:

- The empirical work done in the paper is on fairly small tasks, considering that many of the learning rules considered in the paper are supposed to work on networks as complex as the human brain.

- Some of the paper’s insights are not very surprising. For example, the fact that Hebbian learning converges slowly, or that it does not closely follow the gradient, are generally accepted as true in the community.


**Summary Of The Paper:**

The authors investigate an important and open question: how does the the learning rule (gradient descent, Feedback alignment, Direct Feedback Alignment, error-modulated Hebbian learning, Gated linear networks) influence the learning dynamics during neural network training. The authors focus on wide neural networks, and study these using dynamical mean field theory. Their approach allows them to make statements about the learning dynamics with different learning rules. Some of these statements are expected (but still good to formally analyze), e.g. that error-modulated Hebbian learning does not enable effective learning. Other statements are more surprising (e.g. that smaller initial feedback alignment later leads to better feature learning in the Feedback Alignment method).

**Summary Of The Review:**

The paper is relevant to a big portion of the ICLR community, and provides theory and insights that will likely result in follow-up research, and possible better learning rules.

While the paper is generally well-written, the authors should make small fixes/improvements during the rebuttal period.

In my opinion, the biggest weakness of the paper is that many of the findings that the paper works out are unsurprising. The authors could strengthen the paper by highlighting which of the findings are "common knowledge", but have not yet been investigated formally.

---

> ### Author Response · Authors · 2022-11-12
> **Response to Reviewer ZSh1 (Part 1)**
>
>
> We thank the reviewer for a detailed reading of our paper and for offering useful suggestions. We will now try to address the concerns identified in the review.
>
> *The empirical work done in the paper is on fairly small tasks, considering that many of the learning rules considered in the paper are supposed to work on networks as complex as the human brain.*
>
> We thank the reviewer for this critique. Our focus on small tasks was merely to allow efficient computation of our saddle point equations which require $O(T^3 P^3)$ operations for $T$ timesteps and $P$ training examples. This somewhat limits the scale of problems we can solve in the rich regime. However, below we describe some special cases of our theory which can be solved efficiently.
>
> 1. In lazy limit $\gamma_0 \to 0$ we can solve kernel regression problem with the effective NTK $K_{\mu\nu}$ with $O(P^3)$ time complexity. The performance of such models on realistic learning tasks (like CIFAR-10) would be equivalent to the infinite width network performance reported here (https://arxiv.org/abs/2007.15801). For instance, Figure 1 here (https://arxiv.org/abs/2007.15801) reports infinite width GD test error, which would be identical to training $\rho=1$ FA and GD in the lazy infinite width limit. The $\rho=0$ FA, DFA, and Hebb rules at infinite width would all have performances identical to the infinite width Bayesian network.
> 2. In deep linear network trained on whitened data, we can solve problem in $O(T^3)$ time complexity, regardless of sample size.
>
>
> We added Appendix section D.1 to comment on the first point and to relate our limits to this prior empirical study
>
> > "We note that, while the DMFT equations on $P$ datapoints and $T$ timesteps require $O(P^3 T^3)$ time complexity to solve in the rich regime, the lazy limit gives neural network predictions in $O(P^3)$ time, since the predictor can be obtained by solving a linear system of equations. The performance of these lazy limit kernels on realistic tasks would match the performances reported by \citet{lee2020finite}. Specifically, GD and $\rho = 1$ FA would match the test accuracy reported for *infinite width GD*, while $\rho=0$ FA, DFA, and Hebbian rules would match *infinite width Bayesian networks* in Figure 1 of \citet{lee2020finite}."
>
> *Some of the paper’s insights are not very surprising. For example, the fact that Hebbian learning converges slowly, or that it does not closely follow the gradient, are generally accepted as true in the community.*
>
> We appreciate this comment. First, we hope the reviewer agrees that a result being surprising is not a requirement for it being useful/insightful/interesting. Part of our contribution is giving an exact mathematical description of how, for instance, Hebbian learning is different than GD or FA. While it is widely appreciated that Hebb rule does not follow a gradient, we mathematically characterize what impact this has on the training dynamics and learned representations.
>
> Second, we did find many aspects of our analysis of the Hebb rule as interesting and surprising. Our analysis of Hebbian learning and feedback alignment at infinite width in the lazy limit $\gamma_0 \to 0$ revealed that these learning rules cannot be distinguished from regression with the initial NNGP kernel $\Phi^L_{\mu\nu}$. We did not expect this result at the beginning of the project. In addition, we found it interesting that in the rich regime $\gamma_0 > 0$ the Hebb rule can still achieve zero train error but learns very different feature kernels than FA or GD. In addition, we discuss in Appendix G various modifications to the classic Hebb rule, some of which display nontrivial behaviors. For example, an update of the form $\frac{d}{dt} W^\ell = \sum_\mu \Delta_\mu \phi(h^{\ell+1}_\mu) \phi(h^\ell_\mu)^\top$ is unable to learn examples with negative targets ($y_\mu < 0$) in the rich $\gamma_0 > 0$ regime. This motivated our study of the two-powers of error Hebb rule $\frac{d}{dt} W^\ell = \sum_\mu \Delta_\mu^2 \ \phi(h^{\ell+1}_\mu) \phi(h^\ell_\mu)^\top$.
>
> We also discovered many other things that were not obvious to us. For example, all considered learning rules can be characterized with an effective NTK, that gradient/pseudo-gradient alignment improves with feature learning strength $\gamma_0$, and that $\rho-FA$ rule exhibits more representation learning for small initial correlation $\rho$ between feedforward and feedback weights. Further, it wasn't obvious that the infinite width limits of these networks would converge to a kernel limit, and what those kernels would be.
>
>
> To acknowledge pre-existing expectations about the Hebb rule, we modified our contribution list. When listing our contributions, we added the phrase "as expected" when discussing how Hebb rule does not tend to align to task structure
> > ... while Hebb networks,  as expected, do not exhibit task relevant adaptation of feature kernels, but rather evolve according to the input statistics."

---

> > ### Author Response · Authors · 2022-11-12
> > **Response to Reviewer ZSh1 (Part 2)**
> >
> >
> > ### Detailed Questions/Suggestions
> >
> > *In Equation (2), is there an extra gamma_0 that should not be there?*
> >
> > Thank you for this qurestion. No, currently equation 2 is written as intended. We include the additional factor of $\gamma_0$ so that the output of the network $f_\mu$ has initial derivative $\frac{d}{dt} f_\mu = O_{\gamma_0}(1)$. This corresponds to choosing a learning rate proportional to $\gamma_0^2 N$ which gives equation 2. We can verify that this gives the correct scaling for GD
> >
> > $\frac{d}{dt} f_\mu = \frac{df_\mu}{d\theta} \cdot \frac{d\theta}{dt} = \gamma_0^2 N \sum_{\nu}\Delta_\nu \frac{\partial f_\mu}{\partial \theta} \cdot \frac{\partial f_\nu}{\partial \theta}$
> >
> > Using the fact that
> >
> > $\frac{\partial f}{\partial W^\ell} = \frac{\partial f}{\partial h^{\ell+1}} \cdot \frac{\partial h^{\ell+1}}{\partial W^{\ell}} = \frac{1}{\gamma_0 N} g^{\ell+1} \left[ \frac{1}{\sqrt N} \phi(h^\ell)^\top \right] = \frac{1}{\gamma_0 N^{3/2}} g^{\ell+1} \phi(h^\ell)$, we find
> >
> > $\frac{d}{dt} f_\mu = \gamma_0^2 N \sum_{\nu} \Delta_{\nu} \sum_{\ell} \frac{1}{\gamma_0^2 N^3} (g^{\ell+1}_\mu \cdot g^{\ell+1}_\nu ) (\phi( h^{\ell}  ) \cdot \phi( h^{\ell} ) ) = O(1)$
> >
> > Thus we see that the output of the network evolves in $O(1)$ time but the internal features evolve in time $\sim 1/\gamma_0$. This is what allows us to alter the strength of feature learning while keeping the initial function dynamics the same.
> >
> > In the Appendix B.1 (page 14), we added the following sentence explaining this choice
> >
> > > "The inclusion of the prefactor $\frac{\gamma_0}{\sqrt N}$ in the weight dynamics ensures that $\frac{d}{dt} f = O_{\gamma_0,N}(1)$ and $\frac{d}{dt} h^\ell = O_{\gamma_0,N}(\gamma_0)$ at initialization (Chizat & Bach 2019, Bordelon & Pehlevan 2022)."
> >
> > *In equation (3): why is there $\dot\phi$ in the line for GD? From the paragraph above equation (2), it seems that $\dot\phi$ is only used for GLNs.*
> >
> > Thank you again for your careful reading. This $\dot\phi(h)= \frac{d}{dh} \phi(h)$ in equation (3) comes from the back-prop recursion for $g^\ell$ which is
> >
> > $g^{\ell} = N \gamma_0 \frac{\partial f}{\partial h^\ell} = \left[\frac{\partial h^{\ell+1}}{\partial h^{\ell}} \right]^\top \frac{\partial f}{\partial h^{\ell+1}} N\gamma_0$
> >
> > We use the fact that $\frac{\partial h_i^{\ell+1}}{\partial h^\ell_j} = \frac{1}{\sqrt N} W_{ij} \dot\phi(h_j^\ell)$ and $\frac{\partial f}{\partial h^{\ell+1}} N \gamma_0 = g^{\ell+1}$. Thus
> >
> > $g^{\ell} = \dot\phi(h^\ell) \odot \frac{1}{\sqrt N} W^{\ell+1,\top} g^{\ell+1}$
> >
> > Thank you for catching this. We see that we need the $\dot\phi$ to relate $g^{\ell}$ to $g^{\ell+1}$. To mimic this effect in the gated linear network, we use gating variable $\dot\phi(m)$ instead of the prefactor $\dot\phi(h)$. For ReLU networks, this would correspond to step-function gating $\dot\phi(m) = \Theta(m)$.
> >
> > *In equation (5), the authors use $z^\ell_\mu$, which I believe is the pre-gradient signal from page 2 of https://arxiv.org/pdf/2205.09653 . Could the authors introduce this new symbol in their text as well, just before or after equation (5)?*
> >
> > We added a sentence below equation 5 defining the pregradient $z^\ell$.
> >
> > *Figure (1) is for a 2-layer neural network. How well would DMFT approximate learning if the network were deeper?*
> >
> > This is a good question. We actually present example simulations for deeper networks:
> > 1. GD, GLN, FA, $\rho$-FA for depth 4 linear networks (see Figure 4)
> > 2. GD and DFA for depth 4 nonlinear networks (see Figure 3)
> >
> > We think that Figure 3(e) is representative of DMFT's accuracy. We also implements all learning rules for 2 layer nonlinear networks (see Figure 1).
> >
> > We also have developed an implementation of $\rho$-FA for nonlinear deep (more than 2 layer) networks. We added a plot of the predicted and experimental $\Phi^\ell$ kernels for $\rho=0$ and $\rho=1$ depth $3$ tanh network is provided in the Appendix Figure 6.
> >
> >
> > #### Typos
> >
> > *“Gausssian ” should be “Gaussian ”*
> >
> > *The end of this sentence in the appendix is missing: “this learning rule will continue to update the weights even once the task is fully learned, leading ”*
> >
> > *Equation (9) in Appendix B.1 needs a period at the end, as do a few other equations in the appendices.*
> >
> > We have addressed these typos. Thank you for a careful reading.

---

> > > ### Author Response · Authors · 2022-11-12
> > > **Response to Reviewer ZSh1 (Part 3)**
> > >
> > >
> > > ### Summary of Review
> > > *The paper is relevant to a big portion of the ICLR community, and provides theory and insights that will likely result in follow-up research, and possible better learning rules.*
> > >
> > > Thank you for these words of support.
> > >
> > > *While the paper is generally well-written, the authors should make small fixes/improvements during the rebuttal period.*
> > >
> > > To improve the paper we made the following improvements to the writing
> > > 1. We added additional references to related works in the DMFT literature and the Cao et al 2020 paper on learning rules and representation dynamics.
> > > 2. We mentioned explicitly how the structure of the data enters into the theory below equation 5.
> > > 3. We simplified the exposition in the bullet point list in Section 3.1.
> > > 4. We acknowledged additional limitations (lack of a theory of convergence of our fixed point iteration scheme) in the last paragraph of the discussion.
> > > 5. We added a more detailed analytical computation of finite size effects in Appendix H and Figure 7.
> > >
> > > *In my opinion, the biggest weakness of the paper is that many of the findings that the paper works out are unsurprising. The authors could strengthen the paper by highlighting which of the findings are “common knowledge”, but have not yet been investigated formally.*
> > >
> > >
> > >
> > > We appreciate this comment. It is hard to judge what is `common knowledge' to the community given the community is large. We have a sense of this through our own interactions with others, but, more importanly through publications. Our conclusion from these interactions was that we contribute many new findings here that were not ``common knowledge'' which was the reason why we wrote this paper. We list our novel contributions in page 2:
> > >
> > >
> > > > In summary, our novel contributions are the following:
> > > >1. We identify a class of learning rules for which function evolution is described by a dynamical effective Neural Tangent Kernel (eNTK). We provide a dynamical mean field theory (DMFT) for these learning rules which can be used to compute this eNTK. We show both theoretically and empirically that convergence to this DMFT occurs at large width $N$ with error $O(N^{-1/2})$.
> > > >2. We characterize precisely the inductive biases of infinite width networks in the lazy limit by computing their eNTKs at initialization. We generalize FA to allow partial correlation between the feedback weights and initial feedforward weights and show how this alters the eNTK.
> > > >3. We then study the *rich* regime so that the features are allowed to adapt during training. In this regime, the eNTK is dynamical and we give a DMFT to compute it.  For deep linear networks, the DMFT equations close algebraically, while for nonlinear networks we provide a numerical procedure to solve them.
> > > > 4. We compare the learned features and dynamics among these rules, analyzing the effect of richness, initial feedback correlation, and depth. We find that rich training enhances gradient-pseudogradient alignment for both FA and DFA. Counterintuitively, smaller initial feedback correlation generates more dramatic feature evolution for FA. The GLN networks have dynamics comparable to GD, while Hebb networks,  as expected, do not exhibit task relevant adaptation of feature kernels, but rather evolve according to the input statistics.
> > >
> > >
> > > Note that we added the phrase `as expected` to the last sentence following the reviewer's comments.

---

> ### Comment · Reviewer_ZSh1 · 2022-11-16
> **Update to initial review, following authors' response**
>
> Based on the authors' response, I change my review score to 8.
>
> The authors convinced me that more aspects of their work than I had mentally tallied earlier are surprising. The authors also explained parts of the paper that I had asked about and not fully understood earlier, and improved the overall clarity of the paper.

---

### Official Review · Reviewer_k9iv · 2022-10-25

**Confidence:** 3
**Correctness:** 3
**Technical Novelty And Significance:** 3
**Empirical Novelty And Significance:** 3
**Recommendation:** 8

**Clarity, Quality, Novelty And Reproducibility:**

### Clarity

The paper is hard to read due to the density of theoretical results, but it seems inevitable given the nature of the work. On the upside, the key take-home messages of the theory are clearly formulated.

### Quality

The technical side of the paper seems solid (although I haven't carefully checked the appendix). The experiments are standard for an NTK paper.

Question 1: how width-dependent are the specific results on non-SGD rules (e.g., for feedback alignment)? Fig. 1e-f and Fig. 3d show that the experimental kernels more or less match the DFMT ones for large networks, but would the *structure* (rather than the precise match) change for e.g. N=100? If yes, would that imply different predictions for networks far from the rich NTK regime?

Question 2: are any of the conclusions require the rich training regime? $\rho$-FA result is highlighted as being rich regime-specific, but it seems that the lazy regime already tells us that $\rho=0$ results in less feature learning than non-zero $\rho$.

### Novelty

Math-wise, the paper seems to be a straightforward (but interesting) extension of [Bordelon & Pehlevan, 2022]. Are there other technical contributions besides re-deriving DFMT for alternative learning rules?

There's one very relevant background citation worth mentioning (doesn't change the contribution):
Characterizing emergent representations in a space of candidate learning rules for deep networks
Y Cao, C Summerfield, A Saxe

### Reproducibility
The code is provided but I did not run it.


**Strength And Weaknesses:**

### Strengths

The paper clearly theoretical differences between solutions found by different learning rules.

The theoretical analysis of learning rules' dynamics is important for determining how the brain learns.

The calculations are written in detail and can be repeated for other learning rules.

### Weaknesses

The paper seems to be a straightforward application of [Bordelon & Pehlevan, 2022] to other learning rules.

It's not clear to me if the rich NTK regime contributes to any of the final conclusions.

**Summary Of The Paper:**

The paper studies several biologically (more) plausible alternatives to backprop in the lazy/rich NTK regime. This is done using the recent developments in dynamic mean field theory for backprop, and produces non-trivial predictions for activity of those learning rules in wide networks.

**Summary Of The Review:**

Good paper with a mostly technical contribution. Some parts could be improved/better explained, but overall this could be an important result for biologically plausible learning studies.

---

> ### Author Response · Authors · 2022-11-12
> **Response to Reviewer k9iv (Part 1)**
>
>
> We thank the reviewer for their careful reading of our paper. We also thank the reviewer for their positive comments. Below we address the main weaknesses and questions brought up in the review.
>
> ### Weaknesses
> *The paper seems to be a straightforward application of [Bordelon & Pehlevan, 2022] to other learning rules.*
>
> We thank the reviewer for allowing us to explain the novelty of our contribution compared to the prior work of [Bordelon & Pehlevan 2022]. While our calculation certainly relies on the ideas and the framework proposed in [Bordelon & Pehlevan, 2022], we would like to stress the novelty of our present submission in extending this framework to these alternative learning rules.
> 1. We had to recognize that other learning rules have an effective NTK which depends on the $\tilde{G}^\ell$ order-parameter. Our calculation required including new fields (such as pseudo-gradient fields $\tilde{g}^\ell$) and new correlation and response functions $\{\tilde{G}^\ell, \tilde{\tilde{G}}^\ell, C^\ell, D^\ell \}$ so that we could find the effective NTK dynamics.
> 2. Unlike the work of [Bordelon & Pehlevan] these DMFT saddle point equations have 3 Gaussian sources $\{ u^\ell, r^\ell, v^\ell \}$ in each layer $\ell$, the $r^\ell, v^\ell$ are correlated.
> 3. We also provided exact solutions to certain learning rules in linear networks (Figure 5), which required identifying conservation laws for each learning rule (Appendix F).
> 4. Lastly, we have added exact finite size correction results for deep ReLU networks on a single sample in the Appendix H.2 and H.3 and Figure 7, which go beyond the formal expressions of [Bordelon & Pehlevan, 2022]. Please also see our detailed response to Reviewer 4.
>
> *It’s not clear to me if the rich NTK regime contributes to any of the final conclusions.*
>
> We thank the reviewer for allowing us to clarify why we felt the need to study the rich regime. In lazy regime many of these learning rules ($\rho=0$ FA, DFA, Hebb) do not get adequate feedback signals in early layers to benefit from depth. In this lazy regime, the dynamics of all of these learning rules are equivalent to the dynamics obtained by training only the last layer weights. To see any behavior different than training and prediction with the NNGP kernel $\Phi^L$, the gradient/pseudo-gradient alignment $\tilde{G}^\ell$ must evolve during training. This "alignment" can only increase in the rich regime. This is also a regime of interest since in this setting, representations can adapt to the learning task. This alignment is indeed what we plot in Figure 1c, Figure 3c, Figure 4b. We observe, for example, in Figure 3c that more representation learning leads to better alignment in DFA.
>
>
> ### Clarity
>
> *The paper is hard to read due to the density of theoretical results, but it seems inevitable given the nature of the work. On the upside, the key take-home messages of the theory are clearly formulated.*
>
> We agree that the paper is currently dense but we aimed to clearly state our primary results in the main text with only detailed computations relegated to the Appendix.

---

> > ### Author Response · Authors · 2022-11-12
> > **Response to Reviewer k9iv (Part 2)**
> >
> >
> > ### Questions
> >
> > *Question 1: how width-dependent are the specific results on non-SGD rules (e.g., for feedback alignment)? Fig. 1e-f and Fig. 3d show that the experimental kernels more or less match the DFMT ones for large networks, but would the structure (rather than the precise match) change for e.g. N=100? If yes, would that imply different predictions for networks far from the rich NTK regime?*
> >
> > This is a great question. We think that all learning rules have similar finite size effects qualitatively, and in particular the scaling of these effects are identical, but that there can be small discrepancies across learning rules.
> >
> > For any of the considered learning rules, the discrepancy between infinite width kernels and the finite width $N$ DMFT kernels at initialization is provided in Figure 2 (d,e,f), which show that all kernels $\{\Phi^\ell, G^\ell\}_{\ell=1}^L, K$ have squared deviation from DMFT of size $\frac{1}{N}$.
> >
> > The above facts should also hold dynamically for all of the learning rules. We verify this in Figure 3e for DFA.
> >
> > In the revised submission, we provide a more principled theoretical approach. In Appendix H we attempt to provide a more in-depth discussion of the finite size effects, and show that the kernels effectively fluctuate (over inits) around their infinite width values with variance $\frac{1}{N} [- \nabla^2 S]^{-1}$. In addition, for a single training sample and ReLu activation function, we can fully work out the exact 1/N corrections to the kernels in the lazy regime. Please also see our detailed response to Reviewer 4.
> >
> > *Question 2: are any of the conclusions require the rich training regime?*
> >
> > The rich regime is where the learning rules are most distinct. Below are some facts about the lazy limit which do not hold in the rich regime
> > 1. In the lazy limit $\rho=1$ FA is identical to GD.
> > 2. In the lazy limit $\rho=0$ FA, DFA, and Hebb rule are all identical and are equivalent to doing regression with the initial NNGP kernel $\Phi^L$.
> > 3. In the lazy regime, gradient-pseudogradient alignment measured with $\tilde{G}^\ell$ cannot change.
> >
> > In the rich regime, all of these algorithms exhibit distinct dynamics and converge to different representations. For instance
> > 1. In the rich regime, $\rho=1$ FA is distinct from GD (for example Figure 5a)
> > 2. In the rich regime, $\rho=0$ FA and DFA are distinct from Hebbian learning (for example Figure 1)
> > 3. In the rich regime, gradient-pseudogradient alignment $\tilde{G}^\ell$ can increase, allowing more layers to participate in prediction dynamics for $\rho=0$ FA, DFA and Hebb rules.
> >
> > Overall, the generically nonlinear dynamics of training deep (rather than shallow) networks will only appear in the rich regime.
> >
> > *$\rho$-FA result is highlighted as being rich regime-specific, but it seems that the lazy regime already tells us that $\rho=0$ results in less feature learning than non-zero $\rho$*
> >
> > While $\rho$-FA has a different effective NTK for each value of $\rho$ in the lazy limit $\gamma_0 \to 0$, none of these kernels evolve during training in the lazy limit. This motivated us to study rich feature learning regime, where we found surprisingly that smaller $\rho$ is associated with larger changes in the eNTK (Figure 4 c).
> >
> > ### Novelty
> >
> > *Math-wise, the paper seems to be a straightforward (but interesting) extension of [Bordelon & Pehlevan, 2022]. Are there other technical contributions besides re-deriving DFMT for alternative learning rules?*
> >
> > The main machinery of the path-integral DMFT is the same as the work of [Bordelon & Pehlevan], who we try to credit with ample citations throughout the work. Our novel technical contributions above and beyond their work include
> > 1. Establishing different learning rules have different infinite width limits.
> > 2. Identifying the definition of the effective NTK for a learning rule as $K = \sum_{\ell} \tilde{G}^{\ell+1} \Phi^\ell$.
> > 3. Showing the importance of the gradient pseudo-gradient correlation $\tilde{G}^\ell$ in learning dynamics for alternative learning rules
> > 4. Setting up a path-integral over a larger collection of fields for each layer
> > 5. Solving for a larger collection of order parameters
> > 6. Giving exact results for different learning rules in a simple two layer linear model.
> >
> > *There’s one very relevant background citation worth mentioning (doesn’t change the contribution): Characterizing emergent representations in a space of candidate learning rules for deep networks Y Cao, C Summerfield, A Saxe*
> >
> > We thank the reviewers for bringing this interesting work to our attention. We added this work to our discussion of relevant works in the introduction (bottom of page 2).
> >
> > > "Cao et al (2020) analyzed the kernel and loss dynamics of linear networks trained with learning rules from a space that includes GD, contrastive Hebbian, and predictive coding rules, showing strong dependence of hierarchical representations on learning rule."

---

> > > ### Comment · Reviewer_k9iv · 2022-11-15
> > > **Good response, keeping the same score (8)**
> > >
> > > Thank you for the response! It clarified the parts where I got confused; the results on finite-width corrections also make sense. I think it's a good paper with interesting contributions, so I'm leaving the score as it is (8).

---

### Official Review · Reviewer_6oec · 2022-10-27

**Confidence:** 2
**Correctness:** 4
**Technical Novelty And Significance:** 4
**Empirical Novelty And Significance:** 3
**Recommendation:** 8

**Clarity, Quality, Novelty And Reproducibility:**

Overall, the paper is clearly written. To my knowledge the contributions are novel.

**Strength And Weaknesses:**

Strengths:
- Nice use of the theory of infinite width neural networks to study learning dynamics in networks trained with variants of gradient descent.
- Identifies a number of interesting phenomena regarding similarities and differences between the representations learned using the different learning rules.

Weaknesses:
-  Analysis is limited to MLP networks

**Summary Of The Paper:**

This paper studies the learning dynamics of infinite-width networks trained using various biologically plausible variations to gradient descent, including feedback alignment, direct feedback alignment, error modulated Hebbian learning, and gated linear networks. The paper derives an effective Neural Tangent Kernel (NTK) that governs the learning dynamics. Through theory and simulation, the paper identifies a number of interesting phenomena about the learning dynamics of these biologically plausible learning rules.

**Summary Of The Review:**

Overall, I think this paper is an important contribution in our understanding of the learning dynamics of networks trained with biologically plausible (or at least biologically inspired) learning rules. Feedback Alignment and subsequent variants are an intriguing learning rule because they seem so far from gradient descent; this paper helps us understand when and why they work.

---

> ### Author Response · Authors · 2022-11-12
> **Response to Reviewer 6oec**
>
> We thank the reviewer for a careful reading of our paper and for appreciating our use of infinite width limits to probe representation dynamics of different learning rules. Below we address the main weakness pointed out by the reviewer.
>
> *Weakness: Analysis is limited to MLP networks*
>
> We thank the reviewer for pointing out the limitation of the present submission, which is its heavy focus on MLP architectures. We want to clarify that while we have not yet developed a solver for the DMFT equations for CNNs, we have derived the DMFT saddle point equations for CNNs (Appendix C.2). We mention this above equation (1) on page 3
>
> > "Other architectures such as multi-class outputs and CNN architectures with infinite channel count can also be analyzed as we show in the Appendix C."
>
> We would ideally like to develop a DMFT equation solver for other architectures like CNNs and RNNs but these are necessarily more computationally demanding.
>
> In the lazy regime $\gamma_0 \to 0$, all learning rules are governed by static CNN kernels which depend on the level of correlation between feedforward and feedback weights. These are much like the NNGP and NTK kernels for convolutional architectures (like those in these works https://openreview.net/forum?id=B1g30j0qF7, https://arxiv.org/abs/2007.15801).

---

> > ### Comment · Reviewer_6oec · 2022-12-01
> > **thanks**
> >
> > Dear authors, thanks for your response

---

### Author Response · Authors · 2022-11-12
**Global Response to Reviews: Summary of Changes**

We appreciate the detailed reading and thoughtful comments and suggestions from all of the reviewers. While we responded to each of the reviewer comments individually, we will provide a brief summary of changes made to the paper in response to the reviews.

1. Based on reviewer BwSf 's suggestion, we gave a detailed calculation of next to leading order (NLO) finite size effects in a simplified setting where analytical formulas for the kernel variances can be obtained. We calculate these corrections in the new Appendix H.2 and H.3 and show good agreement with experiment in the new Figure 7.
2. We mention explicitly the role of the data $(x_\mu, y_\mu)$ and define the pre-gradient signal $z$ below the statement of DMFT equations (equation $5$).
3. We add references to other works on dynamical mean field theory (DMFT) in machine learning literature. We discuss that many of these works characterize typical performance of algorithms on random data, while our study treats data as fixed and weights of the deep network as random.
4. We acknowledged our lack of a current proof of the fixed point iteration scheme for our solver and mentioned such a proof as a future direction in the discussion.
5. We added a reference to Cao et al 2020, which studied how learned hierarchical representations depend on learning rules in linear networks.
6. We added Appendix D.1 where we provide links to relevant large-scale empirical work on NNGP and NTK regression and explain that lazy infinite width networks for $\rho=0$ FA, Hebb and DFA would match the performance of NNGP regression and that $\rho=1$ FA and GD would match the performance of NTK on realistic large scale tasks.
7. We added in Appendix B.1 (page 14) a sentence explaining our choice of scaling the learning rate with $\gamma_0^2$ so that $\frac{df}{dt} = O(1)$ and $\frac{d h^\ell}{dt} = O(\gamma_0)$ at initialization.
8. Throughout the paper, we tried simplifying sentences and eliminating redundant equations. We fixed typos and added appropriate punctuation to equations in the Appendix.

We think these changes have improved the paper and again want to thank the reviewers for suggesting them.

---

### Decision · Program_Chairs · 2023-01-20

**Decision:**

Accept: notable-top-25%

**Justification For Why Not Higher Score:**

I mainly opt for the spotlight because of the average score. The paper is interesting, but I do not see anything super remarkable to opt for oral.

**Justification For Why Not Lower Score:**

It could be just a poster, there is not particular reason for not giving that. Clearly not a reject.

**Metareview: Summary, Strengths And Weaknesses:**

All reviewers agree this paper should be accepted. I think the reviews summarize very well the strengths and weaknesses of the paper as well as points that the authors should include in the revised version. I think this will be a great addition to the conference.

**Note From Pc:**

if the above contains the word "oral" or "spotlight" please see: "oral" presentation means -> notable-top-5% and "spotlight" means -> notable-top-25%. As stated in our emails, we are disassociating presentation type from AC recommendations